# High-dimensional Asymptotics of Denoising Autoencoders

**Hugo Cui**
Statistical Physics of Computation Lab
Department of Physics
EPFL, Lausanne, Switzerland
`hugo.cui@epfl.ch`

**Lenka Zdeborová**
Statistical Physics of Computation Lab
Department of Physics
EPFL, Lausanne, Switzerland

## Abstract

We address the problem of denoising data from a Gaussian mixture using a two-layer non-linear autoencoder with tied weights and a skip connection. We consider the high-dimensional limit where the number of training samples and the input dimension jointly tend to infinity while the number of hidden units remains bounded. We provide closed-form expressions for the denoising mean-squared test error. Building on this result, we quantitatively characterize the advantage of the considered architecture over the autoencoder without the skip connection that relates closely to principal component analysis. We further show that our results accurately capture the learning curves on a range of real data sets.

## 1 Introduction

Machine learning techniques have a long history of success in denoising tasks. The recent breakthrough of diffusion-based generation [1, 2] has further revived the interest in denoising networks, demonstrating how they can also be leveraged, beyond denoising, for generative tasks. However, this rapidly expanding range of applications stands in sharp contrast to the relatively scarce theoretical understanding of denoising neural networks, even for the simplest instance thereof – namely Denoising Auto Encoders (DAEs) [3].

Theoretical studies of autoencoders have hitherto almost exclusively focused on data compression tasks using Reconstruction Auto Encoders (RAEs), where the goal is to learn a concise latent representation of the data. A majority of this body of work addresses *linear* autoencoders [4, 5, 6, 7]. The authors of [8, 9] analyze the gradient-based training of non-linear autoencoders with online stochastic gradient descent or in population, thus implicitly assuming the availability of an infinite number of training samples. Furthermore, two-layer RAEs were shown to learn to essentially perform Principal Component Analysis (PCA) [10, 11, 12], i.e. to learn a linear model. Ref. [13] shows that this is also true for infinite-width architectures. Learning in DAEs has been the object of theoretical investigations only in the linear case [14], while the case of non-linear DAEs remains theoretically largely unexplored.

**Main contributions–** The present work considers the problem of denoising data sampled from a Gaussian mixture by learning a two-layer DAE with a skip connection and tied weights via empirical risk minimization. Throughout the manuscript, we consider the high-dimensional limit where the number of training samples $n$ and the dimension $d$ are large ($n, d \to \infty$) while remaining comparable, i.e. $\alpha \equiv {}^n/d = \Theta(1)$. Our main contributions are:

- Leveraging the replica method, we provide sharp, closed-form formulae for the mean squared denoising test error (MSE) for DAEs, as a function of the sample complexity $\alpha$ and

37th Conference on Neural Information Processing Systems (NeurIPS 2023).

the problem parameters. We also provide a sharp characterization for other learning metrics including the weights norms, skip connection strength, and cosine similarity between the weights and the cluster means. These formulae encompass as a corollary the case of RAEs. We show that these formulae also describe quantitatively rather well the denoising MSE for *real* data sets, including MNIST [15] and FashionMNIST [16].

- We find that PCA denoising (namely denoising by projecting the noisy data along the principal component of the training samples) is widely sub-optimal compared to the DAE, leading to a MSE superior by a difference of $\Theta(d)$, thereby establishing that DAEs do *not* simply learn to perform PCA.

- Building on the formulae, we quantify the role of each component of the DAE architecture (skip connection and the bottleneck network) in its overall performance. We find that the two components have complementary effects in the denoising process –namely preserving the data nuances and removing the noise– and discuss how the training of the DAE results from a tradeoff between these effects.

The code used in the present manuscript can be found in the following repository.

**Related works**   *Theory of autoencoders–* Various aspects of RAEs have been studied, for example, memorization [17], or latent space alignment [18]. However, the largest body of work has been dedicated to the analysis of gradient-based algorithms when training RAEs. Ref. [5] established that minimizing the training loss leads to learning the principal components of the data. Authors of [11, 12] have analyzed how a linear RAE learns these components during training. These studies were later extended to non-linear networks by [19, 8, 9], at the sacrifice of further assuming an infinite number of training samples to be available –either by considering online stochastic gradient descent, or the population loss. Refs. [20, 13] are able to address a finite sample complexity, but in exchange, have to consider infinite-width architectures, which [13] further shows, also tend to a large extent to learn to perform PCA.

*Exact asymptotics from the replica method–* The replica method [21, 22, 23, 24] has proven a very valuable gateway to access sharp asymptotic characterizations of learning metrics for high-dimensional machine learning problems. Past works have addressed –among others– single-[25, 26, 27, 28] and multi-index models [29], or kernel methods [30, 31, 32, 33]. While the approach has traditionally addressed convex problems, for which its prediction can be proven e.g. using the convex Gordon minimax theorem [34], the replica method allows to average over *all* the global minimizers of the loss, and therefore also accommodates non-convex settings. Refs. [35, 36] are two recent examples of its application to non-convex losses. In the present manuscript, we leverage this versatility to study the minimization of the empirical risk of DAEs, whose non-convexity represents a considerable hurdle to many other types of analyses.

## 2   Setting

**Data model**   We consider the problem of denoising data $x \in \mathbb{R}^d$ corrupted by Gaussian white noise of variance $\Delta$,

$$\tilde{x} = \sqrt{1-\Delta}x + \sqrt{\Delta}\xi,$$

where we denoted $\tilde{x}$ the noisy data point, and $\xi \sim \mathcal{N}(0, \mathbb{I}_d)$ the additive noise. The rescaling of the clean data point by a factor $\sqrt{1-\Delta}$ is a practical choice that entails no loss of generality, and allows to easily interpolate between the noiseless case ($\Delta = 0$) and the case where the signal-to-noise ratio vanishes ($\Delta = 1$). Furthermore, it allows us to seamlessly connect with works on diffusion-based generative models, where the rescaling naturally follows from the way the data is corrupted by an Ornstein-Uhlenbeck process [1, 2]. In the present work, we assume the clean data $x$ to be drawn from a Gaussian mixture distribution $\mathbb{P}$ with $K$ clusters

$$x \sim \sum_{k=1}^{K} \rho_k \mathcal{N}(\mu_k, \Sigma_k). \tag{1}$$

The $k-$th cluster is thus centered around $\mu_k \in \mathbb{R}^d$, has covariance $\Sigma_k \succeq 0$, and relative weight $\rho_k$.

**DAE model** An algorithmic way to retrieve the clean data $\boldsymbol{x}$ from the noisy data $\tilde{\boldsymbol{x}}$ is to build a neural network taking the latter as an input and yielding the former as an output. A particularly natural choice for such a network is an autoencoder architecture [3]. The intuition is that the narrow hidden layer of an autoencoder forces the network to learn a succinct latent representation of the data, which is robust against noise corruption of the input. In this work, we analyze a two-layer DAE. We further assume that the weights are tied. Additionally, mirroring modern denoising architectures like U-nets [37] or [38, 39, 40, 41], we also allow for a (trainable) skip-connection:

$$f_{b,\boldsymbol{w}}(\tilde{\boldsymbol{x}}) = b \times \tilde{\boldsymbol{x}} + \frac{\boldsymbol{w}^\top}{\sqrt{d}} \sigma \left( \frac{\boldsymbol{w}\tilde{\boldsymbol{x}}}{\sqrt{d}} \right). \tag{2}$$

The DAE (2) is therefore parametrized by the scalar skip connection strength $b \in \mathbb{R}$ and the weights $\boldsymbol{w} \in \mathbb{R}^{p \times d}$, with $p$ the width of the DAE hidden layer. The normalization of the weight $\boldsymbol{w}$ by $\sqrt{d}$ in (2) is the natural choice which ensures for high dimensional settings $d \gg 1$ that the argument $\boldsymbol{w}\tilde{\boldsymbol{x}}/\sqrt{d}$ of the non-linearity $\sigma(\cdot)$ stays $\Theta(1)$. Like [8], we focus on the case with $p \ll d$. The assumption of weight-tying affords a more concise theoretical characterization and thus clearer discussions. Note that it is also a strategy with substantial practical history, dating back to [3], as it prevents the DAE from functioning in the linear region of its non-linearity $\sigma(\cdot)$. This choice of architecture is also motivated by a particular case of Tweedie's formula [42] (see eq. (79) in Appendix B), which will be the object of further discussion in Section 4.

We also consider two other simple architectures

$$u_{\boldsymbol{v}}(\tilde{\boldsymbol{x}}) = \frac{\boldsymbol{v}^\top}{\sqrt{d}} \sigma \left( \frac{\boldsymbol{v}\tilde{\boldsymbol{x}}}{\sqrt{d}} \right), \qquad\qquad r_c(\tilde{\boldsymbol{x}}) = c \times \tilde{\boldsymbol{x}}, \tag{3}$$

which correspond to the building blocks of the complete DAE architecture $f_{b,\boldsymbol{w}}$ (2) (hereafter referred to as the *full* DAE). Note that indeed $f_{b,\boldsymbol{w}} = r_b + u_{\boldsymbol{w}}$. The part $u_{\boldsymbol{v}}(\cdot)$ is a DAE without skip connection (hereafter called the *bottleneck network* component), while $r_c(\cdot)$ correspond to a simple single-parameter trainable rescaling of the input (hereafter called the *rescaling* component).

To train the DAE (2), we assume the availability of a training set $\mathcal{D} = \{\tilde{\boldsymbol{x}}^\mu, \boldsymbol{x}^\mu\}_{\mu=1}^n$, with $n$ clean samples $\boldsymbol{x}^\mu$ drawn i.i.d from $\mathbb{P}$ (1) and the corresponding noisy samples $\tilde{\boldsymbol{x}}^\mu = \boldsymbol{x}^\mu + \boldsymbol{\xi}^\mu$ (with the noises $\boldsymbol{\xi}^\mu$ assumed mutually independent). The DAE is trained to recover the clean samples $\boldsymbol{x}^\mu$ from the noisy samples $\tilde{\boldsymbol{x}}^\mu$ by minimizing the empirical risk

$$\hat{\mathcal{R}}(b, \boldsymbol{w}) = \sum_{\mu=1}^n \left\| \boldsymbol{x}^\mu - f_{b,\boldsymbol{w}}(\tilde{\boldsymbol{x}}^\mu) \right\|^2 + g(\boldsymbol{w}), \tag{4}$$

where $g : \mathbb{R}^{p \times d} \to \mathbb{R}_+$ is an arbitrary convex regularizing function. We denote by $\hat{b}, \hat{\boldsymbol{w}}$ the minimizers of the empirical risk (4) and by $\hat{f} \equiv f_{\hat{b},\hat{\boldsymbol{w}}}$ the corresponding trained DAE (2). For future discussion, we also consider training independently the components (3) via empirical risk minimization, by which we mean replacing $f_{b,\boldsymbol{w}}$ by $u_{\boldsymbol{v}}$ or $r_c$ in (4). We similarly denote $\hat{\boldsymbol{v}}$ (resp. $\hat{c}$) the learnt weight of the bottleneck network (resp. rescaling) component and $\hat{u} \equiv u_{\hat{\boldsymbol{v}}}$ (resp. $\hat{r} \equiv r_{\hat{c}}$). Note that generically, $\hat{\boldsymbol{v}} \neq \hat{\boldsymbol{w}}$ and $\hat{c} \neq \hat{b}$, and therefore $\hat{f} \neq \hat{u} + \hat{r}$, since $\hat{b}, \hat{\boldsymbol{w}}$ result from their joint optimization as parts of the full DAE $f_{b,\boldsymbol{w}}$, while $\hat{c}$ (or $\hat{\boldsymbol{v}}$) are optimized independently. As we discuss in Section 4, training the sole rescaling $r_c$ does not afford an expressive enough denoiser, while an independently learnt bottleneck network component $u_{\boldsymbol{v}}$ essentially only learns to implement PCA. However, when *jointly* trained as components of the full DAE $f_{b,\boldsymbol{w}}$ (2), the resulting denoiser $\hat{f}$ is a genuinely non-linear model which yields a much lower test error than PCA, and learns to leverage flexibly its two components to balance the preservation of the data nuances and the removal of the noise.

**Learning metrics** The performance of the DAE (2) trained with the loss (4) is quantified by its reconstruction (denoising) test MSE, defined as

$$\mathrm{mse}_{\hat{f}} \equiv \mathbb{E}_{\mathcal{D}} \mathbb{E}_{\boldsymbol{x} \sim \mathbb{P}} \mathbb{E}_{\boldsymbol{\xi} \sim \mathcal{N}(0, \mathbb{I}_d)} \left\| \boldsymbol{x} - f_{\hat{b},\hat{\boldsymbol{w}}} \left( \sqrt{1 - \Delta}\boldsymbol{x} + \sqrt{\Delta}\boldsymbol{\xi} \right) \right\|^2. \tag{5}$$

The expectations run over a fresh test sample $\boldsymbol{x}$ sampled from the Gaussian mixture $\mathbb{P}$ (1), and a new additive noise $\boldsymbol{\xi}$ corrupting it. Note that an expectation over the train set $\mathcal{D}$ is also included to make $\mathrm{mse}_{\hat{f}}$ a metric that does not depend on the particular realization of the train set. The denoising

test MSEs $\mathrm{mse}_{\hat{u}}, \mathrm{mse}_{\hat{r}}$ are defined similarly as the denoising test errors of the independently learnt components (3). Aside from the denoising MSE (5), another question of interest is how much the DAE manages to learn the structure of the data distribution, as described by the cluster means $\boldsymbol{\mu}_k$. This is measured by the cosine similarity matrix $\boldsymbol{\theta} \in \mathbb{R}^{p \times K}$, where for $i \in [\![1, p]\!]$ and $k \in [\![1, K]\!]$,

$$\theta_{ik} \equiv \mathbb{E}_{\mathcal{D}} \left[ \frac{\hat{\boldsymbol{w}}_i^\top \boldsymbol{\mu}_k}{\|\hat{\boldsymbol{w}}_i\| \|\boldsymbol{\mu}_k\|} \right]. \tag{6}$$

In other words, $\theta_{ik}$ measures the alignment of the $i-$th row $\hat{\boldsymbol{w}}_i$ of the trained weight matrix $\hat{\boldsymbol{w}}$ with the mean of the $k-$th cluster $\boldsymbol{\mu}_k$.

**High-dimensional limit**   We analyze the optimization problem (4) in the high-dimensional limit where the input dimension $d$ and number of training samples $n$ jointly tend to infinity, while their ratio $\alpha = {}^n\!/_d$ –hitherto referred to as the *sample complexity*–stays $\Theta(1)$. The hidden layer width $p$, the noise level $\Delta$, the number of clusters $K$ and the norm of the cluster means $\|\boldsymbol{\mu}_k\|$ are also assumed to remain $\Theta(1)$. This corresponds to a rich limit, where the number of parameters of the DAE is not large compared to the number of samples like in [20, 13], and therefore cannot trivially fit the train set, or simply memorize it [17]. Conversely, the number of samples $n$ is not infinite like in [8, 9, 19], and therefore importantly allows to study the effect of a finite train set on the representation learnt by the DAE.

## 3   Asymptotic formulae for DAEs

We now state the main result of the present work, namely the closed-form asymptotic formulae for the learning metrics $\mathrm{mse}_{\hat{f}}$ (5) and $\theta$ (6) for a DAE (2) learnt with the empirical loss (4). These characterizations are obtained by first recasting the optimization problem into an analysis of an associated probability measure, and then carrying out this analysis using the heuristic replica method, which we here employ in its replica-symmetric formulation (see Appendix A).

**Assumption 3.1.** The covariances $\{\boldsymbol{\Sigma}\}_{k=1}^K$ admit a common set of eigenvectors $\{\boldsymbol{e}_i\}_{i=1}^d$. We further note $\{\lambda_i^k\}_{i=1}^d$ the eigenvalues of $\boldsymbol{\Sigma}_k$. The eigenvalues $\{\lambda_i^k\}_{i=1}^d$ and the projection of the cluster means on the eigenvectors $\{\boldsymbol{e}_i^\top \boldsymbol{\mu}_k\}_{i,k}$ are assumed to admit a well-defined joint distribution $\nu$ as $d \to \infty$ – namely, for $\gamma = (\gamma_1, ..., \gamma_K) \in \mathbb{R}^K$ and $\tau = (\tau_1, ..., \tau_K) \in \mathbb{R}^K$:

$$\frac{1}{d} \sum_{i=1}^d \prod_{k=1}^K \delta \left( \lambda_i^k - \gamma_k \right) \delta \left( \sqrt{d} \boldsymbol{e}_i^\top \boldsymbol{\mu}_k - \tau_k \right) \xrightarrow{d \to \infty} \nu \left( \gamma, \tau \right). \tag{7}$$

Moreover, the marginals $\nu_\gamma$ (resp. $\nu_\tau$) are assumed to have a well-defined first (resp. second) moment.

**Assumption 3.2.** $g(\cdot)$ is a $\ell_2$ regularizer with strength $\lambda$, i.e. $g(\cdot) = {}^\lambda\!/_2 \|\cdot\|_F^2$.

We are now in a position to discuss the main result of this manuscript, which we state under Assumptions 3.1 and 3.2 for definiteness and clarity. These assumptions can actually be relaxed, as we further discuss after the result statement.

**Conjecture 3.3.** *(Closed-form asymptotics for DAEs trained with empirical risk minimization) Under Assumptions 3.1 and 3.2, in the high-dimensional limit $n, d \to \infty$ with fixed ratio $\alpha$, the denoising test MSE $\mathrm{mse}_{\hat{f}}$ (5) admits the expression*

$$\mathrm{mse}_{\hat{f}} - \mathrm{mse}_\circ = \sum_{k=1}^K \rho_k \mathbb{E}_z \, \mathrm{Tr} \left[ q \sigma \left( \sqrt{1-\Delta} m_k + \sqrt{\Delta q + (1-\Delta) q_k} z \right)^{\otimes 2} \right] \tag{8}$$

$$- 2 \sum_{k=1}^K \rho_k \mathbb{E}_{u,v} \left[ \sigma \left( \sqrt{1-\Delta} m_k + \sqrt{q_k(1-\Delta)} u + \sqrt{\Delta q} v \right)^\top \left( (1 - \hat{b}\sqrt{1-\Delta})(m_k + \sqrt{q_k} u) - \hat{b}\sqrt{\Delta q} v \right) \right] + o(1),$$

*where the averages bear over independent Gaussian variables $z, u, v \sim \mathcal{N}(0, \mathbb{I}_p)$. We denoted*

$$\mathrm{mse}_\circ = d\Delta \hat{b}^2 + \left( 1 - \sqrt{1-\Delta} \hat{b} \right)^2 \left[ \sum_{k=1}^K \rho_k \left( \int d\nu_\tau(\tau) \tau_k^2 + d \int d\nu_\gamma(\gamma) \gamma_k \right) \right]. \tag{9}$$

*The learnt skip connection strength $\hat{b}$ is*

$$\hat{b} = \frac{\left(\sum\limits_{k=1}^{K} \rho_k \int d\nu_\gamma(\gamma)\gamma_k\right)\sqrt{1-\Delta}}{\left(\sum\limits_{k=1}^{K} \rho_k \int d\nu_\gamma(\gamma)\gamma_k\right)(1-\Delta) + \Delta} + o(1). \tag{10}$$

*The cosine similarity $\theta$ (6) admits the compact formula for $i \in [\![1,p]\!]$ and $k \in [\![1,K]\!]$*

$$\theta_{ik} = \frac{(m_k)_i}{\sqrt{q_{ii} \int d\nu_\tau(\tau)\tau_k^2}}, \tag{11}$$

*where we have introduced the summary statistics*

$$q = \lim_{d\to\infty} \mathbb{E}_{\mathcal{D}}\left[\frac{\hat{\boldsymbol{w}}\hat{\boldsymbol{w}}^\top}{d}\right], \qquad q_k = \lim_{d\to\infty} \mathbb{E}_{\mathcal{D}}\left[\frac{\hat{\boldsymbol{w}}\boldsymbol{\Sigma}_k\hat{\boldsymbol{w}}^\top}{d}\right], \qquad m_k = \lim_{d\to\infty} \mathbb{E}_{\mathcal{D}}\left[\frac{\hat{\boldsymbol{w}}\boldsymbol{\mu}_k}{\sqrt{d}}\right]. \tag{12}$$

*Thus $q, q_k \in \mathbb{R}^{p\times p}$, $m_k \in \mathbb{R}^p$. The existence of these limits is an assumption of the replica method. The summary statistics $q, q_k, m_k$ can be determined as solutions of the system of equations*

$$\begin{cases}
\hat{q}_k = \alpha\rho_k \mathbb{E}_{\xi,\eta} V_k^{-1}\left(\text{prox}_y^k - q_k^{\frac{1}{2}}\eta - m_k\right)^{\otimes 2} V_k^{-1} \\
\hat{V}_k = -\alpha\rho_k q_k^{-\frac{1}{2}} \mathbb{E}_{\xi,\eta} V_k^{-1}\left(\text{prox}_y^k - q_k^{\frac{1}{2}}\eta - m_k\right)\eta^\top \\
\hat{m}_k = \alpha\rho_k \mathbb{E}_{\xi,\eta} V_k^{-1}\left(\text{prox}_y^k - q_k^{\frac{1}{2}}\eta - m_k\right) \\
\hat{q} = \frac{\alpha}{\Delta} \sum\limits_{k=1}^{K} \rho_k \mathbb{E}_{\xi,\eta} V^{-1}\left(\text{prox}_x^k - \sqrt{\Delta}q^{\frac{1}{2}}\xi\right)^{\otimes 2} V^{-1} \\
\hat{V} = -\alpha \sum\limits_{k=1}^{K} \rho_k \mathbb{E}_{\xi,\eta}\left[\frac{1}{\sqrt{\Delta}}q^{-\frac{1}{2}}V^{-1}\left(\text{prox}_x^k - \sqrt{\Delta}q^{\frac{1}{2}}\xi\right)\xi^\top - \sigma\left(\sqrt{1-\Delta}\text{prox}_y^k + \text{prox}_x^k\right)^{\otimes 2}\right]
\end{cases}$$

$$\begin{cases}
q_k = \int d\nu(\gamma,\tau)\gamma_k \left(\lambda\mathbb{I}_p + \hat{V} + \sum\limits_{j=1}^{K}\gamma_j\hat{V}_j\right)^{-2}\left(\hat{q} + \sum\limits_{j=1}^{K}\gamma_j\hat{q}_j + \sum\limits_{1\leq j,l\leq K}\tau_j\tau_l\hat{m}_j\hat{m}_l^\top\right) \\
V_k = \int d\nu(\gamma,\tau)\gamma_k \left(\lambda\mathbb{I}_p + \hat{V} + \sum\limits_{j=1}^{K}\gamma_j\hat{V}_j\right)^{-1} \\
m_k = \int d\nu(\gamma,\tau)\tau_k \left(\lambda\mathbb{I}_p + \hat{V} + \sum\limits_{j=1}^{K}\gamma_j\hat{V}_j\right)^{-1}\sum\limits_{j=1}^{K}\tau_j\hat{m}_j \\
q = \int d\nu(\gamma,\tau)\left(\lambda\mathbb{I}_p + \hat{V} + \sum\limits_{j=1}^{K}\gamma_j\hat{V}_j\right)^{-2}\left(\hat{q} + \sum\limits_{j=1}^{K}\gamma_j\hat{q}_j + \sum\limits_{1\leq j,l\leq K}\tau_j\tau_l\hat{m}_j\hat{m}_l^\top\right) \\
V = \int d\nu(\gamma,\tau)\left(\lambda\mathbb{I}_p + \hat{V} + \sum\limits_{j=1}^{K}\gamma_j\hat{V}_j\right)^{-1}
\end{cases} \tag{13}$$

*In (13), $\hat{q}_k, \hat{V}_k, \hat{V}, V \in \mathbb{R}^{p\times p}$ and $\hat{m}_k \in \mathbb{R}^p$, and the averages bear over finite-dimensional i.i.d Gaussians $\xi, \eta \sim \mathcal{N}(0, \mathbb{I}_p)$. Finally, $\text{prox}_x^k, \text{prox}_y^k$ are given as the solutions of the finite-dimensional optimization*

$$\text{prox}_x^k, \text{prox}_y^k = \underset{x,y\in\mathbb{R}^p}{\arg\inf}\left\{\text{Tr}\left[V_k^{-1}\left(y - q_k^{\frac{1}{2}}\eta - m_k\right)^{\otimes 2}\right] + \frac{1}{\Delta}\text{Tr}\left[V^{-1}\left(x - \sqrt{\Delta}q^{\frac{1}{2}}\xi\right)^{\otimes 2}\right]\right.$$

$$\left. + \text{Tr}\left[q\sigma(\sqrt{1-\Delta}y + x)^{\otimes 2}\right] - 2\sigma(\sqrt{1-\Delta}y + x)^\top((1 - \sqrt{1-\Delta}\hat{b})y - \hat{b}x)\right\}. \tag{14}$$

Conjecture 3.3 provides a gateway to probe and characterize the asymptotic properties of the model (2) at the global optimum of the empirical risk (4), whereas a purely experimental study would not have been guaranteed to reach the global solution, and would suffer from finite-size effects. Equation

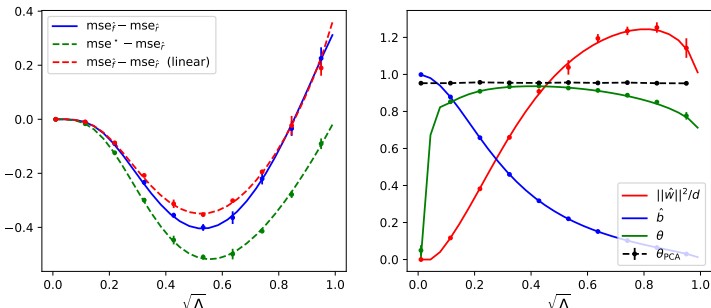

Figure 1: $\alpha = 1, K = 2, \rho_{1,2} = {}^1\!/2, \boldsymbol{\Sigma}_{1,2} = 0.09 \times \mathbb{I}_d, p = 1, \lambda = 0.1, \sigma(\cdot) = \tanh(\cdot)$; the cluster mean $\boldsymbol{\mu}_1 = -\boldsymbol{\mu}_2$ was taken as a random Gaussian vector of norm 1. (left) In blue, the difference in MSE between the full DAE $\hat{f}$ (2) and the rescaling component $\hat{r}$ (3). Solid lines correspond to the sharp asymptotic characterization of Conjecture 3.3. Dots represent numerical simulations for $d = 700$, training the DAE using the Pytorch implementation of full-batch Adam, with learning rate $\eta = 0.05$ over 2000 epochs, averaged over $N = 10$ instances. Error bars represent one standard deviation. For completeness, the MSE of the oracle denoiser is given as a baseline in green, see Section 4. The performance of a linear DAE ($\sigma(x) = x$) is represented in dashed red. (right) Cosine similarity $\theta$ (6) (green), squared weight norm $\|\hat{w}\|_F^2/d$ (red) and skip connection strength $\hat{b}$ (blue). Solid lines correspond to the formulae (11)(12) and (10) of Conjecture 3.3; dots are numerical simulations. For completeness, the cosine similarity of the first principal component of the clean train data $\{\boldsymbol{x}^\mu\}_{\mu=1}^n$ is plotted in dashed black.

(13) provides a closed-form expression for the summary statistics (12), in terms of the solutions of the low-dimensional optimization (14). The latter can be loosely viewed as an effective loss, subsuming the averaged effect of the finite training set. Further remark that while the regularization $\lambda$ does not explicitly appear in the expression for the MSE (8), the statistics $q_k, m_k$ in (8) depend thereon through (13). In fact, Assumptions 3.1 and 3.2 are not strictly necessary, and can be simultaneously relaxed to address arbitrary convex regularizer $g(\cdot)$ and generically non-commuting $\{\boldsymbol{\Sigma}_k\}_{k=1}^K$ – but at the price of more intricate formulae. For this reason, we choose to discuss here Conjecture 3.3, and defer a discussion and detailed derivation of the generic asymptotically exact formulae to Appendix A, see eq. (58). Let us mention that a sharp asymptotic characterization of the *train* MSE can also be derived; for conciseness, we do not present it here and refer the interested reader to equation (68) in Appendix A. Conjecture 3.3 encompasses as special cases the asymptotic characterization of the components $\hat{r}, \hat{u}$ (3):

**Corollary 3.4.** *(MSE of components) The test MSE of $\hat{r}$ (3) is given by* $\mathrm{mse}_{\hat{r}} = \mathrm{mse}_\circ$ *(9). Furthermore, the learnt value of its single parameter $\hat{c}$ is given by (10). The test MSE, cosine similarity and summary statistics of the bottleneck network $\hat{u}$ (3) follow from Conjecture 3.3 by setting $\hat{b} = 0$.*

The implications of Corollary 3.4 shall be further discussed in Section 4, and a full derivation is provided in Appendix E. Finally, remark that in the noiseless limit $\Delta = 0$, the denoising task reduces to a reconstruction task, with the autoencoder being tasked with reproducing the clean data as an output when taking the same clean sample as an input. Therefore Conjecture 3.3 also includes RAEs (by definition, without skip connection) as a special case.

**Corollary 3.5.** *(RAEs) In the $n, d \to \infty$ limit, the MSE, cosine similarity and summary statistics for an RAE follow from Conjecture 3.3 by setting $x = 0$ in (14), removing the first term in the brackets in the equation of $\hat{V}$ (13) and taking the limit $\Delta, \hat{q}, \hat{b} \to 0$.*

Corollary 3.5 will be the object of further discussion in Section 4. A detailed derivation is presented in Appendix F. Note that Corollary 3.5 provides a characterization of RAEs as a function of the sample complexity $\alpha$, where previous studies on non-linear RAEs rely on the assumption of an infinite number of available training samples [13, 8, 9].

Equations (10) and (12) of Conjecture 3.3 thus characterize the statistics of the learnt parameters $\hat{b}, \hat{\boldsymbol{w}}$ of the trained DAE (2). These summary statistics are, in turn, sufficient to fully characterize the

learning metrics (5) and (6) via equations (8) and (11). We thus have reduced the high-dimensional optimization (4) and the high-dimensional average over the train set $\mathcal{D}$ involved in the definition of the metrics (5) and (6) to a simpler system of equations over $4 + 6K$ variables (13) which can be solved numerically. It is important to note that all the summary statistics involved in (13) are *finite-dimensional* as $d \to \infty$, and therefore Conjecture 3.3 is a fully asymptotic characterization, in the sense that it does not involve any high-dimensional object. Finally, let us stress once more that the replica method employed in the derivation of these results should be viewed as a strong heuristic, but does not constitute a rigorous proof. While Conjecture 3.3 is stated in full generality, we focus for definiteness in the rest of the manuscript on the simple case $r = 1, K = 2$, which is found to already display all the interesting phenomenology discussed in this work, and leave the thorough exploration of $r > 1, K > 2$ settings to future work. In the next paragraphs, we give two examples of applications of Conjecture 3.3, to a simple binary isotropic mixture, and to real data sets.

**Example 1: Isotropic homoscedastic mixture** We give as a first example the case of a synthetic binary Gaussian mixture with $K = 2, \boldsymbol{\mu}_1 = -\boldsymbol{\mu}_2, \boldsymbol{\Sigma}_{1,2} = 0.09 \times \mathbb{I}_d, \rho_{1,2} = 1/2$, using a DAE with $\sigma = \tanh$ and $p = 1$. Since this simple case exhibits the key phenomenology discussed in the present work, we refer to it in future discussions. The MSE $\mathrm{mse}_{\hat{f}}$ (8) evaluated from the solutions of the self-consistent equations (13) is plotted as the solid blue line in Fig. 1 (left) and compared to numerical simulations corresponding to training the DAE (2) with the `Pytorch` implementation of the Adam optimizer [43] (blue dots), for sample complexity $\alpha = 1$ and $\ell_2$ regularization (weight decay) $\lambda = 0.1$. The agreement between the theory and simulation is compelling. The green solid line and corresponding green dots in Fig. 1 (right) correspond to the replica prediction (11) and simulations for the cosine similarity $\theta$ (6), and again display very good agreement. Note that for large noise levels, the DAE achieves a worse MSE than the rescaling –as shown by the positive value of $\mathrm{mse}_{\hat{f}} - \mathrm{mse}_{\hat{r}}$–, despite the former being a priori expressive enough to realize the latter. This in fact signals that the DAE overfits the training data. That such an overfitting is captured is a strength of our analysis, which allows to cover the effect of a limited sample complexity. Finally, this overfitting can be mitigated by increasing the weight decay $\lambda$, see Fig. 9 in Appendix A.

A particularly striking observation is that due to the non-convexity of the loss (4), there is a priori no guarantee that an Adam-optimized DAE should find a global minimum, as described by the Conjecture 3.3, rather than a local minimum. The compelling agreement between theory and simulations in Fig. 1 temptingly suggests that –at least in this case– the loss landscape of DAEs (2) trained with the loss (4) for the data model (1) should in some way be benign. Authors of [12] have shown, for *linear RAEs*, that there exists a unique global and local minimum for the square loss and no regularizer. Ref. [14] offers further insight for a linear DAE in dimension $d = 1$, and shows that, aside from the global minima, the loss landscape only includes an unstable saddle point from which the dynamics easily escapes. Extending these works and intuitions to non-linear DAEs is an exciting research topic for future work.

**Example 2: MNIST, FashionMNIST** It is reasonable to ask whether Conjecture 3.3 is restricted to Gaussian mixtures (1). The answer is negative – in fact, Conjecture 3.3 also describes well a number of real data distributions. We provide such an example for FashionMNIST [16] (from which, for simplicity, we only kept boots and shoes) and MNIST [15] (1s and 7s), in Fig. 2. For each data set, samples sharing the same label were considered to belong to the same cluster. Note that we purposefully chose closely related classes, for which the clusters are expected to be closer, leading to an *a priori* harder – and thus more interesting – learning problem. The mean and covariance thereof were estimated numerically, and combined with Conjecture 3.3. The resulting denoising MSE predictions $\mathrm{mse}_{\hat{f}}$ are plotted as solid lines in Fig. 2, and agree very well with numerical simulations of DAEs optimized over the real data sets using the `Pytorch` implementation of Adam [43]. A full description of this experiment is given in Appendix D.

The observation that the MSEs of real data sets are to such degree of accuracy captured by the equivalent Gaussian mixture strongly hints at the presence of Gaussian universality [44]. This opens a gateway to future research, as Gaussian universality has hitherto been exclusively addressed in classification and regression (rather than denoising) settings, see e.g. [44, 45, 46]. Denoising tasks further constitute a particularly intriguing setting for universality results, as Gaussian universality would signify that only second-order statistics of the data can be reconstructed using a shallow autoencoder.

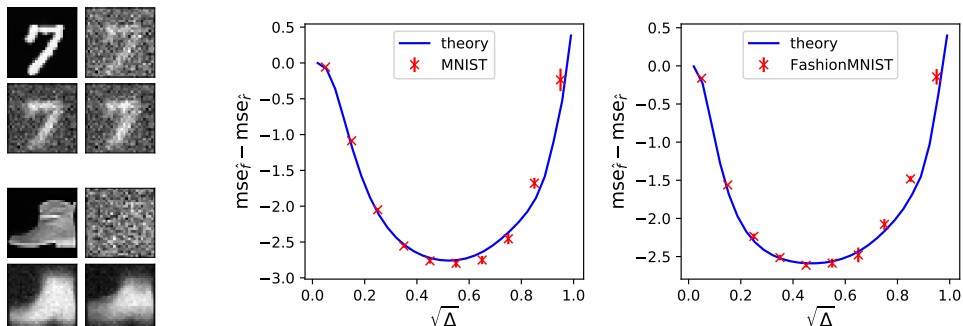

Figure 2: Difference in MSE between the full DAE (2) and the rescaling component (3) for the MNIST data set (middle), of which for simplicity only 1s and 7s were kept, and FashionMNIST (right), of which only boots and shoes were kept. In blue, the theoretical predictions resulting from using Conjecture 3.3 with the empirically estimated covariances and means, see Appendix D for further details. In red, numerical simulations of a DAE ($p = 1$, $\sigma = \tanh$) trained with $n = 784$ training points, using the `Pytorch` implementation of full-batch Adam, with learning rate $\eta = 0.05$ and weight decay $\lambda = 0.1$ over 2000 epochs, averaged over $N = 10$ instances. Error bars represent one standard deviation. (left) illustration of the denoised images: (top left) original image, (top right) noisy image, (bottom left) DAE $\hat{f}$ (2), (bottom right) rescaling $\hat{r}$ (3).

## 4 The role and importance of the skip connection.

Conjecture 3.3 for the full DAE $\hat{f}$ (2) and Corollary 3.4 for its components $\hat{r}$, $\hat{u}$ (3) allow to disentangle the contribution of each part, and thus to pinpoint their respective roles in the DAE architecture. We sequentially present a comparison of $\hat{f}$ with $\hat{r}$, and $\hat{f}$ with $\hat{u}$. We remind that $\hat{f}$, $\hat{r}$ and $\hat{u}$ result from *independent* optimizations over the same train set $\mathcal{D}$, and that while $f_{b,\boldsymbol{w}} = u_{\boldsymbol{w}} + h_b$, $\hat{f} \neq \hat{u} + \hat{r}$.

**Full DAE and the rescaling component**   We start this section by observing that for noise levels $\Delta$ below a certain threshold, the full DAE $\hat{f}$ yields better MSE than the learnt rescaling $\hat{r}$, as can be seen by the negative value of $\mathrm{mse}_{\hat{f}} - \mathrm{mse}_{\hat{r}}$ in Fig. 1 and Fig. 2. The improvement is more sizeable at intermediate noise levels $\Delta$, and is observed for a growing region of $\Delta$ as the sample complexity $\alpha$ increases, see Fig. 3 (a). This lower MSE further translates into visible qualitative changes in the result of denoising. As can be seen from Fig. 2 (left), the full DAE $\hat{f}$ (2) (bottom left) yields denoised images with sensibly higher definition and overall contrast, while a simple rescaling $\hat{r}$ (bottom right) leads to a still largely blurred image.

We provide one more comparison: for the isotropic binary mixture (see Fig. 1), the DAE test error $\mathrm{mse}_{\hat{f}}$ in fact approaches the information-theoretic lowest achievable MSE $\mathrm{mse}^\star$ as the sample complexity $\alpha$ increases. To see this, note that $\mathrm{mse}^\star$ is given by the application of Tweedie's formula [42], that requires perfect knowledge of the cluster means $\boldsymbol{\mu}_k$ and covariances $\boldsymbol{\Sigma}_k$ – it is, therefore, an *oracle* denoiser. A sharp asymptotic characterization of the oracle denoiser is provided in Appendix B. As can be observed from Fig. 3 (a), the DAE MSE (2) approaches –but does not exactly converges to (see Appendix C)– the oracle test error $\mathrm{mse}^\star$ as the number of available training samples $n$ grows, and is already sensibly close to the optimal value for $\alpha = 8$.

**DAEs with(out) skip connection**   We now turn our attention to comparing the full DAE $\hat{f}$ (2) to the bottleneck network component $\hat{u}$ (3). It follows from Conjecture 3.3 and Corollary 3.4 that $\hat{u}$ (3) leads to a higher MSE than the full DAE $\hat{f}$ (2), with the gap being $\Theta(d)$. More precisely,

$$\frac{1}{d}\left(\mathrm{mse}_{\hat{u}} - \mathrm{mse}_{\hat{f}}\right) = \frac{\left(\int d\nu_\gamma(\gamma) \sum_{k=1}^{K} \rho_k \gamma_k\right)^2 (1 - \Delta)}{\left(\int d\nu_\gamma(\gamma) \sum_{k=1}^{K} \rho_k \gamma_k\right)(1 - \Delta) + \Delta}. \tag{15}$$

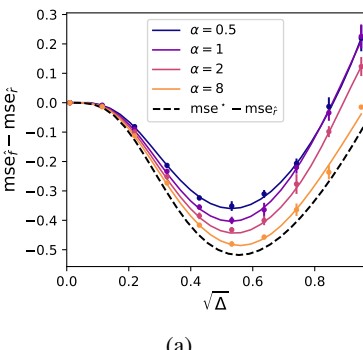
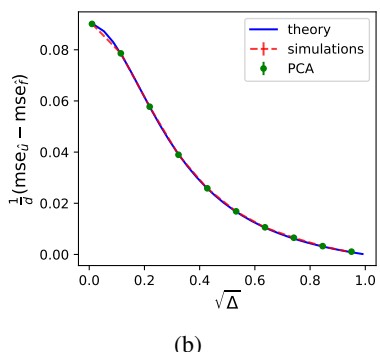

(a)            (b)

Figure 3: (left) Solid lines: difference in MSE between the full DAE $\hat{f}$ (2), with $\sigma = \tanh$, $p = 1$, and the rescaling $\hat{r}$ (3). Dashed: the same curve for the oracle denoiser. Different colours represent different sample complexities $\alpha$ (solid lines). (right) Difference in MSE between the bottleneck network $\hat{u}$ (3) and the complete DAE $\hat{f}$ (2). In blue, the theoretical prediction (15); in red, numerical simulations for the bottleneck network (3) ($\sigma = \tanh$, $p = 1$) trained with the Pytorch implementation of full-batch Adam, with learning rate $\eta = 0.05$ and weight decay $\lambda = 0.1$ over 2000 epochs, averaged over $N = 5$ instances, for $d = 700$. In green, the MSE (minus the MSE of the complete DAE (2)) achieved by PCA. Error bars represent one standard deviation. The model and parameters are the same as in Fig. 1.

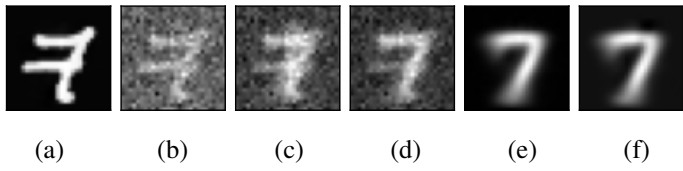

(a)       (b)       (c)       (d)       (e)       (f)

Figure 4: Illustration of the denoised image for the various networks and algorithms. (a) original image (b) noisy image, for $\sqrt{\Delta} = 0.2$ (c) trained rescaling $\hat{r}$ (3) (d) full DAE $\hat{f}$ (2) (e) bottleneck network $\hat{u}$ (3) (f) PCA. The DAE and training parameters are the same as Fig. 2, see also Appendix D.

The theoretical prediction (15) compares excellently with numerical simulations; see Fig. 3 (right). Strikingly, we find that PCA denoising yields an MSE almost indistinguishable from $\hat{u}$, see Fig. 3, strongly suggesting that $\hat{u}$ essentially learns, also in the denoising setting, to project the noisy data $\tilde{x}$ along the principal components of the training set. The last two images of Fig. 4 respectively correspond to $\hat{u}$ and PCA, which can indeed be observed to lead to visually near-identical results. This echoes the findings of [10, 11, 8, 9, 13] in the case of RAEs that bottleneck networks are limited by the PCA reconstruction performance – a conclusion that we also recover from Corollary 3.5, see Appendix F. Crucially however, it *also* means that compared to the *full* DAE $\hat{f}$ (2), *PCA is sizeably suboptimal*, since $\text{mse}_{\text{PCA}} \approx \text{mse}_{\hat{u}} = \text{mse}_{\hat{f}} + \Theta(d)$.

This last observation has an important consequence: in contrast to previously studied RAEs [10, 12, 11, 9, 13], the full DAE $\hat{f}$ does *not* simply learn to perform PCA. In contrast to bottleneck RAE networks [8, 9, 13], the non-linear DAE hence does not reduce to a linear model after training. The non-linearity is important to improve the denoising MSE, see Fig. 1. We stress this finding: trained alone, the bottleneck network $\hat{u}$ only learns to perform PCA; trained jointly with the rescaling component as part of the full DAE $f_{b,w}$ (2), it learns a richer, non-linear representation. The full DAE (2) thus offers a genuinely non-linear learning model and opens exciting research avenues for the theory of autoencoders, beyond linear (or effectively linear) cases. In the next paragraph, we explore further the interaction between the rescaling component and the bottleneck network.

**A tradeoff between the rescaling and the bottleneck network** Conjecture (3.3), alongside Corollary (3.4) and the discussion in Section 4 provide a firm theoretical basis for the well-known empirical intuition (discussed e.g. in [38]) that skip-connections allow to better propagate information

from the input to the output of the DAE, thereby contributing to preserving intrinsic characteristics of the input. This effect is clearly illustrated in Fig. 4, where the resulting denoised image of an MNIST 7 by $\hat{r}$, $\hat{f}$, $\hat{u}$ and PCA are presented. While the bottleneck network $\hat{u}$ perfectly eliminates the background noise and produces an image with a very good resolution, it essentially collapses the image to the cluster mean, and yields, like PCA, the average MNIST 7. As a consequence, the denoised image bears little resemblance with the original image – in particular, the horizontal bar of the 7 is lost in the process. Conversely, the rescaling $\hat{r}$ preserves the nuances of the original image, but the result is still largely blurred and displays overall poor contrast. Finally, the complete DAE (2) manages to preserve the characteristic features of the original data, while enhancing the image resolution by slightly overlaying the average 7 thereupon.

The optimization of the DAE (4) is therefore described by a tradeoff between two competing effects – namely the preservation of the input nuances by the skip connection, and the enhancement of the resolution/noise removal by the bottleneck network. This allows us to discuss the curious non-monotonicity of the cosine similarity $\theta$ as a function of the noise level $\Delta$, see Fig. 1 (left). While it may at first seem curious that the DAE seemingly does not manage to learn the data structure better for low $\Delta$ than for intermediate $\Delta$ (where the cosine similarity $\theta$ is observed to be higher), this is actually due to the afore-dicussed tradeoff. Indeed, for small $\Delta$, the data is still substantially clean, and there is therefore no incentive to enhance the contrast by using the cluster means –which are consequently not learnt. This phase is thus characterized by a large skip connection strength $\hat{b}$, and small cosine similarity $\theta$ and weight norm $\|\hat{\boldsymbol{w}}\|_F$. Conversely, at high noise levels $\Delta$, the nuances of the data are already lost because of the noise. Hence the DAE does not rely on the skip connection component (whence the small values of $\hat{b}$), and the only way to produce reasonably denoised data is to collapse to the cluster mean using the network component (whence a large $\|\hat{\boldsymbol{w}}\|_F$).

## 5    Conclusion

We consider the problem of denoising a high-dimensional Gaussian mixture, by training a DAE via empirical risk minimization, in the limit where the number of training samples and the dimension are proportionally large. We provide a sharp asymptotic characterization of a number of summary statistics of the trained DAE weight, average MSE, and cosine similarity with the cluster means. These results contain as a corollary the case of RAEs. Building on these findings, we isolate the role of the skip connection and the bottleneck network in the DAE architecture and characterize the tradeoff between those two components in terms of preservation of the data nuances and noise removal – thereby providing some theoretical insight into a longstanding practical intuition in machine learning.

We believe the present work also opens exciting research avenues. First, our real data experiments hint at the presence of Gaussian universality. While this topic has gathered considerable attention in recent years, only classification/regression supervised learning tasks have been hitherto addressed. Which aspects of universality carry over to denoising tasks, and how they differ from the current understanding of supervised regression/classification, is an important question. Second, the DAE with skip connection (2) provides an autoencoder model which does not just simply learn the principal components of the training set. It, therefore, affords a genuinely non-linear network model where richer learning settings can be investigated.

### Acknowledgements

We thank Eric Vanden-Eijnden for his wonderful lecture at the les Houches Summer School of July 2022, which inspired parts of this work. We thank Maria Refinetti and Sebastian Goldt for insightful discussions at the very early stages of this project.

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

## A Derivation of Conjecture 3.3

In this section, we detail the derivation of Conjecture (3.3).

### A.1 Derivation technique

In order to access sharp asymptotic characterizations for the metrics and summary statistics mse (5),$\theta$ (6), $\hat{b}$ and $\|\hat{w}\|/d$, the first step is to observe that any test function $\varphi(\hat{w}, \hat{b})$ of the learnt network parameters $\hat{w}, \hat{b}$ can be written as an average over a limit probability distribution

$$\mathbb{E}_{\mathcal{D}}\left[\varphi(\hat{w}, \hat{b})\right] = \lim_{\beta \to \infty} \mathbb{E}_{\mathcal{D}} \frac{1}{Z} \int dw db \varphi(w, b) e^{-\beta \hat{\mathcal{R}}(w, b)} \tag{16}$$

where $\hat{\mathcal{R}}(\cdot, \cdot)$ is the empirical risk (4) and the corresponding probability density

$$\mathbb{P}_{\beta}(w, b) = \frac{e^{-\beta \hat{\mathcal{R}}(w, b)}}{Z} \tag{17}$$

is known as the *Boltzmann measure* in statistical physics, with the parameter $\beta$ typically referred to as the *inverse temperature*. The normalization $Z$ is called the partition function of $\mathbb{P}_{\beta}$:

$$Z = \int dw db e^{-\beta \hat{\mathcal{R}}(w, b)}. \tag{18}$$

The idea of the replica method [21, 22, 23, 24], building on the expression (16), is to compute the moment generating function (also known as the *free entropy*) as

$$\mathbb{E}_{\mathcal{D}} \ln Z = \lim_{s \to 0} \frac{\ln \left[\mathbb{E}_{\mathcal{D}} Z^s\right]}{s}. \tag{19}$$

With the moment generating function, it then becomes possible to compute averages like (16). The backbone of the derivation, which we detail in the following subsections, therefore lies in the computation of the replicated partition function $\mathbb{E}_{\mathcal{D}} Z^s$.

### A.2 Replicated partition function

For subsequent clarity, we shall introduce the variable

$$\eta^{\mu} \equiv x^{\mu} - \mu_k \tag{20}$$

if $x^{\mu}$ belongs to the $k-$th cluster. By definition of the Gaussian mixture model, this is just a Gaussian variable: $\eta^{\mu} \sim \mathcal{N}(0, \Sigma_k)$, and the average over the data set $\{x^{\mu}\}$ reduces to averaging over the cluster index $k$ and $\{\eta^{\mu}\}$. For notational brevity, we shall name in this appendix $\xi^{\mu}$ what is called $\sqrt{\Delta}\xi^{\mu}$ in the main text (i.e., absorbing the variance in the definition of $\xi^{\mu}$). The replicated partition function then reads

$$\mathbb{E}_{\mathcal{D}} Z^s = \int \prod_{a=1}^{s} dw_a db_a e^{-\beta \sum_{a=1}^{s} g(w^a)}$$

$$\times \prod_{\mu=1}^{n} \sum_{k=1}^{K} \rho_k \mathbb{E}_{\eta, \xi} \underbrace{e^{-\frac{\beta}{2} \sum_{a=1}^{s} \left\| \mu_k + \eta - \left[ b_a \times (\sqrt{1-\Delta}(\mu_k + \eta) + \xi) + \frac{w_a^{\top}}{\sqrt{d}} \sigma\left(\frac{w_a(\sqrt{1-\Delta}(\mu_k + \eta) + \xi)}{\sqrt{d}}\right) \right] \right\|^2}}_{(\star)}. \tag{21}$$

One can expand the exponent $(\star)$ as

$$e^{-\frac{\beta}{2} \sum_{a=1}^{s} \left[ \text{Tr}\left[ \frac{w_a w_a^{\top}}{d} \sigma\left( \frac{w_a(\sqrt{1-\Delta}(\mu_k + \eta) + \xi)}{\sqrt{d}} \right)^{\otimes 2} \right] - 2\sigma\left( \frac{w_a(\sqrt{1-\Delta}(\mu_k + \eta) + \xi)}{\sqrt{d}} \right)^{\top} \frac{w_a((1 - b_a\sqrt{1-\Delta})(\mu_k + \eta) - b_a\xi)}{\sqrt{d}} \right]}$$

$$\times e^{-\frac{\beta}{2} \sum_{a=1}^{s} \left[ (1 - \sqrt{1-\Delta} b_a)^2 (\|\mu_k\|^2 + \|\eta\|^2 + 2\mu_k^{\top}\eta) + b_a^2 \|\xi\|^2 + 2b_a(\sqrt{1-\Delta} b_a - 1)(\mu_k^{\top}\xi + \eta^{\top}\xi) \right]}. \tag{22}$$

Therefore

$$\mathbb{E}_{s,\boldsymbol{\eta},\boldsymbol{\xi}}(\star) = \sum_{k=1}^{K} \rho_k e^{-\frac{\beta}{2}\|\boldsymbol{\mu}_k\|^2 \sum_{a=1}^{s} (\sqrt{1-\Delta}b_a-1)^2} \int \frac{d\boldsymbol{\eta}d\boldsymbol{\xi}}{(2\pi\sqrt{\Delta})^d \sqrt{\det \boldsymbol{\Sigma}_k}}$$

$$\underbrace{e^{-\frac{1}{2}\boldsymbol{\eta}^\top \left(\boldsymbol{\Sigma}_k^{-1} + \beta \sum_{a=1}^{s} (\sqrt{1-\Delta}b_a-1)^2 \mathbb{I}_d\right)\boldsymbol{\eta} - \frac{1}{2}\|\boldsymbol{\xi}\|^2 \left(\frac{1}{\Delta} + \beta \sum_{a=1}^{s} b_a^2\right) - \beta \sum_{a=1}^{s} \left((1-\sqrt{1-\Delta}b_a)^2 \boldsymbol{\mu}_k^\top \boldsymbol{\eta} + b_a(\sqrt{1-\Delta}b_a-1)\boldsymbol{\mu}_k^\top \boldsymbol{\xi} + b_a(\sqrt{1-\Delta}b_a-1)\boldsymbol{\eta}^\top \boldsymbol{\xi}\right)}}_{P_{\text{eff.}}(\boldsymbol{\eta},\boldsymbol{\xi})}$$

$$\times e^{-\frac{\beta}{2} \sum_{a=1}^{s} \left[\text{Tr}\left[\frac{\boldsymbol{w}_a \boldsymbol{w}_a^\top}{d} \sigma\left(\frac{\boldsymbol{w}_a(\sqrt{1-\Delta}(\boldsymbol{\mu}_k+\boldsymbol{\eta})+\boldsymbol{\xi})}{\sqrt{d}}\right)^{\otimes 2}\right] - 2\sigma\left(\frac{\boldsymbol{w}_a(\sqrt{1-\Delta}(\boldsymbol{\mu}_k+\boldsymbol{\eta})+\boldsymbol{\xi})}{\sqrt{d}}\right)^\top \frac{\boldsymbol{w}_a((1-b_a\sqrt{1-\Delta})(\boldsymbol{\mu}_k+\boldsymbol{\eta})-b_a\boldsymbol{\xi})}{\sqrt{d}}\right]}. \quad (23)$$

The effective prior over the noises $\boldsymbol{\eta}, \boldsymbol{\xi}$ is Gaussian with means $\boldsymbol{\mu}_\xi, \boldsymbol{\mu}_\eta$ and covariance $C$:

$$P_{\text{eff.}}(\boldsymbol{\eta},\boldsymbol{\xi}) = \mathcal{N}\left(\boldsymbol{\eta},\boldsymbol{\xi};\boldsymbol{\mu}_\eta,\boldsymbol{\mu}_\xi, \begin{bmatrix} \boldsymbol{C}_{\eta\eta}^{-1} & C_{\xi\eta}^{-1}\mathbb{I}_d \\ C_{\xi\eta}^{-1}\mathbb{I}_d & C_{\xi\xi}^{-1}\mathbb{I}_d \end{bmatrix}^{-1}\right). \quad (24)$$

Identifying the covariance components leads to

$$C_{\xi\xi}^{-1} = \frac{1}{\Delta} + \beta \sum_{a=1}^{s} b_a^2, \quad \boldsymbol{C}_{\eta\eta}^{-1} = \boldsymbol{\Sigma}_k^{-1} + \beta \sum_{a=1}^{s} (\sqrt{1-\Delta}b_a - 1)^2 \mathbb{I}_d, \quad C_{\xi\eta}^{-1} = \beta \sum_{a=1}^{s} b_a(\sqrt{1-\Delta}b_a - 1). \quad (25)$$

Note that all the sums indexed by $a$ introduce terms of $\mathcal{O}(s)$, which can safely be neglected. Identifying the means leads to the following system of equations:

$$\begin{cases} C_{\xi\xi}^{-1}\boldsymbol{\mu}_\xi + C_{\xi\eta}^{-1}\boldsymbol{\mu}_\eta = \beta \sum_{a=1}^{s} b_a(1 - \sqrt{1-\Delta}b_a)\boldsymbol{\mu}_k \\ \boldsymbol{C}_{\eta\eta}^{-1}\boldsymbol{\mu}_\eta + C_{\xi\eta}^{-1}\boldsymbol{\mu}_\xi = -\beta \sum_{a=1}^{s} (1 - \sqrt{1-\Delta}b_a)^2 \boldsymbol{\mu}_k \end{cases}, \quad (26)$$

which is solved as

$$\boldsymbol{\mu}_\xi = \underbrace{\beta\left(\boldsymbol{C}_{\eta\eta}^{-1}C_{\xi\xi}^{-1} - (C_{\xi\eta}^{-1})^2\mathbb{I}_d\right)\left(\sum_{a=1}^{s}\left(\boldsymbol{C}_{\eta\eta}^{-1}b_a(1-\sqrt{1-\Delta}b_a) + C_{\xi\eta}^{-1}(1-\sqrt{1-\Delta}b_a)^2\mathbb{I}_d\right)\right)}_{\equiv g_\xi^\mu}\boldsymbol{\mu}_k,$$

$$\boldsymbol{\mu}_\eta = \underbrace{-\beta\left(\boldsymbol{C}_{\eta\eta}^{-1}C_{\xi\xi}^{-1} - (C_{\xi\eta}^{-1})^2\mathbb{I}_d\right)\left(\sum_{a=1}^{s}\left(C_{\xi\xi}^{-1}(1-\sqrt{1-\Delta}b_a)^2 + C_{\xi\eta}^{-1}b_a(1-\sqrt{1-\Delta}b_a)\mathbb{I}_d\right)\right)}_{\equiv g_\eta^\mu}\boldsymbol{\mu}_k. \quad (27)$$

Again, observe that $g_\eta^\mu, g_\xi^\mu = \mathcal{O}(s)$ and will be safely neglected later in the derivation. Therefore

$$\mathbb{E}_{s,\boldsymbol{\eta},\boldsymbol{\xi}}(\star) = \sum_{k=1}^{K} \rho_k \frac{e^{-\frac{\beta}{2}\|\boldsymbol{\mu}_k\|^2 \sum_{a=1}^{s}(\sqrt{1-\Delta}b_a-1)^2 + \frac{1}{2}C_{\xi\xi}^{-1}\|\boldsymbol{\mu}_\xi\|^2 + \frac{1}{2}\boldsymbol{\mu}_\eta^\top \boldsymbol{C}_{\eta\eta}^{-1}\boldsymbol{\mu}_\eta + C_{\xi\eta}^{-1}\boldsymbol{\mu}_\xi^\top \boldsymbol{\mu}_\eta}}{\sqrt{\det \boldsymbol{\Sigma}_k}\Delta^{\frac{d}{2}} \det\left(\boldsymbol{C}_{\eta\eta}^{-1}C_{\xi\xi}^{-1} - (C_{\xi\eta}^{-1})^2\mathbb{I}_d\right)^{\frac{1}{2}}}$$

$$\times \underbrace{\left\langle e^{-\frac{\beta}{2}\sum_{a=1}^{s}\left[\text{Tr}\left[\frac{\boldsymbol{w}_a\boldsymbol{w}_a^\top}{d}\sigma\left(\frac{\boldsymbol{w}_a(\sqrt{1-\Delta}(\boldsymbol{\mu}_k+\boldsymbol{\eta})+\boldsymbol{\xi})}{\sqrt{d}}\right)^{\otimes 2}\right] - 2\sigma\left(\frac{\boldsymbol{w}_a(\sqrt{1-\Delta}(\boldsymbol{\mu}_k+\boldsymbol{\eta})+\boldsymbol{\xi})}{\sqrt{d}}\right)^\top \frac{\boldsymbol{w}_a((1-b_a\sqrt{1-\Delta})(\boldsymbol{\mu}_k+\boldsymbol{\eta})-b_a\boldsymbol{\xi})}{\sqrt{d}}\right]}\right\rangle_{P_{\text{eff.}}(\boldsymbol{\eta},\boldsymbol{\xi})}}_{(a)}. \quad (28)$$

We are now in a position to introduce the local fields:

$$\lambda_a^\xi = \frac{\boldsymbol{w}_a(\boldsymbol{\xi} - \boldsymbol{\mu}_\xi)}{\sqrt{d}}, \qquad \lambda_a^\eta = \frac{\boldsymbol{w}_a(\boldsymbol{\eta} - \boldsymbol{\mu}_\eta)}{\sqrt{d}}, \qquad h_a^k = \frac{\boldsymbol{w}_a\boldsymbol{\mu}_k}{\sqrt{d}}, \quad (29)$$

with statistics

$$\langle \lambda_a^\eta \lambda_b^\eta \rangle \approx \frac{\boldsymbol{w}_a \boldsymbol{\Sigma}_k \boldsymbol{w}_b^\top}{d}, \qquad \langle \lambda_a^\xi \lambda_b^\eta \rangle = C_{\xi\eta}\frac{\boldsymbol{w}_a\boldsymbol{w}_b^\top}{d} \approx 0, \qquad \langle \lambda_a^\xi \lambda_b^\xi \rangle \approx \Delta\frac{\boldsymbol{w}_a\boldsymbol{w}_b^\top}{d}. \quad (30)$$

We used the leading order of the covariances $C_{\xi\xi,\eta\eta,\eta\xi}$. One therefore has to introduce the summary statistics:

$$Q_{ab} = \frac{\boldsymbol{w}_a \boldsymbol{w}_b^\top}{d} \in \mathbb{R}^{p\times p}, \qquad S_{ab}^k = \frac{\boldsymbol{w}_a \boldsymbol{\Sigma}_k \boldsymbol{w}_b^\top}{d} \in \mathbb{R}^{p\times p}, \qquad m_a^k = \frac{\boldsymbol{w}_a \boldsymbol{\mu}_k}{\sqrt{d}} \in \mathbb{R}^p. \qquad (31)$$

Note that the local fields $\lambda_a^{\eta,\xi}$ thus follow a Gaussian distribution:

$$(\lambda_a^\xi, \lambda_a^\eta)_{a=1}^s \sim \mathcal{N}\left( 0, \underbrace{\begin{bmatrix} S^k & 0 \\ 0 & \Delta Q \end{bmatrix}}_{\equiv \Omega_k} \right). \qquad (32)$$

Going back to the computation, $(a)$ can be rewritten as

$$\left\langle e^{-\frac{\beta}{2}\sum\limits_{a=1}^s \operatorname{Tr}\left[ Q_{aa}\sigma\left( m_a^k(\sqrt{1-\Delta}(1+g_\eta^\mu)+g_\xi^\mu)+\sqrt{1-\Delta}\lambda_a^\eta+\lambda_a^\xi \right)^{\otimes 2} \right]} \right.$$

$$\left. \times\, e^{-\frac{\beta}{2}\sum\limits_{a=1}^s \left[ -2\sigma\left( m_a^k(\sqrt{1-\Delta}(1+g_\eta^\mu)+g_\xi^\mu)+\sqrt{1-\Delta}\lambda_a^\eta+\lambda_a^\xi \right)^\top \left( m_a^k((1-\sqrt{1-\Delta}b_a)(1+g_\eta^\mu)-b_a g_\xi^\mu)+(1-\sqrt{1-\Delta}b_a)\lambda_a^\eta-b_a\lambda_a^\xi \right) \right]} \right\rangle_{\{\lambda_a^\xi,\lambda_a^\eta\}_{a=1}^s}. \qquad (33)$$

### A.3 Reformulating as a saddle-point problem

Introducing Dirac functions enforcing the definitions of $Q_{ab}, m_a$ brings the replicated function in the following form:

$$\mathbb{E}Z^s = \int \prod_{a=1}^s db_a \prod_{a,b} dQ_{ab} d\hat{Q}_{ab} \prod_{k=1}^K \prod_{a=1}^s dm_a d\hat{m}_a \prod_{k=1}^K \prod_{a,b} dS_{ab}^k d\hat{S}_{ab}^k$$

$$\underbrace{e^{-d\sum\limits_{a\leq b} Q_{ab}\hat{Q}_{ab} - d\sum\limits_{k=1}^K \sum\limits_{a\leq b} S_{ab}^k \hat{S}_{ab}^k - \sum\limits_{k=1}^K \sum\limits_a dm_a^k \hat{m}_a^k}}_{e^{\beta s d \Psi_t}}$$

$$\underbrace{\int \prod_{a=1}^s dw_a e^{-\beta\sum\limits_a g(\boldsymbol{w}^a) + \sum\limits_{a\leq b}\hat{Q}_{ab}\boldsymbol{w}_a\boldsymbol{w}_b^\top + \sum\limits_{k=1}^K \sum\limits_{a\leq b}\hat{S}_{ab}^k \boldsymbol{w}_a\boldsymbol{\Sigma}_k\boldsymbol{w}_b^\top + \sum\limits_{k=1}^K \sum\limits_a \hat{m}_a^k \sqrt{d}\boldsymbol{w}_a\boldsymbol{\mu}_k}}_{e^{\beta s d \Psi_w}}$$

$$\underbrace{\left[ \sum_{k=1}^K \rho_k \frac{e^{\mathcal{O}(s^2)e^{-\frac{\beta}{2}\|\boldsymbol{\mu}_k\|^2 \sum\limits_{a=1}^s(\sqrt{1-\Delta}b_a-1)^2}}}{\sqrt{\det \boldsymbol{\Sigma}_k}\Delta^{\frac{d}{2}} \det\left( \boldsymbol{C}_{\eta\eta}^{-1} C_{\xi\xi}^{-1} - \mathcal{O}(s^2) \right)^{\frac{1}{2}}} (a) \right]^{\alpha d}}_{e^{\alpha s d^2 \Psi_{\text{quad.}} + \beta s d \Psi_y}}. \qquad (34)$$

We introduced the trace, entropic and energetic potentials $\Psi_t, \Psi_w, \Psi_y$. Since all the integrands scale exponentially (or faster) with $d$, this integral can be computed using a saddle-point method. To proceed further, note that the energetic term encompasses two types of terms, scaling like $sd^2$ and $sd$. More precisely,

$$\left[ \sum_{k=1}^K \rho_k \frac{e^{-\frac{\beta}{2}\|\boldsymbol{\mu}_k\|^2 \sum\limits_{a=1}^s(\sqrt{1-\Delta}b_a-1)^2}}{\sqrt{\det \boldsymbol{\Sigma}_k}\Delta^{\frac{d}{2}} \det\left( \boldsymbol{C}_{\eta\eta}^{-1} C_{\xi\xi}^{-1} \right)^{\frac{1}{2}}} (a) \right]^{\alpha d}$$

$$= \left[ 1 - sd\sum_{k=1}^K \rho_k \frac{1}{sd} \ln\left( \sqrt{\det \boldsymbol{\Sigma}_k}\Delta^{\frac{d}{2}} \det\left( \boldsymbol{C}_{\eta\eta}^{-1} C_{\xi\xi}^{-1} \right)^{\frac{1}{2}} \right) + s\sum_{k=1}^K \rho_k \frac{1}{s}\ln(a) - \frac{\beta}{2}\sum_{k=1}^K \rho_k\|\boldsymbol{\mu}_k\|^2 \sum_{a=1}^s(\sqrt{1-\Delta}b_a-1)^2 \right]^{\alpha d}$$

$$\approx e^{-s\alpha d^2 \sum\limits_{k=1}^K \rho_k \frac{1}{sd}\ln\left( \sqrt{\det \boldsymbol{\Sigma}_k}\Delta^{\frac{d}{2}} \det\left( \boldsymbol{C}_{\eta\eta}^{-1} C_{\xi\xi}^{-1} \right)^{\frac{1}{2}} \right) + s\alpha d \sum\limits_{k=1}^K \rho_k \frac{1}{s}\ln(a)}. \qquad (35)$$

Note that the saddle-point method involves an intricate extremization over $s \times s$ matrices $Q, S_k$. To make further progress, we assume the extremum is realized at the Replica Symmetric (RS) fixed point

$$
\begin{aligned}
\forall a, \qquad & b_a = b \\
\forall a, \qquad & m_a^k = m_k, \qquad \hat{m}_a^k = \hat{m}_k \\
\forall a, b, \qquad & Q_{ab} = (r - q)\delta_{ab} + q, \qquad \hat{Q}_{ab} = -\left(\frac{\hat{r}}{2} + \hat{q}\right)\delta_{ab} + \hat{q} \\
\forall a, b, \qquad & S_{ab}^k = (r_k - q_k)\delta_{ab} + q_k, \qquad \hat{S}_{ab}^k = -\left(\frac{\hat{r}_k}{2} + \hat{q}_k\right)\delta_{ab} + \hat{q}_k
\end{aligned}
\tag{36}
$$

This is a standard assumption known as the *RS ansatz* [21, 22, 23, 24].

**Quadratic potential**  The quadratic potential $\Psi_{\text{quad.}}$, which correspond to the leading order term in $d$ in the exponent, needs to be extremized first in the framework of our saddle-point analysis. Its expression can be simplified as

$$
\begin{aligned}
\sum_{k=1}^{K} \rho_k \ln\left(\sqrt{\det \boldsymbol{\Sigma}_k}\Delta^{\frac{d}{2}}\det(C_{\eta\eta}^{-1}C_{\xi\xi}^{-1})^{\frac{1}{2}}\right) &= \frac{1}{2}\sum_{k=1}^{K} \rho_k \ln\det\left[(1 + \Delta\beta s b^2)\left(\mathbb{I}_d + \boldsymbol{\Sigma}_k \beta s(\sqrt{1-\Delta}b - 1)^2\right)\right] \\
&= \frac{\beta s}{2}\sum_{k=1}^{K} \rho_k \operatorname{Tr}\left[\Delta b^2 \mathbb{I}_d + \boldsymbol{\Sigma}_k(\sqrt{1-\Delta}b - 1)^2\right] \\
&= \frac{\beta d s}{2}\left[\Delta b^2 + (\sqrt{1-\Delta}b - 1)^2\frac{1}{d}\sum_{k=1}^{K} \rho_k \operatorname{Tr}\boldsymbol{\Sigma}_k\right],
\end{aligned}
\tag{37}
$$

which is extremized for

$$
b = \frac{\frac{1}{d}\left(\sum\limits_{k=1}^{K} \rho_k \operatorname{Tr}\boldsymbol{\Sigma}_k\right)\sqrt{1-\Delta}}{\frac{1}{d}\left(\sum\limits_{k=1}^{K} \rho_k \operatorname{Tr}\boldsymbol{\Sigma}_k\right)(1-\Delta) + \Delta}.
\tag{38}
$$

This fixes the skip connection strength $b$.

**Entropic potential**  We now turn to the entropic potential $\Psi_w$. It is convenient to introduce the variance order parameters

$$
\hat{V} \equiv \hat{r} + \hat{q}, \qquad\qquad\qquad \hat{V}_k \equiv \hat{r}_k + \hat{q}_k.
\tag{39}
$$

The entropic potential can then be expressed as

$$e^{\beta s d \Psi_w}$$

$$= \int \prod_{a=1}^{s} dw_a e^{-\beta \sum_a g(w^a) - \frac{1}{2} \sum_{a=1}^{s} \mathrm{Tr}[\hat{V} w_a w_a^\top] + \frac{1}{2} \sum_{a,b} \mathrm{Tr}[\hat{q} w_a w_b^\top] + \hat{m} \sum_{k=1}^{K} \sum_{a=1}^{s} \sqrt{d} \hat{m}_k^\top w_a^\top \mu_k}$$

$$\times e^{-\frac{1}{2} \sum_{k=1}^{K} \sum_{a=1}^{s} \mathrm{Tr}[\hat{V}_k w_a \Sigma_k w_a^\top] + \frac{1}{2} \sum_{k=1}^{K} \sum_{a,b} \mathrm{Tr}[\hat{q}_k w_a \Sigma_k w_b^\top]}$$

$$= \int D\Xi_0 \prod_{k=1}^{K} D\Xi_k$$

$$\left[ \int dw e^{-\beta g(w) - \frac{1}{2} \mathrm{Tr}\left[\hat{V} w w^\top + \sum_{k=1}^{K} \hat{V}_k w \Sigma_k w^\top\right] + \left( \sum_{k=1}^{K} \sqrt{d} \hat{m}_k \mu^\top + \sum_{k=1}^{K} \Xi_k \odot (\hat{q}_k \otimes \Sigma_k)^{\frac{1}{2}} + \Xi_0 \odot (\hat{q} \otimes \mathbb{I}_d)^{\frac{1}{2}} \right) \odot w} \right]^s$$

$$= \int \underbrace{D\Xi_0 \prod_{k=1}^{K} D\Xi_k}_{\equiv D\Xi}$$

$$\left[ \int dw e^{-\beta g(w) - \frac{1}{2} w \odot \left[\hat{V} \otimes \mathbb{I}_d + \sum_{k=1}^{K} \hat{V}_k \otimes \Sigma_k\right] \odot w + \left( \sum_{k=1}^{K} \sqrt{d} \hat{m}_k \mu^\top + \sum_{k=1}^{K} \Xi_k \odot (\hat{q}_k \otimes \Sigma_k)^{\frac{1}{2}} + \Xi_0 \odot (\hat{q} \otimes \mathbb{I}_d)^{\frac{1}{2}} \right) \odot w} \right]^s. \tag{40}$$

Therefore

$$\beta \Psi_w = \frac{1}{d} \int D\Xi \ln \left[ \int dw e^{-\beta g(w) - \frac{1}{2} w \odot \left[\hat{V} \otimes \mathbb{I}_d + \sum_{k=1}^{K} \hat{V}_k \otimes \Sigma_k\right] \odot w + \left( \sum_{k=1}^{K} \sqrt{d} \hat{m}_k \mu^\top + \sum_{k=1}^{K} \Xi_k \odot (\hat{q}_k \otimes \Sigma_k)^{\frac{1}{2}} + \Xi_0 \odot (\hat{q} \otimes \mathbb{I}_d)^{\frac{1}{2}} \right) \odot w} \right]. \tag{41}$$

For a matrix $\Xi \in \mathbb{R}^{p \times d}$ and tensors $A, B \in \mathbb{R}^{p \times d} \otimes \mathbb{R}^{p \times d}$, we denoted $(\Xi \odot A)_{kl} = \sum_{ij} \Xi^{ij} A_{ij,kl}$ and $(A \odot B)_{ij,kl} = \sum_{rs} A_{ij,rs} B_{rs,kl}$.

**Energetic potential** In order to compute the energetic potential $\Psi_y$, one must first compute the inverse of the covariance $\Omega_k$, given by

$$\Omega_k^{-1} = \begin{bmatrix} S_k^{-1} & 0 \\ 0 & \frac{1}{\Delta} Q^{-1} \end{bmatrix}. \tag{42}$$

It is straightforward to see that $S_k^{-1}, Q^{-1}$ share the same block structure as $S_k, Q$, and we shall name for clarity

$$\begin{aligned} \left(S_k^{-1}\right)_{ab} &= (\tilde{r}_k - \tilde{q}_k) \delta_{ab} + \tilde{q}_k, \\ \left(Q^{-1}\right)_{ab} &= (\tilde{r} - \tilde{q}) \delta_{ab} + \tilde{q}. \end{aligned} \tag{43}$$

From the Sherman-Morisson lemma, and noting

$$V_k \equiv r_k - q_k \in \mathbb{R}^{p \times p}, \qquad\qquad V \equiv r - q \in \mathbb{R}^{p \times p}, \tag{44}$$

one reaches

$$\begin{cases} \tilde{r} = V^{-1} - (V + sq)^{-1} q V^{-1} \\ \tilde{q} = -(V + sq)^{-1} q V^{-1} \end{cases}, \qquad \begin{cases} \tilde{r}_k = V_k^{-1} - (V_k + sq_k)^{-1} q_k V_k^{-1} \\ \tilde{q}_k = -(V_k + sq_k)^{-1} q_k V_k^{-1} \end{cases}. \tag{45}$$

We will also need the expression of the determinants

$$\ln \det S_k = s \det V_k + s \mathrm{Tr}\left[V_k^{-1} q_k\right], \qquad\qquad Q = s \det V + s \mathrm{Tr}\left[V^{-1} q\right]. \tag{46}$$

One can finally proceed in analyzing the term $(a)$:

$$
(a) = \int_{\mathbb{R}^p} \frac{\prod\limits_{a=1}^{s} d\lambda_a^\eta d\lambda_a^\xi}{(2\pi)^{ps}\sqrt{\det Q \det S_k}\sqrt{\Delta}^{ps}}
$$

$$
\times\, e^{-\frac{1}{2}\sum\limits_{a=1}^{s}(\lambda_a^\eta)^\top(\tilde{r}_k-\tilde{q}_k)\lambda_a^\eta-\frac{1}{2}\sum\limits_{a,b}(\lambda_a^\eta)^\top \tilde{q}_k\lambda_b-\frac{1}{2\Delta}\sum\limits_{a=1}^{s}(\lambda_a^\xi)^\top(\tilde{r}-\tilde{q})\lambda_a^\xi-\frac{1}{2\Delta}\sum\limits_{a,b}(\lambda_a^\xi)^\top \tilde{q}\lambda_b}\, e^{-\frac{\beta}{2}\sum\limits_{a=1}^{s}(*)}
$$

$$
= \int_{\mathbb{R}^p} \frac{D\xi D\eta}{(2\pi)^{ps}\sqrt{\det Q \det S_k}\sqrt{\Delta}^{ps}}
$$

$$
\times \left[\int_{\mathbb{R}^p} d\lambda_\eta d\lambda_\xi e^{-\frac{1}{2}\lambda_\eta^\top V_k^{-1}\lambda_\eta+\lambda_\eta^\top V_k^{-1}q_k^{\frac{1}{2}}\eta-\frac{1}{2\Delta}\lambda_\xi^\top V^{-1}\lambda_\xi+\frac{1}{\sqrt{\Delta}}\lambda_\xi^\top V^{-1}q^{\frac{1}{2}}\xi-\frac{\beta}{2}(*)}\right]^s
$$

$$
= \int_{\mathbb{R}^p} D\xi D\eta \left[\int \mathcal{N}\left(\lambda_\xi; \sqrt{\Delta}q^{\frac{1}{2}}\xi, \Delta V\right)\mathcal{N}\left(\lambda_\eta; q_k^{\frac{1}{2}}\xi, V_k\right)e^{-\frac{\beta}{2}(*)}\right]^s
$$

$$
= 1 + s\int_{\mathbb{R}^p} D\xi D\eta \ln\left[\int \mathcal{N}\left(\lambda_\xi; \sqrt{\Delta}q^{\frac{1}{2}}\xi, \Delta V\right)\mathcal{N}\left(\lambda_\eta; q_k^{\frac{1}{2}}\xi, V_k\right)e^{-\frac{\beta}{2}(*)}\right] \tag{47}
$$

where we noted with capital $D$ an integral over a $p-$ dimensional Gaussian distribution $\mathcal{N}(0, \mathbb{I}_p)$. Therefore

$$
\beta\Psi_y = \sum_{k=1}^{K}\rho_k \int_{\mathbb{R}^p} D\xi D\eta \ln\left[\int \mathcal{N}\left(\lambda_\xi; \sqrt{\Delta}q^{\frac{1}{2}}\xi, \Delta V\right)\mathcal{N}\left(\lambda_\eta; q_k^{\frac{1}{2}}\xi, V_k\right)e^{-\frac{\beta}{2}(*)}\right]. \tag{48}
$$

### A.4 zero-temperature limit

In this subsection, we take the zero temperature $\beta \to \infty$ limit. Rescaling

$$
\frac{1}{\beta}V \leftarrow V, \qquad\qquad \beta\hat{V} \leftarrow \hat{V}, \qquad\qquad \beta^2\hat{q} \leftarrow \hat{q},
$$

$$
\frac{1}{\beta}V_k \leftarrow V_k, \qquad\qquad \beta\hat{V}_k \leftarrow \hat{V}_k, \qquad\qquad \beta^2\hat{q}_k \leftarrow \hat{q}_k, \qquad\qquad \beta\hat{m}_k \leftarrow \hat{m}_k, \tag{49}
$$

one has that

$$
\Psi_w
$$

$$
= \frac{1}{2d}\operatorname{Tr}\left[\left(\hat{V}\otimes\mathbb{I}_d+\sum_{k=1}^{K}\hat{V}_k\otimes\boldsymbol{\Sigma}_k\right)^{-1}\odot\left(\hat{q}\otimes\mathbb{I}_d+\sum_{k=1}^{K}\hat{q}_k\otimes\boldsymbol{\Sigma}_k+d\left(\sum_{k=1}^{K}\hat{m}_k\boldsymbol{\mu}_k^\top\right)^{\otimes 2}\right)\right]-\frac{1}{d}\mathbb{E}_{\boldsymbol{\Xi}}\mathcal{M}_r(\boldsymbol{\Xi}), \tag{50}
$$

where we introduced the Moreau enveloppe

$$
\mathcal{M}_r(\boldsymbol{\Xi})
$$

$$
= \inf_{\boldsymbol{w}}\left\{\frac{1}{2}\left\|\left(\hat{V}\otimes\mathbb{I}_d+\sum_{k=1}^{K}\hat{V}_k\otimes\boldsymbol{\Sigma}_k\right)^{\frac{1}{2}}\odot\boldsymbol{w}-\left(\hat{V}\otimes\mathbb{I}_d+\sum_{k=1}^{K}\hat{V}_k\otimes\boldsymbol{\Sigma}_k\right)^{-\frac{1}{2}}\odot\left((\hat{q}\otimes\mathbb{I}_d)^{\frac{1}{2}}\odot\boldsymbol{\Xi}_0+\sum_{k=1}^{K}(\hat{q}_k\otimes\boldsymbol{\Sigma}_k)^{\frac{1}{2}}\odot\boldsymbol{\Xi}_k+\sqrt{d}\sum_{k=1}^{K}\hat{m}_k\boldsymbol{\mu}_k^\top\right)\right\|^2+g(\boldsymbol{w})\right\}. \tag{51}
$$

Note that while this optimization problem is still cast in a space of dimension $p \times d$, for some regularizers $g(\cdot)$ including the usual $\ell_{1,2}$ penalties, the Moreau enveloppe admits a compact analytical expression.

The energetic potential $\Psi_y$ also simplifies in this limit to

$$
\Psi_y = -\sum_{k=1}^{K}\rho_k \mathbb{E}_{\eta,\xi}\mathcal{M}_k(\eta, \xi), \tag{52}
$$

Where

$$\mathcal{M}_k(\xi, \eta) = \frac{1}{2}\inf_{x,y}\left\{ \mathrm{Tr}\left[V_k^{-1}\left(y - q_k^{\frac{1}{2}}\eta - m_k\right)^{\otimes 2}\right] + \frac{1}{\Delta}\mathrm{Tr}\left[V^{-1}\left(x - \sqrt{\Delta}q^{\frac{1}{2}}\xi\right)^{\otimes 2}\right]\right.$$
$$\left.\mathrm{Tr}\left[q\sigma(\sqrt{1-\Delta}y + x)^{\otimes 2}\right] - 2\sigma(\sqrt{1-\Delta}y + x)^{\top}((1 - \sqrt{1-\Delta}b)y - bx)\right\}. \tag{53}$$

It is immediate to see that the trace potential $\Psi_t$ can be expressed as

$$\Psi_t = \frac{\mathrm{Tr}\left[\hat{V}q\right] - \mathrm{Tr}[\hat{q}V]}{2} + \frac{1}{2}\sum_{k=1}^{K}(\mathrm{Tr}\left[\hat{V}_k q_k\right] - \mathrm{Tr}[\hat{q}_k V_k]) - \sum_{k=1}^{K} m_k \hat{m}_k. \tag{54}$$

Putting these results together, the total free entropy reads

$$\Phi = \underset{q,m,V,\hat{q},\hat{m},\hat{V},\{q_k,m_k,V_k,\hat{q}_k,\hat{m}_k,\hat{V}_k\}_{k=1}^{K}}{\mathrm{extr}} \frac{\mathrm{Tr}\left[\hat{V}q\right] - \mathrm{Tr}[\hat{q}V]}{2} + \frac{1}{2}\sum_{k=1}^{K}(\mathrm{Tr}\left[\hat{V}_k q_k\right] - \mathrm{Tr}[\hat{q}_k V_k]) - \sum_{k=1}^{K} m_k \hat{m}_k$$
$$- \alpha \sum_{k=1}^{K} \rho_k \mathbb{E}_{\xi,\eta} \mathcal{M}_k(\xi, \eta) - \frac{1}{d}\mathbb{E}_{\Xi}\mathcal{M}_r(\Xi)$$
$$+ \frac{1}{2d}\mathrm{Tr}\left[\left(\hat{V}\otimes\mathbb{I}_d + \sum_{k=1}^{K}\hat{V}_k\otimes\mathbf{\Sigma}_k\right)^{-1}\odot\left(\hat{q}\otimes\mathbb{I}_d + \sum_{k=1}^{K}\hat{q}_k\otimes\mathbf{\Sigma}_k + d\left(\sum_{k=1}^{K}\hat{m}_k\boldsymbol{\mu}_k^{\top}\right)^{\otimes 2}\right)\right] \tag{55}$$

where we remind that

$$\mathcal{M}_k(\xi, \eta) = \frac{1}{2}\inf_{x,y}\left\{ \mathrm{Tr}\left[V_k^{-1}\left(y - q_k^{\frac{1}{2}}\eta - m_k\right)^{\otimes 2}\right] + \mathrm{Tr}\left[V^{-1}\left(x - q^{\frac{1}{2}}\xi\right)^{\otimes 2}\right]\right.$$
$$\left.+ \mathrm{Tr}\left[q\sigma(\sqrt{1-\Delta}y + x)^{\otimes 2}\right] - 2\sigma(\sqrt{1-\Delta}y + x)^{\top}((1 - \sqrt{1-\Delta}b)y - bx)\right\}. \tag{56}$$

and

$$\mathcal{M}_r(\Xi)$$
$$= \inf_{\boldsymbol{w}}\left\{\frac{1}{2}\left\|\left(\hat{V}\otimes\mathbb{I}_d + \sum_{k=1}^{K}\hat{V}_k\otimes\mathbf{\Sigma}_k\right)^{\frac{1}{2}}\odot\boldsymbol{w} - \left(\hat{V}\otimes\mathbb{I}_d + \sum_{k=1}^{K}\hat{V}_k\otimes\mathbf{\Sigma}_k\right)^{-\frac{1}{2}}\odot\left((\hat{q}\otimes\mathbb{I}_d)^{\frac{1}{2}}\odot\Xi_0 + \sum_{k=1}^{K}(\hat{q}_k\otimes\mathbf{\Sigma}_k)^{\frac{1}{2}}\odot\Xi_k + \sqrt{d}\sum_{k=1}^{K}\hat{m}_k\boldsymbol{\mu}_k^{\top}\right)\right\|^2 + r(\boldsymbol{w})\right\} \tag{57}$$

## A.5 Self-consistent equations

The extremization problem (58) can be recast as a system of self-consistent equations by requiring the gradient with respect to each summary statistic involved in (58) to be zero. This translates into:

$$
\begin{cases}
\hat{q}_k = \alpha\rho_k \mathbb{E}_{\xi,\eta} V_k^{-1} \left( \mathrm{prox}_y^k - q_k^{\frac{1}{2}}\eta - m_k \right)^{\otimes 2} V_k^{-1} \\
\hat{V}_k = -\alpha\rho_k q_k^{-\frac{1}{2}} \mathbb{E}_{\xi,\eta} V_k^{-1} \left( \mathrm{prox}_y^k - q_k^{\frac{1}{2}}\eta - m_k \right) \eta^\top \\
\hat{m}_k = \alpha\rho_k \mathbb{E}_{\xi,\eta} V_k^{-1} \left( \mathrm{prox}_y^k - q_k^{\frac{1}{2}}\eta - m_k \right) \\
\hat{q} = \alpha \sum_{k=1}^K \rho_k \mathbb{E}_{\xi,\eta} V^{-1} \left( \mathrm{prox}_x^k - q^{\frac{1}{2}}\xi \right)^{\otimes 2} V^{-1} \\
\hat{V} = -\alpha \sum_{k=1}^K \rho_k \mathbb{E}_{\xi,\eta} \left[ q^{-\frac{1}{2}} V^{-1} \left( \mathrm{prox}_x^k - q^{\frac{1}{2}}\xi \right) \xi^\top - \sigma\left( \sqrt{1-\Delta}\mathrm{prox}_y + \mathrm{prox}_x \right)^{\otimes 2} \right]
\end{cases}
\tag{58}
$$

$$
\begin{cases}
V_k = \frac{1}{d}\mathbb{E}_{\boldsymbol{\Xi}} \left[ \left( \mathrm{prox}_r \odot (\hat{q}_k \otimes \boldsymbol{\Sigma}_k)^{-\frac{1}{2}} \odot (\mathbb{I}_p \otimes \boldsymbol{\Sigma}_k) \right) \boldsymbol{\Xi}_k^\top \right] \\
q_k = \frac{1}{d}\mathbb{E}_{\boldsymbol{\Xi}} \left[ \mathrm{prox}_r \boldsymbol{\Sigma}_k \mathrm{prox}_r^\top \right] \\
m_k = \frac{1}{\sqrt{d}}\mathbb{E}_{\xi,\eta} \left[ \mathrm{prox}_r \boldsymbol{\mu}_k \right] \\
V = \frac{1}{d}\mathbb{E}_{\boldsymbol{\Xi}} \left[ \left( \mathrm{prox}_r \odot (\hat{q} \otimes \mathbb{I}_d)^{-\frac{1}{2}} \right) \boldsymbol{\Xi}_0^\top \right] \\
q = \frac{1}{d}\mathbb{E}_{\boldsymbol{\Xi}} \left[ \mathrm{prox}_r \mathrm{prox}_r^\top \right]
\end{cases}
\tag{59}
$$

We remind that the skip connection $b$ is fixed to

$$
b = \frac{\frac{1}{d}\left( \sum_{k=1}^K \rho_k \operatorname{Tr}\boldsymbol{\Sigma}_k \right)\sqrt{1-\Delta}}{\frac{1}{d}\left( \sum_{k=1}^K \rho_k \operatorname{Tr}\boldsymbol{\Sigma}_k \right)(1-\Delta) + \Delta}.
\tag{60}
$$

## A.6 Sharp asymptotic formulae for the learning metrics

The previous subsections have allowed to obtain sharp asymptotic characterization for a number of summary statistics of the probability (17). These statistics are in turn sufficient to asymptotically characterize the learning metrics discussed in the main text. We successively address the test MSE (5), cosine similarity (6) and training MSE.

**MSE**  The denoising MSE reads

$$
\mathrm{mse} = \mathbb{E}_{k,\boldsymbol{\eta},\boldsymbol{\xi}} \left\| \boldsymbol{\mu}_k + \boldsymbol{\eta} - \left( \hat{b}\sqrt{1-\Delta}(\boldsymbol{\mu}_k + \boldsymbol{\eta}) + \hat{b}\sqrt{\Delta}\boldsymbol{\xi} + \frac{\hat{\boldsymbol{w}}^\top}{\sqrt{d}}\sigma\left( \frac{\hat{\boldsymbol{w}}(\sqrt{1-\Delta}(\boldsymbol{\mu}_k + \boldsymbol{\eta}) + \sqrt{\Delta}\boldsymbol{\xi})}{\sqrt{d}} \right) \right) \right\|^2
$$

$$
= \mathrm{mse}_\circ + \sum_{k=1}^K \rho_k \mathbb{E}_z^{\mathcal{N}(0,1)} \left[ \operatorname{Tr}\left[ q\sigma\left( \sqrt{1-\Delta}m_k + \sqrt{\Delta q + (1-\Delta)q_k}z \right)^{\otimes 2} \right] \right]
$$

$$
- 2\sum_{k=1}^K \rho_k \mathbb{E}_{u,v}^{\mathcal{N}(0,1)} \left[ \sigma\left( \sqrt{1-\Delta}m_k + \sqrt{q_k(1-\Delta)}u + \sqrt{q}\sqrt{\Delta}v \right) \right]^\top \left( (1-b\sqrt{1-\Delta})m_k + (1-b\sqrt{1-\Delta})\sqrt{q_k}u - b\sqrt{q}\sqrt{\Delta}v \right)
\tag{61}
$$

where

$$
\mathrm{mse}_\circ = d\Delta b^2 + (1-\sqrt{1-\Delta}b)^2 \left( \sum_{k=1}^K \rho_k \left( \|\boldsymbol{\mu}_k\|^2 + \operatorname{Tr}\boldsymbol{\Sigma}_k \right) \right)
\tag{62}
$$

**Cosine similarity**  By the very definition of the summary statistics $m_k, q$:

$$
\theta_{ik} \equiv \frac{\hat{\boldsymbol{w}}_i \boldsymbol{\mu}_k}{\|\hat{\boldsymbol{w}}\| \times \|\boldsymbol{\mu}_k\|} = \frac{(m_k)_i}{\sqrt{q_{ii}}\|\boldsymbol{\mu}_k\|}
\tag{63}
$$

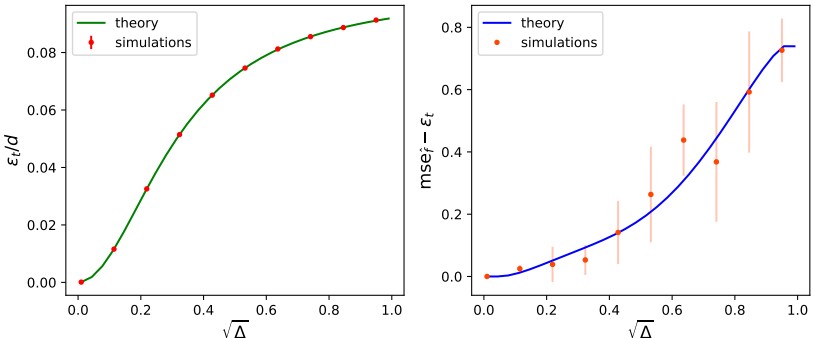

Figure 5: (left) Training MSE for the full DAE (2) ($p = 1$, $\sigma = \tanh$). Solid lines represent the sharp asymptotic formula (68); dots correspond to simulation, training the DAE with the `Pytorch` implementation of full-batch Adam, over $T = 2000$ epochs using learning rate $\eta = 0.05$ and weight decay $\lambda = 0.1$. The data was averaged over $N = 5$ instances; error bars are smaller than the point size. (right) Generalization gap $\mathrm{mse}_{\hat{f}} - \epsilon_t$. Solid lines correspond to the asymptotic prediction of Conjecture 3.3 (for the test MSE) and of (68) (for the train MSE), while dots correspond to simulations. Error bars represent one standard deviation. The Gaussian mixture is the isotropic binary mixture, whose parameters are specified in the caption of Fig. 1 in the main text.

**Training error**   The replica formalism also allows to compute a sharp asymptotic characterization for the training MSE

$$\epsilon_t \equiv \mathbb{E}_{\mathcal{D}} \left[ \frac{1}{nd} \sum_{\mu=1}^{n} \left\| \boldsymbol{x}^{\mu} - \hat{f}(\tilde{\boldsymbol{x}}^{\mu}) \right\|^2 \right]. \tag{64}$$

We have skipped the discussion of this learning metric in the main text for the sake of conciseness, but will now detail the derivation. Note that the training error $\epsilon_t$ can be deduced from the risk (4) and the average regularization as

$$\frac{1}{2}\epsilon_t = \mathbb{E}_{\mathcal{D}} \left[ \hat{\mathcal{R}}(\hat{\boldsymbol{w}}, \hat{b}) \right] - \mathbb{E}_{\mathcal{D}} \left[ r(\hat{\boldsymbol{w}}) \right]. \tag{65}$$

Note that in turn, from the definition of the free entropy, the average risk (training loss) can be computed as

$$\mathbb{E}_{\mathcal{D}} \left[ \hat{\mathcal{R}}(\hat{\boldsymbol{w}}, \hat{b}) \right] = - \lim_{\beta \to \infty} \frac{1}{\beta} \frac{\partial \ln Z}{\partial \beta} = - \lim_{\beta \to \infty} \frac{1}{\beta} \frac{\partial \Phi}{\partial \beta}. \tag{66}$$

Doing so reveals

$$\epsilon_t = -2 \lim_{\beta \to \infty} \frac{\partial (\Psi_y - d\Psi_{\mathrm{quad}} - \frac{\beta}{2} m (\sqrt{1-\Delta} b - 1)^2))}{\partial \beta}$$

$$= \mathrm{mse}_\circ + \lim_{\beta \to \infty} \sum_{k=1}^{K} \rho_k \int_{\mathbb{R}^p} D\xi D\eta \frac{\int \mathcal{N}\left(\lambda_\xi; q^{\frac{1}{2}}\xi, V_k\right) \mathcal{N}\left(\lambda_\eta; q_k^{\frac{1}{2}}\xi, V\right)(*) e^{-\frac{\beta}{2}(*)}}{\int \mathcal{N}\left(\lambda_\xi; q^{\frac{1}{2}}\xi, V_k\right) \mathcal{N}\left(\lambda_\eta; q_k^{\frac{1}{2}}\xi, V\right) e^{-\frac{\beta}{2}(*)}}. \tag{67}$$

Taking the $\beta \to \infty$ limit finally leads to

$$\epsilon_t = \mathrm{mse}_\circ + \sum_{k=1}^{K} \rho_k \mathbb{E}_{\xi, \eta} \left[ \mathrm{Tr}\left[ q\sigma(\sqrt{1-\Delta}\mathrm{prox}_y + \mathrm{prox}_x)^{\otimes 2} \right] \right.$$

$$\left. - 2\sigma(\sqrt{1-\Delta}\mathrm{prox}_y + \mathrm{prox}_x)^\top ((1 - \sqrt{1-\Delta}\hat{b})\mathrm{prox}_y - \hat{b}\mathrm{prox}_x) \right] \tag{68}$$

The sharp asymptotic formula (68) agrees well with numerical simulations, Fig. 5. As is intuitive, the generalization gap $\mathrm{mse}_{\hat{f}} - \epsilon_t$ grows with the noise level $\Delta$, as the learning problem grows harder.

## A.7 Generic asymptotically exact formulae

This last result ends the derivation of the generic version of Conjecture 3.3, i.e. *not assuming* Assumptions 3.1 and 3.2.

Importantly, note that the characterization (58) is, like the formulae in e.g. [47, 48], *asymptotically exact*, but not *fully* asymptotic, as the equations (58) still involve high-dimensional quantities. In practice however, for standard regularizers $r(\cdot)$ like $\ell_1$ or $\ell_2$, the proximal $\mathrm{prox}_r$ admits a simple closed-form expression, and the average over $\Xi$ can be carried out analytically. The only high-dimensional operation left is then taking the trace of linear combinations of $\Sigma_k$ matrices (or the inverse of such combinations), which is numerically less demanding, and analytically much simpler, than the high dimensional optimization (4) and averages (5) involved in the original problem. We give in the next paragraph an example of how Assumption 3.1 can be relaxed, for a binary mixture with arbitrary (i.e. not necessarily jointly diagonalizable) covariances $\Sigma_{1,2}$.

**Example: Anisotropic, heteroscedastic binary mixture**   As an illustration, we provide the equivalent of (58) for a binary mixture, with generically anisotropic and distinct covariances, for $p = 1$, thereby breaking free from Assumption 3.1. We index the clusters by $+, -$ rather than $1, 2$ for notational convenience. We remind that

$$\mathcal{M}_\pm(\xi,\eta) = \frac{1}{2}\inf_{x,y}\left\{\tfrac{1}{\Delta V}(x-\sqrt{q}\sqrt{\Delta}\xi)^2 + \tfrac{1}{V_\pm}(y-\sqrt{q_\pm}\eta\mp m)^2 + q\sigma(\sqrt{1-\Delta}y+x)^2 - 2\sigma(\sqrt{1-\Delta}y+x)((1-\sqrt{1-\Delta}b)y-bx)\right\}. \tag{69}$$

Then the self-consistent equations (58) simplify to

$$\begin{cases} q = \frac{1}{d}\,\mathrm{Tr}\left[\left((\lambda+\hat{V})\mathbb{I}_d + \hat{V}_+\Sigma_+ + \hat{V}_-\Sigma_-\right)^{-2}\left(\hat{q}\mathbb{I}_d + \hat{q}_+\Sigma_+ + \hat{q}_-\Sigma_- + \hat{m}^2 d\boldsymbol{\mu}\boldsymbol{\mu}^\top\right)\right] \\[4pt] m = \hat{m}\,\mathrm{Tr}\left[\left((\lambda+\hat{V})\mathbb{I}_d + \hat{V}_+\Sigma_+ + \hat{V}_-\Sigma_-\right)^{-1}\boldsymbol{\mu}\boldsymbol{\mu}^\top\right] \\[4pt] V = \frac{1}{d}\,\mathrm{Tr}\left[\left((\lambda+\hat{V})\mathbb{I}_d + \hat{V}_+\Sigma_+ + \hat{V}_-\Sigma_-\right)^{-1}\right] \\[4pt] q_\pm = \frac{1}{d}\,\mathrm{Tr}\left[\left((\lambda+\hat{V})\mathbb{I}_d + \hat{V}_+\Sigma_+ + \hat{V}_-\Sigma_-\right)^{-1}\Sigma_\pm\left((\lambda+\hat{V})\mathbb{I}_d + \hat{V}_+\Sigma_+ + \hat{V}_-\Sigma_-\right)^{-1}\left(\hat{q}\mathbb{I}_d + \hat{q}_+\Sigma_+ + \hat{q}_-\Sigma_- + \hat{m}^2 d\boldsymbol{\mu}\boldsymbol{\mu}^\top\right)\right] \\[4pt] V_\pm = \frac{1}{d}\,\mathrm{Tr}\left[\Sigma_\pm\left((\lambda+\hat{V})\mathbb{I}_d + \hat{V}_+\Sigma_+ + \hat{V}_-\Sigma_-\right)^{-1}\right] \end{cases}$$

$$\begin{cases} \hat{m} = \alpha\left[\rho\frac{1}{V_+}\mathbb{E}_{\xi,\eta}\left(\mathrm{prox}_y^+ - \sqrt{q_+}\eta - m\right) - (1-\rho)\frac{1}{V_-}\mathbb{E}_{\xi,\eta}\left(\mathrm{prox}_y^- - \sqrt{q_-}\eta + m\right)\right] \\[4pt] \hat{q} = \frac{\alpha}{V^2\Delta}\left[\rho\mathbb{E}_{\xi,\eta}(\mathrm{prox}_x^+ - \sqrt{q}\sqrt{\Delta}\xi)^2 + (1-\rho)\mathbb{E}_{\xi,\eta}(\mathrm{prox}_x^- - \sqrt{q}\sqrt{\Delta}\xi)^2\right] \\[4pt] \hat{V} = -\alpha\left[\rho\mathbb{E}_{\xi,\eta}\left[\frac{\xi(\mathrm{prox}_x^+ - \sqrt{q}\sqrt{\Delta}\xi)}{\sqrt{q}\sqrt{\Delta}V} - \sigma(\sqrt{1-\Delta}\mathrm{prox}_y^+ + \mathrm{prox}_x^+)^2\right]\right. \\[4pt] \qquad\qquad\left. + (1-\rho)\mathbb{E}_{\xi,\eta}\left[\frac{\xi(\mathrm{prox}_x^- - \sqrt{q}\sqrt{\Delta}\xi)}{\sqrt{q}\sqrt{\Delta}V} - \sigma(\sqrt{1-\Delta}\mathrm{prox}_y^- + \mathrm{prox}_x^-)^2\right]\right] \\[4pt] \hat{q}_+ = \alpha\rho\frac{1}{V_+^2}\mathbb{E}_{\xi,\eta}\left(\mathrm{prox}_y^+ - \sqrt{q_+}\eta - m\right)^2 \\[4pt] \hat{q}_- = \alpha(1-\rho)\frac{1}{V_-^2}\mathbb{E}_{\xi,\eta}\left(\mathrm{prox}_y^- - \sqrt{q_-}\eta + m\right)^2 \\[4pt] \hat{V}_+ = -\alpha\rho\frac{1}{\sqrt{q_+}V_+}\mathbb{E}_{\xi,\eta}\left(\mathrm{prox}_y^+ - \sqrt{q_+}\eta - m\right)\eta \\[4pt] \hat{V}_- = -\alpha(1-\rho)\frac{1}{\sqrt{q_-}V_-}\mathbb{E}_{\xi,\eta}\left(\mathrm{prox}_y^- - \sqrt{q_-}\eta + m\right)\eta \end{cases} \tag{70}$$

These equations are, as previously discussed, asymptotically exact as $d \to \infty$. While they still involve traces over high-dimensional matrices $\Sigma_k$, this is a very simple operation. We have therefore reduced the original high-dimensional optimization problem (4) to the much simpler one of computing traces like (70). Crucially, while these traces cannot be generally simplified without Assumption 3.1, they provide the benefit of a simple and compact expression which bypasses the need of jointly diagonalizable covariances, and which can thus be readily evaluated for any covariance, like real-data covariances. This versatility is leveraged in Appendix D.

## A.8 End of the derivation: Conjecture 3.3

We finally provide the last step in the derivation of the fully asymptotic characterization of Conjecture 3.3. Assuming $r(\cdot) = \lambda/2\|\cdot\|_F^2$ (Assumption 3.2), the Moreau envelope $\mathcal{M}_r$ assumes a very simple expression, and no longer needs to

be written as a high-dimensional optimization problem. The resulting free energy can be compactly written as:

$$\Phi = \operatorname*{extr}_{q,m,V,\hat{q},\hat{m},\hat{V},\{q_k,m_k,V_k,\hat{q}_k,\hat{m}_k,\hat{V}_k\}_{k=1}^K} \frac{\mathrm{Tr}\left[\hat{V}q\right] - \mathrm{Tr}[\hat{q}V]}{2} + \frac{1}{2}\sum_{k=1}^K (\mathrm{Tr}\left[\hat{V}_k q_k\right] - \mathrm{Tr}[\hat{q}_k V_k]) - \sum_{k=1}^K m_k \hat{m}_k$$

$$- \alpha \sum_{k=1}^K \rho_k \mathbb{E}_{\xi,\eta} \mathcal{M}_k(\xi,\eta)$$

$$+ \frac{1}{2d}\mathrm{Tr}\underbrace{\left[\left(\lambda\mathbb{I}_p \odot \mathbb{I}_d + \hat{V}\otimes\mathbb{I}_d + \sum_{k=1}^K \hat{V}_k \otimes \boldsymbol{\Sigma}_k\right)^{-1} \odot \left(\hat{q}\otimes\mathbb{I}_d + \sum_{k=1}^K \hat{q}_k \otimes \boldsymbol{\Sigma}_k + d\left(\sum_{k=1}^K \hat{m}_k \boldsymbol{\mu}_k^\top\right)^{\otimes 2}\right)\right]}_{(*)} \quad (71)$$

$(*)$ is the only segment which still involves high-dimensional matrices, and is therefore not yet full asymptotic. Assuming Assumption 3.1, $(*)$ can be massaged into

$$(*) = \frac{1}{2d}\sum_{i=1}^d \mathrm{Tr}\left[\left(\lambda + \hat{V} + \lambda_i^k \hat{V}_k\right)^{-1}\left(\hat{q} + \sum_{k=1}^K \lambda_i^k \hat{q}_k + \sum_{1\le k,j\le K} e_i^\top \mu_j \hat{m}_j \hat{m}_k^\top \boldsymbol{\mu}_k^\top e_i\right)\right]$$

$$\stackrel{d\to\infty}{=} \frac{1}{2}\int d\nu(\gamma,\tau)\,\mathrm{Tr}\left[\left(\lambda + \hat{V} + \gamma_k \hat{V}_k\right)^{-1}\left(\hat{q} + \sum_{k=1}^K \gamma_k \hat{q}_k + \sum_{1\le k,j\le K} \tau_j \tau_k \hat{m}_j \hat{m}_k^\top\right)\right] \quad (72)$$

The fully asymptotic free energy thus becomes

$$\Phi = \operatorname*{extr}_{q,m,V,\hat{q},\hat{m},\hat{V},\{q_k,m_k,V_k,\hat{q}_k,\hat{m}_k,\hat{V}_k\}_{k=1}^K} \frac{\mathrm{Tr}\left[\hat{V}q\right] - \mathrm{Tr}[\hat{q}V]}{2} + \frac{1}{2}\sum_{k=1}^K (\mathrm{Tr}\left[\hat{V}_k q_k\right] - \mathrm{Tr}[\hat{q}_k V_k]) - \sum_{k=1}^K m_k \hat{m}_k$$

$$- \alpha \sum_{k=1}^K \rho_k \mathbb{E}_{\xi,\eta} \mathcal{M}_k(\xi,\eta)$$

$$+ \frac{1}{2}\int d\nu(\gamma,\tau)\,\mathrm{Tr}\left[\left(\lambda + \hat{V} + \gamma_k \hat{V}_k\right)^{-1}\left(\hat{q} + \sum_{k=1}^K \gamma_k \hat{q}_k + \sum_{1\le k,j\le K} \tau_j \tau_k \hat{m}_j \hat{m}_k^\top\right)\right] \quad (73)$$

Requiring the free energy to be extremized implies

$$
\begin{cases}
\hat{q}_k = \alpha\rho_k \mathbb{E}_{\xi,\eta} V_k^{-1} \left( \text{prox}_y^k - q_k^{\frac{1}{2}}\eta - m_k \right)^{\otimes 2} V_k^{-1} \\
\hat{V}_k = -\alpha\rho_k q_k^{-\frac{1}{2}} \mathbb{E}_{\xi,\eta} V_k^{-1} \left( \text{prox}_y^k - q_k^{\frac{1}{2}}\eta - m_k \right) \eta^\top \\
\hat{m}_k = \alpha\rho_k \mathbb{E}_{\xi,\eta} V_k^{-1} \left( \text{prox}_y^k - q_k^{\frac{1}{2}}\eta - m_k \right) \\
\hat{q} = \alpha \sum_{k=1}^{K} \rho_k \mathbb{E}_{\xi,\eta} V^{-1} \left( \text{prox}_x^k - q^{\frac{1}{2}}\xi \right)^{\otimes 2} V^{-1} \\
\hat{V} = -\alpha \sum_{k=1}^{K} \rho_k \mathbb{E}_{\xi,\eta} \left[ q^{-\frac{1}{2}} V^{-1} \left( \text{prox}_x^k - q^{\frac{1}{2}}\xi \right) \xi^\top - \sigma \left( \sqrt{1-\Delta}\,\text{prox}_y^k + \text{prox}_x^k \right)^{\otimes 2} \right]
\end{cases}
\tag{74}
$$

$$
\begin{cases}
q_k = \int d\nu(\gamma,\tau)\gamma_k \left( \lambda\mathbb{I}_p + \hat{V} + \sum_{j=1}^{K} \gamma_j \hat{V}_j \right)^{-2} \left( \hat{q} + \sum_{j=1}^{K} \gamma_j \hat{q}_j + \sum_{1 \le j,l \le K} \tau_j \tau_l \hat{m}_j \hat{m}_l^\top \right) \\
V_k = \int d\nu(\gamma,\tau)\gamma_k \left( \lambda\mathbb{I}_p + \hat{V} + \sum_{j=1}^{K} \gamma_j \hat{V}_j \right)^{-1} \\
m_k = \int d\nu(\gamma,\tau)\tau_k \left( \lambda\mathbb{I}_p + \hat{V} + \sum_{j=1}^{K} \gamma_j \hat{V}_j \right)^{-1} \sum_{j=1}^{K} \tau_j \hat{m}_j \\
q = \int d\nu(\gamma,\tau) \left( \lambda\mathbb{I}_p + \hat{V} + \sum_{j=1}^{K} \gamma_j \hat{V}_j \right)^{-2} \left( \hat{q} + \sum_{j=1}^{K} \gamma_j \hat{q}_j + \sum_{1 \le j,l \le K} \tau_j \tau_l \hat{m}_j \hat{m}_l^\top \right) \\
V = \int d\nu(\gamma,\tau) \left( \lambda\mathbb{I}_p + \hat{V} + \sum_{j=1}^{K} \gamma_j \hat{V}_j \right)^{-1}
\end{cases}
,
\tag{75}
$$

which is exactly (13). This closes the derivation of Conjecture 3.3. $\qquad\square$

As a final heuristic remark, observe that from (74) it is reasonable to expect that all the hatted statistics $\hat{q}_k, \hat{V}_k, \hat{m}_k, \hat{q}, \hat{V}$ should be of order $\Theta(\alpha)$ as $\alpha \gg 1$. As a consequence, $V_k = \Theta(1/\alpha)$ and $q, m, q_k, m_k$ should approach their $\alpha \to \infty$ limit as $\Theta(1/\alpha)$. Therefore, the MSE (61) should also approach its $\alpha \to \infty$ limit as $\Theta(1/\alpha)$.

### A.9 Additional comparisons

We close this appendix by providing further discussion on some points, including the role of the non-linearity $\sigma(\cdot)$, the influence of the weight-tying assumption, and of the (an-)isotropy and (homo/hetero)-scedasticity of the clusters.

**The activation function** The figures in the main text were generated using $\sigma = \tanh$. Several studies for RAEs, however, have highlighted that an auto-encoder would optimally seek to place itself in the *linear* region of its non-linearity, so as to learn the principal components of the training data, at least in the non-linear untied weights case, see e.g. [3, 8]. In light of these findings, it is legitimate to wonder whether setting a linear activation $\sigma(x) = x$ would not yield a better MSE. Fig. 6 shows that it is not the case. For the binary isotropic mixture (13), the linear activations yield a worse performance than the tanh activation.

**Weight-tying** We have assumed that the weights of the DAE (2) were tied. While this assumption originates mainly for technical reasons in the derivation, note that it has been also used and discussed in practical setting [3] as a way to prevent the DAE from going to the linear region of its non-linearity, by making the norm of the first layer weights very small, and that of the second layer very large to compensate [8]. A full extension of Conjecture 3.3 to the case of the untied weight is a lengthy theoretical endeavour, which we leave for future work. However, note that Fig. 6 shows, in the binary isotropic case, that untying the weights actually leads to a *worse* MSE than the DAE with tied weights. This is a very interesting observation, as a DAE with untied weights obviously has the expressive power to implement a DAE with tied weights. Therefore, this effect should be mainly due to the untied landscape presenting local minima trapping the optimization. A full landscape analysis would present a very important future research topic. Finally, remark that this stands in sharp contrast to the observation of [8] for non-linear RAEs, where weight-tying worsened the performance, as it prevents the RAE from implementing a linear principal component analysis.

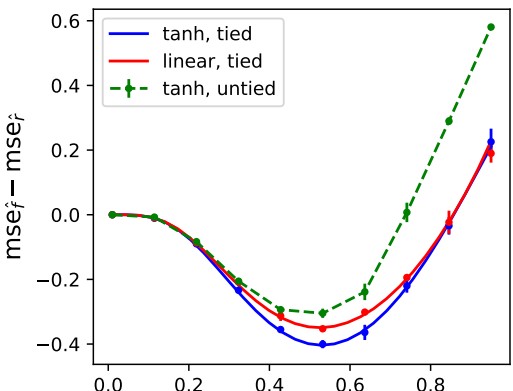

Figure 6: Same parameters as Fig. 1 in the main text. $\alpha = 1, K = 2, \rho_{1,2} = 1/2, \Sigma_{1,2} = 0.3 \times \mathbb{I}_d, p = 1$; the cluster mean $\mu_1 = -\mu_2$ was taken as a random Gaussian vector of norm 1. Difference in MSE between the full DAE $\hat{f}$ (2) and the rescaling network $\hat{r}$ (3) for $\sigma(\cdot) = \tanh$ (blue) or $\sigma(x) = x$ (red). Solid lines correspond to the sharp asymptotic characterization (13), which is a particular case of Conjecture 3.3. Dots represent numerical simulations for $d = 700$, training the DAE using the `Pytorch` implementation of full-batch Adam [43], with learning rate $\eta = 0.05$ over 2000 epochs, averaged over $N = 2$ instances. Error bars represent one standard deviation. In dashed green, numerical simulations for the same architecture, but untied weights. The learning parameters were left unchanged.

An important conclusion from these two observations is that, in contrast to RAEs, the DAE (2) does *not* just implement a linear principal component analysis — weight tying, and the presence of a non-linearity, which are obstacles for RAEs in reaching the PCA MSE, lead for DAEs (2) to a *better* MSE. The model (2) therefore constitutes a model of a genuinely *non-linear* algorithm, where the non-linearity is helpful and is not undone during training. Further discussion can be found in Section 4 of the main text.

**Heteroscedasticity, anisotropy** Fig. 1 and 3 in the main text all represent the isotropic, homoscedastic case. This simple model actually encapsulates all the interesting phenomenology. In this paragraph, we discuss for completeness the generically heteroscedastic, anisotropic case. We consider a binary mixture, with covariances $\Sigma_1, \Sigma_2$ independently drawn from the Wishart-Laguerre ensemble, with aspect ratios $5/7$ and $5/6$. Therefore, the clusters are anisotropic, and eigendirections associated with the largest eigenvalues are more "stretched" (i.e. induce higher cluster variance). Furthermore, since the set of eigenvectors of $\Sigma_1, \Sigma_2$ are independent, the two clusters are stretched in different directions. To ease the comparison with Fig. 1 (isotropic, homoscedastic), these Wishart matrices were further divided by 10, so that the trace is approximately 0.09 like in Fig. 1 – i.e., the clusters have the same average extension. Fig. 7 presents the resulting metrics and summary statistics. Again, the agreement between the theory 3.3 and numerical simulation using `Pytorch` is compelling. Qualitatively, the observations made in Section 4 still hold true in the anisotropic heteroscedastic case:

- The curve of the cosine similarity still displays a non-monotonic behaviour, signalling the preferential activation by the DAE of its skip connection at low $\Delta$ and of its network component at high $\Delta$.

- The skip connection strength $\hat{b}$ is higher at small $\Delta$ and decreases, in connection to the previous remark.

- The norm of the weight matrix $\hat{w}$ overall increases with the noise level $\Delta$, signalling an increasing usage by the DAE of its bottleneck network component, again in accordance to the previous remark.

Therefore, the generic case does not introduce qualitative changes compared to the isotropic case.

**Strength of the weight decay $\lambda$** A $\ell_2$ regularization (weight decay) $\lambda = 0.1$ was adopted in the main text. In fact, the value of the strength of the weight decay was not found to sensibly influence the curves, and again, the qualitative phenomena discussed in Section 4 are observed for any value. Fig. 8 shows, for an isotropic binary mixture, the MSE difference $\mathrm{mse}_{\hat{f}} - \mathrm{mse}_{\hat{r}}$ (5) and the cosine similarity $\theta$ (6) for regularization strength $\lambda = 0.1$ and $\lambda = 0.001$. As can be observed, even reducing the regularization a hundredfold does not change at all the qualitative picture – in particular, the non-monotonicity of the cosine similarity, discussed in Section 4 – , and very little the quantitative values. On the

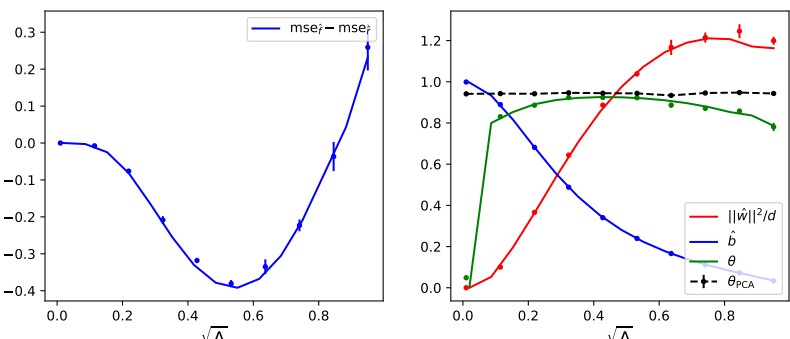

Figure 7: $\alpha = 1, K = 2, \rho_{1,2} = 1/2, p = 1, \lambda = 0.1, \sigma = \tanh$; the cluster mean $\mu_1 = -\mu_2$ was taken as a random Gaussian vector of norm 1. The cluster covariances were independently drawn from the Wishart-Laguerre ensemble, with aspect ratios $5/6$ and $5/7$, before being normalized by 10 to match the trace of the isotropic case in Fig. 1. In particular, the eigenvectors are independent, so the mixture is totally anisotropic with different main variance directions and heteroscedastic. (left) In blue, the difference in MSE between the full DAE $\hat{f}$ (2) and the rescaling network $\hat{r}$ (3). Solid lines correspond to the sharp asymptotic characterization of Conjecture 3.3 in its generic formulation (58). Dots represent numerical simulations for $d = 500$, training the DAE using the `Pytorch` implementation of full-batch Adam [43], with learning rate $\eta = 0.05$ over 2000 epochs, averaged over $N = 5$ instances Error bars represent one standard deviation. (right) Cosine similarity $\theta$ (6) (green), squared weight norm $\|\hat{w}\|^2/d$ (red) and skip connection strength $\hat{b}$ (blue). Solid lines correspond to the formulas (11)(12) and (10) of Conjecture 3.3; dots are numerical simulations. For completeness, the cosine similarity of the first principal component of the train set is plotted in dashed black.

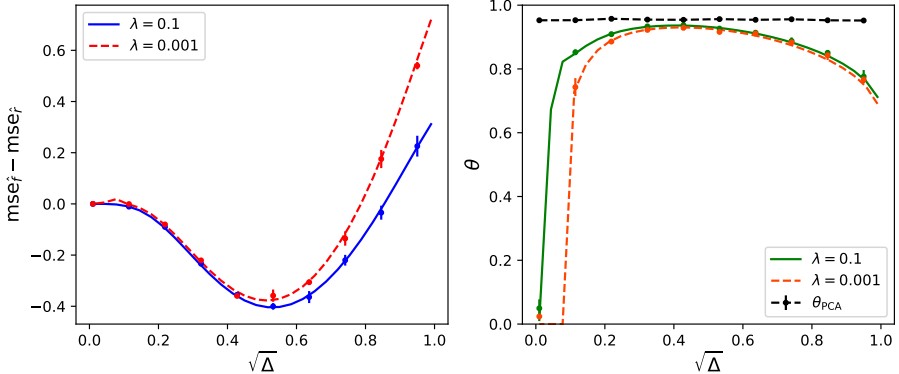

Figure 8: Same parameters as Fig. 1 in the main text. $\alpha = 1, K = 2, \rho_{1,2} = 1/2, \Sigma_{1,2} = 0.09 \times \mathbb{I}_d, p = 1, \sigma = \tanh$; the cluster mean $\mu_1 = -\mu_2$ was taken as a random Gaussian vector of norm 1. Difference in MSE between the full DAE $\hat{f}$ (2) and the rescaling network $\hat{r}$ (3) for $\lambda = 0.1$ (blue) or $\lambda = 0.001$ (red). Lines correspond to the sharp asymptotic characterization (13) of Conjecture 3.3. Dots represent numerical simulations for $d = 700$, training the DAE using the `Pytorch` implementation of full-batch Adam [43], with learning rate $\eta = 0.05$ over 2000 epochs, averaged over $N = 2$ instances. (right) Cosine similarities, for $\lambda = 0.1$ (green) and $\lambda = 0.001$ (orange); lines correspond to the asymptotic formulae of Conjecture 3.3, dots represent the numerical simulations.

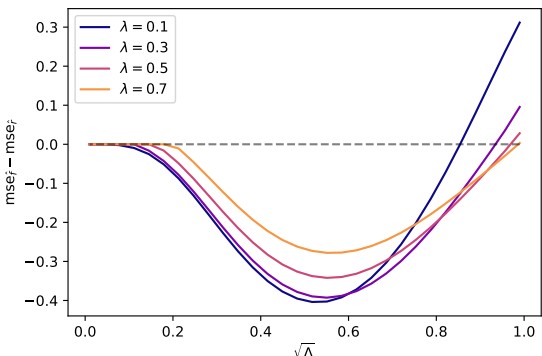

Figure 9: Like Fig. 1 of the main text, $\alpha = 1, K = 2, \rho_{1,2} = \frac{1}{2}, \Sigma_{1,2} = 0.09 \times \mathbb{I}_d, p = 1, \sigma(\cdot) = \tanh(\cdot)$; the cluster mean $\boldsymbol{\mu}_1 = -\boldsymbol{\mu}_2$ was taken as a random Gaussian vector of norm 1. (left) Solid lines correspond to the difference in MSE between the full DAE $\hat{f}$ (2) and the rescaling component $\hat{r}$ (3) as predicted by Conjecture 3.3 Different colors indicate different regularization strength $\lambda$. For large noises $\Delta$ and insufficient regularization $\lambda$, the DAE overfits the train set, leading to performances worse than the simple rescaling $\hat{r}$, as signalled by positive values of $\mathrm{mse}_{\hat{f}} - \mathrm{mse}_{\hat{r}}$. This effect is suppressed for larger regularizations $\lambda$.

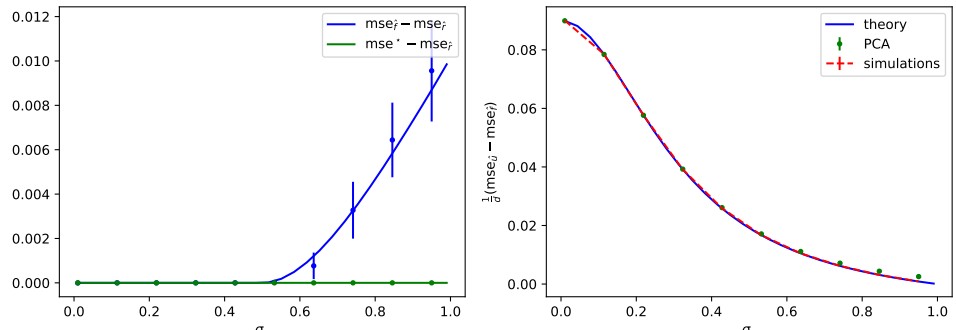

Figure 10: $\alpha = 1, K = 1, \rho_{1,2} = \frac{1}{2}, \Sigma_1 = 0.09 \times \mathbb{I}_d, p = 1, \sigma(\cdot) = \tanh(\cdot)$. (left) Difference between the MSE of the DAE $\hat{f}$ and the MSE of the rescaling $\hat{r}$ $\mathrm{mse}_{\hat{f}} - \mathrm{mse}_{\hat{r}}$ in the case of a single Gaussian cluster. Solid lines correspond to the theoretical prediction of Conjecture 3.3 and dots to numerical simulations. In the case of a single cluster, the bottleneck component of the DAE is not needed, and its presence leads to detrimental overfitting, as signalled by the positive value of $\mathrm{mse}_{\hat{f}} - \mathrm{mse}_{\hat{r}}$. (right) Difference between the MSE of the DAE $\hat{f}$ and the MSE of the bottleneck network $\hat{u}$, divided by $d$. Like the $K > 1$ case (see Fig. 3 (b)), this difference is of order $\Theta_d(d)$, and the MSE of PCA (green dots) is sensibly equal.

other hand, observe that at the level of the MSE increasing the weight decay helps palliate the overfitting at large noise levels $\Delta$, as we further illustrate in Fig. 9

$K = 1$ **cluster** We finally discuss the case of *unstructured* data, where the Gaussian mixture only has a single cluster. Analytically, this corresponds to setting the norm of the cluster mean $\|\boldsymbol{\mu}\|$ to zero in Conjecture 3.3. The bottleneck component, whose role is to learn the data structure – as given by the cluster mean–and leverage it to improve the denoising performance, is no longer needed when the data is unstructured. It presence actually leads the DAE to overfit, as can be observed in Fig.10 (left). Similarly, the PCA denoiser performs worse in the unstructured case, with an associated MSE remaining of order $\Theta(d)$, see Fig.10 (right).

## B Tweedie baseline

### B.1 Oracle denoiser

Tweedie's formula [42] provides the best estimator of a clean sample $x$ given a noisy sample $\tilde{x}$ corrupted by Gaussian noise, as

$$t(\tilde{\boldsymbol{x}}) = \frac{\tilde{\boldsymbol{x}}}{\sqrt{1-\Delta}} + \frac{\Delta}{\sqrt{1-\Delta}}\boldsymbol{\nabla}\mathbb{P}|_{\tilde{\boldsymbol{x}}}. \tag{76}$$

Note that this gives the Bayes-optimal estimator for the MSE, assuming *perfect knowledge* of the clean data distribution $\mathbb{P}$ from which the so-called score $\boldsymbol{\nabla}\mathbb{P}$ has to be computed. Of course, this knowledge is inaccessible in general, where the only information on $\mathbb{P}$ is provided in the form of the train set $\mathcal{D}$. Therefore, Tweedie's formula does not give a learning algorithm, but allows to give an oracle lower-bound on the achievable MSEs, as any learning algorithm will yield a higher MSE than this information-theoretic baseline.

For a generic Gaussian mixture (1), Tweedie's formula reduces to the expression

$$t(\boldsymbol{x}) = \frac{1}{\sqrt{1-\Delta}} \frac{\sum\limits_{k=1}^{K} \rho_k \left[ \tilde{\boldsymbol{\Sigma}}_k(\tilde{\boldsymbol{\Sigma}}_k + \Delta\mathbb{I}_d)^{-1}\boldsymbol{x} + (\tilde{\boldsymbol{\Sigma}}_k + \Delta\mathbb{I}_d)^{-1}\tilde{\boldsymbol{\mu}}_k \right] \mathcal{N}\left(\boldsymbol{x}; \tilde{\boldsymbol{\mu}}_k, \tilde{\boldsymbol{\Sigma}}_k + \Delta\mathbb{I}_d\right)}{\sum\limits_{k=1}^{K} \rho_k \mathcal{N}\left(\boldsymbol{x}; \tilde{\boldsymbol{\mu}}_k, \tilde{\boldsymbol{\Sigma}}_k + \Delta\mathbb{I}_d\right)}, \tag{77}$$

where we noted

$$\tilde{\boldsymbol{\mu}}_k = \sqrt{1-\Delta}\boldsymbol{\mu}_k, \qquad\qquad\qquad \tilde{\boldsymbol{\Sigma}}_k = (1-\Delta)\boldsymbol{\Sigma}_k. \tag{78}$$

### B.2 Oracle test MSE

In general, except in the complete homoscedastic case where all clusters have the same covariance, there is no closed-form asymptotic expression for the MSE achieved by the Tweedie denoiser. In the binary homoscedastic case $\boldsymbol{\mu}_1 = -\boldsymbol{\mu}_2 \equiv \boldsymbol{\mu}$, $\boldsymbol{\Sigma}_1 = \boldsymbol{\Sigma}_2 = \sigma^2\mathbb{I}_d$ (see Fig. 1), Tweedie's formula (77) reduces to the compact form

$$t(\boldsymbol{x}) = \frac{\sqrt{1-\Delta}\sigma^2}{\sigma^2(1-\Delta)+\Delta}\tilde{\boldsymbol{x}} + \frac{\Delta}{\sigma^2(1-\Delta)+\Delta} \tanh\left(\frac{\tilde{\boldsymbol{x}}^\top\boldsymbol{\mu}\sqrt{1-\Delta}}{\sigma^2(1-\Delta)+\Delta}\right) \times \boldsymbol{\mu} \tag{79}$$

Note that this is of the same form as the DAE architecture (2). The associated MSE reads

$$\mathrm{mse}^\star = \mathbb{E}_{\boldsymbol{x},\boldsymbol{\xi}} \left\| \boldsymbol{x} - \left[ b \times (\sqrt{1-\Delta}\boldsymbol{x} + \boldsymbol{\xi}) + \frac{\Delta\boldsymbol{\mu}}{\sigma^2(1-\Delta)+\Delta}\sigma\left(\frac{\sqrt{1-\Delta}\boldsymbol{\mu}^\top(\sqrt{1-\Delta}\boldsymbol{x}+\boldsymbol{\xi})}{\sigma^2(1-\Delta)+\Delta}\right)\right] \right\|^2. \tag{80}$$

A sharp asymptotic formula can be found to be

$$\mathrm{mse}^\star = \mathrm{mse}_\circ + \frac{\sigma_e^4}{(\sigma^2(1-\Delta)+\Delta)^2}\frac{1}{2}\sum_{s=\pm1}\mathbb{E}_z^{\mathcal{N}(0,1)}\left[\sigma\left(\frac{(1-\Delta)s+\sqrt{1-\Delta}z\times\sqrt{\Delta+(1-\Delta)\sigma^2}}{\sigma^2(1-\Delta)+\Delta}\right)^2\right]$$

$$- \frac{2\Delta}{\sigma^2(1-\Delta)+\Delta}\frac{1}{2}\sum_{s=\pm1}\mathbb{E}_z^{\mathcal{N}(0,1)}\left[\sigma\left(\frac{(1-\Delta)s+\sqrt{1-\Delta}z\times\sqrt{\Delta+(1-\Delta)\sigma^2}}{\sigma^2(1-\Delta)+\Delta}\right)\right] \times \left((1-b\sqrt{1-\Delta})s\right). \tag{81}$$

This is the theoretical characterization plotted in Figs. 1 and 3 in the main text.

As discussed in the main text and illustrated in Fig. 3(left), the MSE of the DAE approaches the oracle MSE (81) as the sample complexity $\alpha$ grows. On the other hand, the DAE MSE does not exactly converges to the oracle MSE as $\alpha \to \infty$. Intuitively, this is because of the fact that while the form of the oracle (79) is that of a DAE, it does not have tied weights, like the considered architecture (2). The latter cannot therefore perfectly realize the oracle denoiser, whence a small discrepancy, as more precisely numerically characterized in Fig. 11.

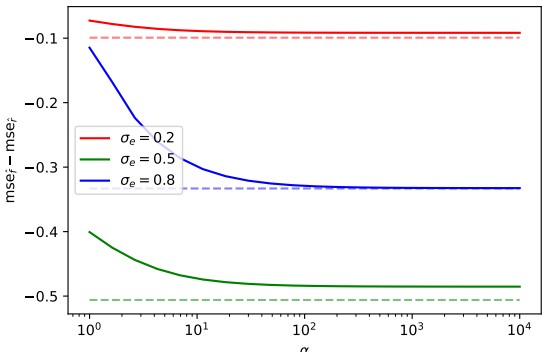

Figure 11: Like Fig. 1 of the main text, $K = 2, \rho_{1,2} = \frac{1}{2}, \Sigma_{1,2} = 0.09 \times \mathbb{I}_d, p = 1, \sigma(\cdot) = \tanh(\cdot)$; the cluster mean $\boldsymbol{\mu}_1 = -\boldsymbol{\mu}_2$ was taken as a random Gaussian vector of norm 1. Solid lines correspond to the difference in MSE between the full DAE $\hat{f}$ and the rescaling component $\hat{r}$ as predicted by Conjecture 3.3. In dashed lines, the oracle MSE (81), minus the rescaling MSE $\text{mse}_{\hat{r}}$. Different colors correspond to different noise levels $\Delta$. As the sample complexity $\alpha$ becomes large, the denoiser MSE approaches the oracle baseline, but does not become exactly equal to it as $\alpha \to \infty$.

## C   Bayes-optimal MSE

### C.1   Assuming imperfect prior knowledge

The Tweedie denoiser discussed in Appendix B is an *oracle* denoiser, in the sense that perfect knowledge of all the parameters $\{\boldsymbol{\mu}_k, \Sigma_k\}_{k=1}^K$ of the Gaussian mixture distribution $\mathbb{P}$ (1) is assumed. Therefore, though the comparison of the DAE (2) MSE with the oracle Tweedie MSE $\text{mse}^\star$ does provide useful intuition (see Fig. 3), it importantly does not allow to disentangle which part of the MSE of the DAE is due to the limited expressivity of the architecture, and which is due to the limited availablility of training samples – which entails *imperfect* knowledge of the parameters $\{\boldsymbol{\mu}_k, \Sigma_k\}_{k=1}^K$ of the Gaussian mixture distribution $\mathbb{P}$ (1). In this appendix, we derive the MSE of the Bayes optimal estimator (with respect to the MSE), assuming only knowledge of the distribution from which the parameters $\{\boldsymbol{\mu}_k, \Sigma_k\}_{k=1}^K$ are drawn. For simplicity and conciseness, we limit ourselves to the binary isotropic and homoscedastic mixture (see e.g. Fig. 1), for which $\boldsymbol{\mu}_1 = -\boldsymbol{\mu}_2 \equiv \boldsymbol{\mu}^\star$ and $\Sigma_1 = \Sigma_2 = \sigma\mathbb{I}_d$. For definiteness, we assume $\boldsymbol{\mu} \sim \mathbb{P}_\mu$, with $\mathbb{P}_\mu = \mathcal{N}(0, \mathbb{I}_d/d)$, so that with high probability $\|\boldsymbol{\mu}\| = 1$. We shall moreover assume for ease of discussion that $\sigma$ is perfectly known. Thus, the centroid $\boldsymbol{\mu}$ is the only unknown parameter.

### C.2   Bayes-optimal denoiser without knowledge of cluster means

The corresponding Bayes-optimal denoiser then reads

$$\mathfrak{b}(\tilde{\boldsymbol{x}}) \equiv \mathbb{E}\left[\boldsymbol{x}|\tilde{\boldsymbol{x}}, \sigma, \mathcal{D}\right] = \int d\boldsymbol{\mu} \underbrace{\mathbb{E}\left[\boldsymbol{x}|\tilde{\boldsymbol{x}}, \boldsymbol{\mu}, \sigma, \mathcal{D}\right]}_{t(\boldsymbol{\mu}, \tilde{\boldsymbol{x}})} \mathbb{P}(\boldsymbol{\mu}|\sigma, \mathcal{D}) \tag{82}$$

Where we have identified the oracle denoiser (79), and emphasized the dependence on $\boldsymbol{\mu}$. Note that this is a slight abuse of notation, as the true oracle $t(\boldsymbol{\mu}^\star, \tilde{\boldsymbol{x}})$ involves the ground truth centroid. Further note that $\mathbb{P}(\boldsymbol{\mu}|\sigma, \mathcal{D}) = \mathbb{P}(\boldsymbol{\mu}|\mathcal{D})$, since knowledge of $\sigma$ does not bring information about $\boldsymbol{\mu}$. Furthermore, note that as the noisy part of the dataset $\tilde{\mathcal{D}} \equiv \{\tilde{\boldsymbol{x}}^\mu\}_{\mu=1}^n$ brings only redundant information, one further has that

$$\mathbb{P}(\boldsymbol{\mu}|\mathcal{D}) = \mathbb{P}(\boldsymbol{\mu}|\check{\mathcal{D}}), \tag{83}$$

where we noted $\check{\mathcal{D}} = \{\boldsymbol{x}^\mu\}_{\mu=1}^n$ the clean part of the dataset. Thus

$$
\begin{aligned}
\mathfrak{b}(\tilde{\boldsymbol{x}}) &= \frac{1}{\mathbb{P}(\check{\mathcal{D}})} \int d\boldsymbol{\mu}\ t(\boldsymbol{\mu}, \tilde{\boldsymbol{x}}) \mathbb{P}(\check{\mathcal{D}}|\boldsymbol{\mu}) \mathbb{P}_\mu(\boldsymbol{\mu}) \\
&= \frac{1}{\mathbb{P}(\check{\mathcal{D}})} \int d\boldsymbol{\mu}\ t(\boldsymbol{\mu}, \tilde{\boldsymbol{x}}) \mathbb{P}_\mu(\boldsymbol{\mu}) \prod_{\mu=1}^n \frac{1}{2} \frac{1}{(2\pi\sigma^2)^{d/2}} \left[ e^{-\frac{1}{2\sigma^2}\|\boldsymbol{x}^\mu - \boldsymbol{\mu}\|^2} + e^{-\frac{1}{2\sigma^2}\|\boldsymbol{x}^\mu + \boldsymbol{\mu}\|^2} \right] \\
&= \frac{1}{\mathbb{P}(\check{\mathcal{D}})} \int d\boldsymbol{\mu}\ t(\boldsymbol{\mu}, \tilde{\boldsymbol{x}}) \frac{1}{(2\pi/d)^{d/2}} e^{-\frac{d}{2}\|\boldsymbol{\mu}\|^2} e^{-\frac{n}{2\sigma^2}\|\boldsymbol{\mu}\|^2} \prod_{\mu=1}^n \frac{e^{-\frac{1}{2\sigma^2}\|\boldsymbol{x}^\mu\|^2}}{(2\pi\sigma^2)^{d/2}} \cosh\left(\frac{\boldsymbol{\mu}^\top \boldsymbol{x}^\mu}{\sigma^2}\right) \\
&= \frac{1}{\sqrt{d}\mathbb{P}(\check{\mathcal{D}})} \int d\boldsymbol{\mu}\ t(\boldsymbol{\mu}/\sqrt{d}, \tilde{\boldsymbol{x}}) \frac{1}{(2\pi/d)^{d/2}} e^{-\frac{1}{2}\|\boldsymbol{\mu}\|^2} e^{-\frac{\alpha}{2\sigma^2}\|\boldsymbol{\mu}\|^2} \prod_{\mu=1}^n \frac{e^{-\frac{1}{2\sigma^2}\|\boldsymbol{x}^\mu\|^2}}{(2\pi\sigma^2)^{d/2}} \cosh\left(\frac{\boldsymbol{\mu}^\top \boldsymbol{x}^\mu}{\sigma^2\sqrt{d}}\right) \\
&= \frac{1}{Z} \int d\boldsymbol{\mu}\ t(\boldsymbol{\mu}/\sqrt{d}, \tilde{\boldsymbol{x}}) \mathbb{P}_b(\boldsymbol{\mu}),
\end{aligned}
\tag{84}
$$

where we noted

$$
\mathbb{P}_b(\boldsymbol{\mu}) \equiv e^{-\frac{1}{2\hat{\sigma}^2}\|\boldsymbol{\mu}\|^2} \prod_{\mu=1}^n \cosh\left(\frac{\boldsymbol{\mu}^\top \boldsymbol{x}^\mu}{\sigma^2\sqrt{d}}\right),
\tag{85}
$$

with the effective variance

$$
\hat{\sigma}^2 \equiv \frac{\sigma^2}{\sigma^2 + \alpha}.
\tag{86}
$$

Finally, the partition function $Z$ is defined as the normalisation of $\mathbb{P}_b$, i.e.

$$
Z \equiv \int d\boldsymbol{\mu}\ \mathbb{P}_b(\boldsymbol{\mu}).
\tag{87}
$$

One is now in a position to compute the MSE associated with the Bayes estimator $\mathfrak{b}(\tilde{\boldsymbol{x}})$:

$$
\begin{aligned}
&\mathrm{mse}_\mathfrak{b} \\
&= \mathbb{E}_\mathcal{D} \mathbb{E}_{\boldsymbol{x} \sim \mathcal{P}} \mathbb{E}_{\boldsymbol{\xi} \sim \mathcal{N}(0, \mathbb{I}_d)} \left\| \boldsymbol{x} - \left\langle t(\boldsymbol{\mu}/\sqrt{d}, \sqrt{1-\Delta}\boldsymbol{x} + \sqrt{\Delta}\boldsymbol{\xi}) \right\rangle_{\boldsymbol{\mu} \sim \mathbb{P}_b} \right\|^2 \\
&= \mathrm{mse}_\circ - \mathbb{E}_\mathcal{D} \sum_{s=\pm 1} \mathbb{E}_{\boldsymbol{\xi}, \boldsymbol{\eta} \sim \mathcal{N}(0, \mathbb{I}_d)} \frac{\Delta}{\sigma^2(1-\Delta) + \Delta} \left\langle \left( \frac{(1-b\sqrt{1-\Delta})s\boldsymbol{\mu}^\top \boldsymbol{\mu}^\star}{d} + \frac{\boldsymbol{\mu}^\top((1-b\sqrt{1-\Delta})\sigma\boldsymbol{\eta} - b\sqrt{\Delta}\boldsymbol{\xi})}{\sqrt{d}} \right) \right. \\
&\qquad \left. \tanh\left( \frac{\sqrt{1-\Delta}\frac{s\boldsymbol{\mu}^\top \boldsymbol{\mu}^\star}{2} + \sqrt{1-\Delta}\sigma\frac{\boldsymbol{\mu}^\top \boldsymbol{\eta}}{\sqrt{d}} + \sqrt{\Delta}\frac{\boldsymbol{\mu}^\top \boldsymbol{\xi}}{\sqrt{d}}}{\sigma^2(1-\Delta) + \Delta} \right) \right\rangle_{\boldsymbol{\mu} \sim \mathbb{P}_b} \\
&\quad + \mathbb{E}_\mathcal{D} \frac{1}{2} \sum_{s=\pm 1} \mathbb{E}_{\boldsymbol{\xi}, \boldsymbol{\eta} \sim \mathcal{N}(0, \mathbb{I}_d)} \left\langle \left( \frac{\Delta}{\sigma^2(1-\Delta) + \Delta} \right)^2 \frac{\boldsymbol{\mu}_1^\top \boldsymbol{\mu}_2}{d} \tanh\left( \frac{\sqrt{1-\Delta}\frac{s\boldsymbol{\mu}_1^\top \boldsymbol{\mu}^\star}{d} + \sqrt{1-\Delta}\sigma\frac{\boldsymbol{\mu}_1^\top \boldsymbol{\eta}}{\sqrt{d}} + \sqrt{\Delta}\frac{\boldsymbol{\mu}_1^\top \boldsymbol{\xi}}{\sqrt{d}}}{\sigma^2(1-\Delta) + \Delta} \right) \right. \\
&\qquad \left. \times \tanh\left( \frac{\sqrt{1-\Delta}\frac{s\boldsymbol{\mu}_2^\top \boldsymbol{\mu}^\star}{d} + \sqrt{1-\Delta}\sigma\frac{\boldsymbol{\mu}_2^\top \boldsymbol{\eta}}{\sqrt{d}} + \sqrt{\Delta}\frac{\boldsymbol{\mu}_2^\top \boldsymbol{\xi}}{\sqrt{d}}}{\sigma^2(1-\Delta) + \Delta} \right) \right\rangle_{\boldsymbol{\mu}_1, \boldsymbol{\mu}_2 \sim \mathbb{P}_b} \\
&= \mathrm{mse}_\circ - \mathbb{E}_\mathcal{D} \sum_{s=\pm 1} \mathbb{E}_{z \sim \mathcal{N}(0,1)} \frac{\Delta}{\sigma^2(1-\Delta) + \Delta} \left\langle \left( \frac{(1-b\sqrt{1-\Delta})s\boldsymbol{\mu}^\top \boldsymbol{\mu}^\star}{d} \right) \tanh\left( \frac{\sqrt{1-\Delta}\frac{s\boldsymbol{\mu}^\top \boldsymbol{\mu}^\star}{d} + \sqrt{(1-\Delta)\sigma^2 + \Delta}\frac{\|\boldsymbol{\mu}\|}{\sqrt{d}}z}{\sigma^2(1-\Delta) + \Delta} \right) \right\rangle_{\boldsymbol{\mu} \sim \mathbb{P}_b} \\
&\quad + \mathbb{E}_\mathcal{D} \frac{1}{2} \sum_{s=\pm 1} \mathbb{E}_{u, v \sim \mathcal{N}(0, \Omega)} \left\langle \left( \frac{\Delta}{\sigma^2(1-\Delta) + \Delta} \right)^2 \frac{\boldsymbol{\mu}_1^\top \boldsymbol{\mu}_2}{d} \tanh\left( \frac{\sqrt{1-\Delta}\frac{s\boldsymbol{\mu}_1^\top \boldsymbol{\mu}^\star}{d} + \sqrt{(1-\Delta)\sigma^2 + \Delta}u}{\sigma^2(1-\Delta) + \Delta} \right) \right. \\
&\qquad \left. \times \tanh\left( \frac{\sqrt{1-\Delta}\frac{s\boldsymbol{\mu}_2^\top \boldsymbol{\mu}^\star}{d} + \sqrt{(1-\Delta)\sigma^2 + \Delta}v}{\sigma^2(1-\Delta) + \Delta} \right) \right\rangle_{\boldsymbol{\mu}_1, \boldsymbol{\mu}_2 \sim \mathbb{P}_b}.
\end{aligned}
\tag{88}
$$

We have adopted the shortcuts

$$b = \frac{\sqrt{1-\Delta}\sigma^2}{\sigma^2(1-\Delta)+\Delta}, \qquad\qquad \Omega = \begin{bmatrix} \frac{\|\boldsymbol{\mu}_1\|^2}{d} & \frac{\boldsymbol{\mu}_1^\top\boldsymbol{\mu}_2}{d} \\ \frac{\boldsymbol{\mu}_1^\top\boldsymbol{\mu}_2}{d} & \frac{\|\boldsymbol{\mu}_2\|^2}{d}. \end{bmatrix}. \qquad (89)$$

(88) shows that the inner averages involved in the Bayes MSE $\mathrm{mse}_b$ only depend on the overlaps $\boldsymbol{\mu}_{1,2}^\top\boldsymbol{\mu}_{1,2}/d$ for $\boldsymbol{\mu}_1, \boldsymbol{\mu}_2$ two independently sampled vectors from $\mathbb{P}_b$. Motivated by similar high-dimensional studies, it is reasonable to expect these quantities to concentrate as $n, d \to \infty$ in the measure $\mathbb{E}_\mathcal{D}\langle\cdot\rangle_{\mathbb{P}_b}$. A more principle – but much more painstaking– way to derive this assumption is to introduce a Fourier representation, and carry out the $\mathbb{E}_\mathcal{D}$ average using the replica method. We refer the interested reader to, for instance, Appendix H of [36], where such a route is taken.

### C.3 Derivation

To compute the summary statistics of $\mathbb{P}_b$, we again resort to the replica method to compute the moment-generating function (free entropy), see also Appendix A. The replicated partition function reads

$$\mathbb{E}_\mathcal{D}Z^s = \int \prod_{a=1}^s d\boldsymbol{\mu}_a e^{-\frac{1}{2\hat{\sigma}^2}\|\boldsymbol{\mu}_a\|^2} \prod_{\mu=1}^n \sum_{s=\pm 1} \frac{1}{2}\mathbb{E}_{\boldsymbol{\eta}\sim\mathcal{N}(0,\mathbb{I}_d)}\left[\prod_{a=1}^s \cosh\left(\frac{s\frac{\boldsymbol{\mu}_a^\top\boldsymbol{\mu}^\star}{d} + \sigma\frac{\boldsymbol{\mu}_a^\top\boldsymbol{\eta}}{\sqrt{d}}}{\sigma^2}\right)\right]. \qquad (90)$$

Introducing the order parameters

$$q_{ab} \equiv \frac{\boldsymbol{\mu}_a^\top\boldsymbol{\mu}_b}{d}, \qquad\qquad m_a \equiv \frac{\boldsymbol{\mu}_a^\top\boldsymbol{\mu}^\star}{d}, \qquad (91)$$

one reaches

$$\mathbb{E}_\mathcal{D}Z^s = \int \prod_{a\leq b} dq_{ab}d\hat{q}_{ab} \prod_{a=1}^s dm_a d\hat{m}_a \underbrace{e^{-d\sum_{a\leq b} q_{ab}\hat{q}_{ab} - d\sum_a m_a\hat{m}_a}}_{e^{sd\Psi_t}} \underbrace{\int \prod_{a=1}^s d\boldsymbol{\mu}_a e^{-\frac{1}{2\hat{\sigma}^2}\|\boldsymbol{\mu}_a\|^2} e^{\sum_{a\leq b}\hat{q}_{ab}\boldsymbol{\mu}_a^\top\boldsymbol{\mu}_b + \sum_a \boldsymbol{\mu}_a^\top\boldsymbol{\mu}^\star}}_{e^{sd\Psi_w}}$$

$$\underbrace{\left[n \sum_{s=\pm 1} \frac{1}{2} \int \mathcal{N}\left(\{\lambda_a\}_{a=1}^s; 0, (q_{ab})_{1\leq a,b\leq s}\right) \left[\prod_{a=1}^s \cosh\left(\frac{sm_a + \sigma\lambda_a}{\sigma^2}\right)\right]\right]^n}_{e^{s\alpha d\Psi_y}}. \qquad (92)$$

The computation of $\Psi_t, \Psi_w, \Psi_y$ is rather standard and follows the main steps presented in Appendix A. One again assumes the replica symmetric ansatz [21, 22]

$$q_{ab} = (r-q)\delta_{ab} + q, \qquad\qquad m_a = m, \qquad (93)$$

$$\hat{q}_{ab} = \left(-\frac{\hat{r}}{2} + -\hat{q}\right)\delta_{ab} + \hat{q}, \qquad\qquad \hat{m}_a = \hat{m}. \qquad (94)$$

Introducing as in Appendix A the variances

$$V \equiv r - q, \qquad\qquad \hat{V} \equiv \hat{r} + \hat{q}, \qquad (95)$$

one reaches

$$\Psi_t = \frac{\hat{V}q - V\hat{q}}{2} - m\hat{m}, \qquad (96)$$

$$\Psi_w = -\frac{1}{2}\ln\left(1 + \hat{V}\hat{\sigma}^2\right) + \frac{\hat{\sigma}^2}{2}\frac{\hat{q} + \hat{m}^2}{1 + \hat{V}\hat{\sigma}^2}, \qquad (97)$$

$$\Psi_y = \mathbb{E}_{\eta\sim\mathcal{N}(0,1)}\ln\left[\int d\lambda \mathcal{N}(\lambda; \sqrt{q}\eta, V)\cosh\left(\frac{\sigma\lambda + m}{\sigma^2}\right)\right]. \qquad (98)$$

Therefore the free entropy reads

$$\Phi = \mathop{\mathrm{extr}}_{q,m,V,\hat{q},\hat{m},\hat{V}} \frac{\hat{V}q - V\hat{q}}{2} - m\hat{m} - \frac{1}{2}\ln\left(1 + \hat{V}\hat{\sigma}^2\right) + \frac{\hat{\sigma}^2}{2}\frac{\hat{q} + \hat{m}^2}{1 + \hat{V}\hat{\sigma}^2}$$

$$+ \alpha\mathbb{E}_{\eta\sim\mathcal{N}(0,1)}\ln\left[\int d\lambda \mathcal{N}(\lambda; \sqrt{q}\eta, V)\cosh\left(\frac{\sigma\lambda + m}{\sigma^2}\right)\right]. \qquad (99)$$

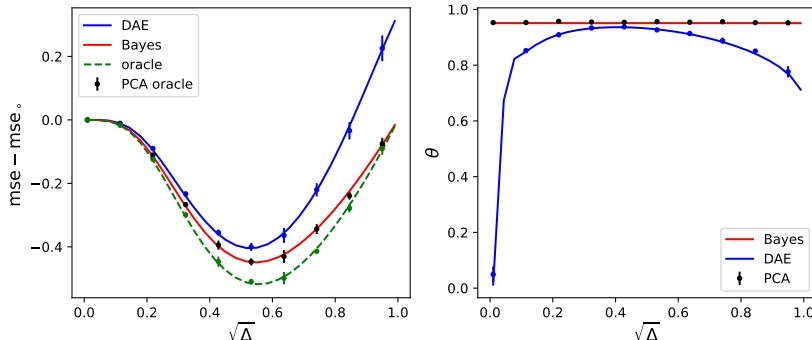

Figure 12: (Binary isotropic mixture, $\boldsymbol{\Sigma}_{1,2} = 0.09\mathbb{I}_d$, $\rho_{1,2} = 1/2$, and $\boldsymbol{\mu}_1 = -\boldsymbol{\mu}_2$, see also Fig. 1). (right) MSE (minus the rescaling baseline $\mathrm{mse}_\circ$) for the DAE ($p = 1, \sigma = \tanh$) (2) (blue), the oracle Tweedie denoiser (76) (green), the Bayes denoiser (101) (red), and the plug-in oracle-PCA denoiser (103) (black). Solid lines correspond to the theoretical formulae 3.3, (81) and (101); simulations correspond to a DAE optimized using the Pytorch implementation of Adam (weight decay $\lambda = 0.1$, learning rate $\eta = 0.05$, full batch, 2000 epochs, sample complexity $\alpha = 1$), averaged over $N = 10$ instances, with error bars representing one standard deviation. (right) Cosine similarity (6) for the same algorithms, with identical color code.

The solution of this extremization problem is given by the set of self-consistent equations, imposing that gradients with respect to all parameters be zero:

$$
\begin{cases}
V = \frac{\hat{\sigma}^2}{1+\hat{V}\hat{\sigma}^2} \\
q = \frac{\hat{\sigma}^4(\hat{q}+\hat{m}^2)}{(1+\hat{V}\hat{\sigma}^2)^2} + \frac{\hat{\sigma}^2}{1+\hat{V}\hat{\sigma}^2} \\
m = \frac{\hat{\sigma}^2\hat{m}}{1+\hat{V}\hat{\sigma}^2}
\end{cases}
\qquad
\begin{cases}
\hat{V} = -\alpha\frac{1}{\sigma\sqrt{q}}\mathbb{E}_{\eta\sim\mathcal{N}(0,1)}\left[\tanh\left(\frac{\sigma\sqrt{q}\eta+m}{\sigma^2}\right)\eta\right] \\
\hat{q} = \frac{\alpha V}{\sigma^2} \\
\hat{m} = \frac{\alpha}{\sigma^2}\mathbb{E}_{\eta\sim\mathcal{N}(0,1)}\left[\tanh\left(\frac{\sigma\sqrt{q}\eta+m}{\sigma^2}\right)\right]
\end{cases}
\tag{100}
$$

This is a simple optimization problem over 6 variables, which can be solved numerically. One can finally use the obtained summary statistics to evaluate the Bayes optimal MSE:

$$
\mathrm{mse}_\mathfrak{b} = \mathrm{mse}_\circ - 2\mathbb{E}_{\mathcal{D}}\mathbb{E}_{z\sim\mathcal{N}(0,1)}\frac{\Delta}{\sigma^2(1-\Delta)+\Delta}\left((1-b\sqrt{1-\Delta})m\right)\tanh\left(\frac{\sqrt{1-\Delta}m+\sqrt{(1-\Delta)\sigma^2+\Delta}\sqrt{q+V}z}{\sigma^2(1-\Delta)+\Delta}\right)
$$

$$
+ \mathbb{E}_{\mathcal{D}}\mathbb{E}_{u,v\sim\mathcal{N}(0,\Omega)}\left(\frac{\Delta}{\sigma^2(1-\Delta)+\Delta}\right)^2 q\tanh\left(\frac{\sqrt{1-\Delta}m+\sqrt{(1-\Delta)\sigma^2+\Delta}u}{\sigma^2(1-\Delta)+\Delta}\right)\tanh\left(\frac{\sqrt{1-\Delta}m+\sqrt{(1-\Delta)\sigma^2+\Delta}v}{\sigma^2(1-\Delta)+\Delta}\right)
\tag{101}
$$

where

$$
\Omega = \begin{bmatrix} q+V & q \\ q & q+V \end{bmatrix}
\tag{102}
$$

This completes the derivation of the MSE of the Bayes estimator agnostic of the cluster means.

### C.4 A simple plug-in denoiser

Another simple denoiser assuming only perfect knowledge of the variance $\sigma$, but imperfect knowledge of $\boldsymbol{\mu}$, is simply to plug the PCA estimate $\hat{\boldsymbol{\mu}}_{\mathrm{PCA}}$ of $\boldsymbol{\mu}$ into the Tweedie oracle (76), i.e.

$$
t(\hat{\boldsymbol{\mu}}_{\mathrm{PCA}}, \tilde{\boldsymbol{x}}) = \frac{\sqrt{1-\Delta}\sigma^2}{\sigma^2(1-\Delta)+\Delta}\tilde{\boldsymbol{x}} + \frac{\Delta}{\sigma^2(1-\Delta)+\Delta}\tanh\left(\frac{\tilde{\boldsymbol{x}}^\top\hat{\boldsymbol{\mu}}_{\mathrm{PCA}}\sqrt{1-\Delta}}{\sigma^2(1-\Delta)+\Delta}\right) \times \hat{\boldsymbol{\mu}}_{\mathrm{PCA}}.
\tag{103}
$$

The performance of this plug-in denoiser is plotted in Fig. 12, and contrasted to the one of the Bayes denoiser $\mathfrak{b}(\cdot)$ (101), the oracle $t(\cdot)$ (81) and the DAE 3.3. Strikingly, the performance of the PCA plug-in denoiser is sizeably identical to the one of the Bayes-optimal denoiser, both in terms of MSE and cosine similarity. It is important to remind at this point that both the plug-in (103) and the Bayes denoiser (101) still rely on perfect knowledge of the cluster variance $\sigma$ and the noise level $\Delta$, while the DAE (2) is agnostic to these parameters. Yet, these two denoisers make for a fairer comparison than the oracle (81), as they importantly take into account the finite training set. The DAE is relatively close to the Bayes baseline for small noise levels $\Delta$, while a gap can be seen to open up for larger $\Delta$.

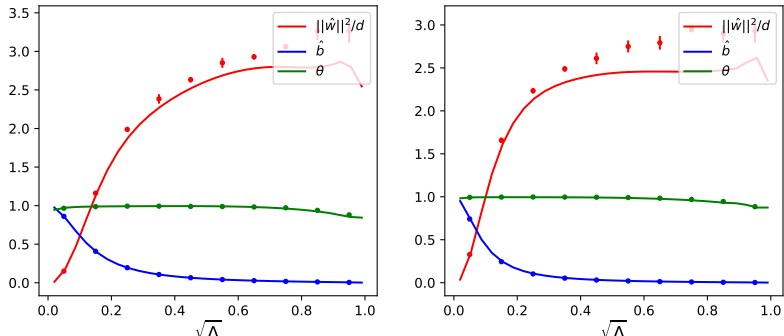

Figure 13: Cosine similarity $\theta$ (6) (green), weight norm $\|\hat{w}\|^2/d$ and skip connection strength $\hat{b}$ for the MNIST dataset (left), of which for simplicity only 1s and 7s were kept, and FashionMNIST (right), of which only boots and shoes were kept. In solid lines, the theoretical predictions resulting from using Conjecture 3.3 in its generic formulation (58) with the empirically estimated covariances and means. Dots represent numerical simulations of a DAE ($p = 1, \sigma = \tanh$) trained with $n = 784$ training points, using the `Pytorch` implementation of full-batch Adam, with learning rate $\eta = 0.05$ over 2000 epochs, weight decay $\lambda = 0.1$, averaged over $N = 5$ instances. Error bars represent one standard deviation. See Appendix D for full details on the preprocessing.

## D  Details on real data simulations

In this Appendix, we provide further details on the real data experiments presented in Fig. 2 and Fig. 4.

**Preprocessing**  The original MNIST [15] and FashionMNIST [16] data-sets were flattened (vectorized), centered, and rescaled by $400$ (MNIST) and $600$ (FashionMNIST). For simplicity and ease of discussion, for each data set, only two labels were kept, namely 1s and 7s for MNIST and boots and shoes for FashionMNIST. Note that the visual similarity between the two selected classes should make the denoising problem more challenging.

**Means and covariance estimation**  For each data-set, data from the same class were considered to belong to the same Gaussian mixture cluster. For simplicity, we kept the same number of training points in each cluster, so as to obtain balanced clusters ($\rho_1 = \rho_2 = {}^1\!/2$) for definiteness. For each cluster, the corresponding mean $\mu$ and covariance $\Sigma$ were numerically evaluated from the empirical mean and covariance over the 6000 boots (shoes) in the FashionMNIST training set, and the 6265 1s (7s) in the MNIST training set. The solid lines in Fig. 2 correspond to using those estimates in the asymptotic formulae of Conjecture 3.3, in their generic form (58) (see Appendix A) for sample complexity $\alpha = 1$ and $\ell_2$ regularization $\lambda = 0.1$.

`Pytorch` **simulations**  The red points in Fig. 2 were obtained from numerical simulations, by optimizing a $p = 1$ DAE (2) with $\sigma = \tanh$ activation using the `Pytorch` implementation of the full-batch Adam [43] optimizer, with weight decay $\lambda = 0.1$, learning rate $\eta = 0.05$, over $T = 2000$ epochs. For each value of $\Delta$, $n = 784$ images were drawn without replacement from the data-set training set, corrupted by noise, and fed to the DAE along with the clean image for training. The denoising test MSE, cosine similarity and summary statistics of the trained DAE were estimated after optimization using $n_{\text{test}} = 1000$ images randomly drawn from the test set (from which also only the relevant classes were kept, and equally represented– i.e. balanced). Each value was averaged over $N = 10$ instances of the train set, noise realization and test set. The obtained values were furthermore found to be robust with respect to the choice of the optimizer, learning rate and number of epochs.

### D.1  Complementary simulations

Fig.2 in the main text addresses the comparison of the theoretical prediction of Conjecture 3.3 with simulations on the real data-set, at the level of the denoising test MSE (5). For completeness, we show here the same comparison for the other learning metrics and summary statistics, namely the cosine similarity $\theta$ (6), weights norm $\|\hat{w}\|^2/d$ and trained skip connection strength $\hat{b}$. Theoretical asymptotic characterization are again provided by Conjecture 3.3 (10), (11) and (13).

They are plotted as solid lines in Fig. 13, and contrasted to numerical simulations (dots) on the real data-sets, by training a $p = 1$ DAE (2) using the `Pytorch` implementation of the Adam optimizer. All experiment details are the same as Fig. 2 of the main text, and can be found in the previous subsection. As can be observed, there is a gap, for large noise levels $\Delta$, between the theory and simulations of the weights norm $\|\hat{w}\|^2/d$ (red). On the other hand, the matchings for the cosine similarity $\theta$ (green) and skip connection strength $\hat{b}$ (blue) are perfect. Overall, Conjecture (3.3) therefore captures well (with the exception of $\|\hat{w}\|^2/d$ at large $\Delta$) the main metrics of the denoising problem on these two real data-sets. A more thorough investigation of the possible Gaussian universality of the problem is left to future work.

## E   Derivation of Corollary 3.4

In this Appendix, we provide the derivation of Corollary 3.4.

**Rescaling component $\hat{h}$**   We first provide a succinct derivation of the sharp asymptotic characterizations for the rescaling component $\hat{h}$ (3). Because this is only a scalar optimization problem over the single scalar parameter $c$ of $h_c$, the derivation is simple and straightforward. The empirical loss reads

$$
\begin{aligned}
\frac{1}{nd}\hat{\mathcal{R}}(c) &= \frac{1}{nd}\sum_{\mu=1}^{n}\left\|\boldsymbol{x}^{\mu} - c\left(\sqrt{1-\Delta}\boldsymbol{x}^{\mu} + \sqrt{\Delta}\boldsymbol{\xi}^{\mu}\right)\right\|^2 \\
&= (1-\sqrt{1-\Delta}c)^2\frac{1}{n}\sum_{\mu=1}^{n}\frac{\|\boldsymbol{x}^{\mu}\|^2}{d} + c^2\Delta\frac{1}{n}\sum_{\mu=1}^{n}\frac{\|\boldsymbol{\xi}^{\mu}\|^2}{d} + \frac{1}{n}\sum_{\mu=1}^{n}\frac{\|\boldsymbol{x}^{\mu}\|^2}{d}\frac{(\boldsymbol{x}^{\mu})^{\top}\boldsymbol{\xi}^{\mu}}{d} \\
&= (1-\sqrt{1-\Delta}c)^2\frac{1}{n}\sum_{\mu=1}^{n}\frac{\|\boldsymbol{x}^{\mu}\|^2}{d} + c^2\Delta + \mathcal{O}\left(1/\sqrt{d}\right) \\
&\stackrel{\text{CLT}}{=} (1-\sqrt{1-\Delta}c)^2\mathbb{E}_{\mathbb{P}}\left[\frac{\|\boldsymbol{x}^{\mu}\|^2}{d}\right] + c^2\Delta \\
&= (1-\sqrt{1-\Delta}c)^2\frac{1}{d}\sum_{k=1}^{K}\rho_k\left(\|\boldsymbol{\mu}_k\|^2 + \operatorname{Tr}\boldsymbol{\Sigma}_k\right) + c^2\Delta
\end{aligned}
\tag{104}
$$

Extremizing this last expression with respect to $c$ leads to

$$
\hat{c} = \frac{\frac{1}{d}\left(\sum_{k=1}^{K}\rho_k\operatorname{Tr}\boldsymbol{\Sigma}_k\right)\sqrt{1-\Delta}}{\frac{1}{d}\left(\sum_{k=1}^{K}\rho_k\operatorname{Tr}\boldsymbol{\Sigma}_k\right)(1-\Delta) + \Delta} = \frac{\frac{1}{d}\left(\sum_{k=1}^{K}\rho_k\int d\nu_{\gamma}(\gamma)\gamma_k\right)\sqrt{1-\Delta}}{\frac{1}{d}\left(\sum_{k=1}^{K}\rho_k\int d\nu_{\gamma}(\gamma)\gamma_k\right)(1-\Delta) + \Delta}.
\tag{105}
$$

which is exactly (10), as stated in Corollary 3.4. The associated MSE can be then readily computed as

$$
\begin{aligned}
\text{mse}_{\hat{r}} &= \mathbb{E}_{\mathcal{D}}\mathbb{E}_{\boldsymbol{x},\boldsymbol{\xi}}\left\|\boldsymbol{x} - \hat{c}\left(\sqrt{1-\Delta}\boldsymbol{x}^{\mu} + \sqrt{\Delta}\boldsymbol{\xi}\right)\right\|^2 \\
&= (1-\sqrt{1-\Delta}\hat{c})^2\frac{1}{d}\sum_{k=1}^{K}\rho_k\left(\|\mu_k\|^2 + \operatorname{Tr}\boldsymbol{\Sigma}_k\right) + \hat{c}^2\Delta \\
&= (1-\sqrt{1-\Delta}\hat{c})^2\frac{1}{d}\sum_{k=1}^{K}\rho_k\left(\int d\nu_{\tau}(\tau)\tau_k^2 + d\int d\mu_{\gamma}(\gamma)\gamma_k\right) + \hat{c}^2\Delta \\
&= \text{mse}_{\circ}.
\end{aligned}
\tag{106}
$$

This completes the proof of the asymptotic characterization of the rescaling component $\hat{r}$.

**Bottleneck network component $\hat{u}$**   For $\hat{u}$, one needs to go through the derivation presented in Appendix A by setting $b = 0$ from the beginning. It is straightforward to realize that the derivation goes through sensibly unaltered, and the only differing step is that the quadratic potential $\Psi_{\text{quad.}}$ is now a constant. One therefore only needs to set $\hat{b}$ in all the formulae of Conjecture 3.3 to access sharp asymptotics for $\hat{u}$. This concludes the derivation of Corollary 3.4    □

# F    Derivation of Corollary 3.5

We now turn to the derivation of Corollary 3.5. This corresponds to the limit $\Delta = 0$, where the input of the auto-encoder is also the clean data point. In terms of equations, essentially, this means that all the integrals over $\boldsymbol{\xi}, \lambda_\xi$ in the derivation of Conjecture 3.3 as presented in Appendix A should be removed. For the sake of clarity, and because this is an important case of particular interest, we provide here a complete, succinct but self-contained, derivation in the case of RAEs. For technical reasons, we limit ourselves to $\ell_2$ regularization. Furthermore, note that the model of an RAE with a skip connection is trivial, since such an architecture can achieve perfect reconstruction accuracy simply by setting the skip connection strength to 1 and the weights to 0. Therefore, we consider an RAE with*out* skip connection

$$f_w(\tilde{x}) = \frac{\boldsymbol{w}^\top}{\sqrt{d}} \sigma \left( \frac{\boldsymbol{w}\tilde{x}}{\sqrt{d}} \right). \tag{107}$$

## F.1    Derivation

The derivation of asymptotic characterizations for the metrics mse (5), $\theta$ (6) and summary statistics for the RAE (107) follow the same lines as for the full DAE (2), as presented in Appendix A. As in Appendix A, the first step is to evaluate the replicated partition function

$$\mathbb{E}_\mathcal{D} Z^s = \int \prod_{a=1}^s d\boldsymbol{w}_a e^{-\beta \sum_{a=1}^s g(\boldsymbol{w}^a)} \prod_{\mu=1}^n \sum_{k=1}^K \rho_k \mathbb{E}_{\boldsymbol{\eta}} \underbrace{e^{-\frac{\beta}{2} \sum_{a=1}^s \left\| \boldsymbol{\mu}_k + \boldsymbol{\eta} - \left[ \frac{\boldsymbol{w}_a^\top}{\sqrt{d}} \sigma \left( \frac{\boldsymbol{w}_a (\boldsymbol{\mu}_k + \boldsymbol{\eta})}{\sqrt{d}} \right) \right] \right\|^2}}_{(\star)}. \tag{108}$$

The exponent $(\star)$ can be expanded as

$$e^{-\frac{\beta}{2} \sum_{a=1}^s \left[ \text{Tr} \left[ \frac{\boldsymbol{w}_a \boldsymbol{w}_a^\top}{d} \sigma \left( \frac{\boldsymbol{w}_a (\boldsymbol{\mu}_k + \boldsymbol{\eta})}{\sqrt{d}} \right)^{\otimes 2} \right] - 2\sigma \left( \frac{\boldsymbol{w}_a (\boldsymbol{\mu}_k + \boldsymbol{\eta})}{\sqrt{d}} \right)^\top \frac{\boldsymbol{w}_a ((\boldsymbol{\mu}_k + \boldsymbol{\eta}))}{\sqrt{d}} + (\|\boldsymbol{\mu}_k\|^2 + \|\boldsymbol{\eta}\|^2 + 2\boldsymbol{\mu}_k^\top \boldsymbol{\eta}) \right]}. \tag{109}$$

Therefore,

$$\mathbb{E}_{\boldsymbol{\eta}}(\star) = \sum_{k=1}^K \rho_k e^{-\frac{\beta s}{2} \|\boldsymbol{\mu}_k\|^2} \int \frac{d\boldsymbol{\eta}}{(2\pi)^{\frac{d}{2}} \sqrt{\det \boldsymbol{\Sigma}_k}} \underbrace{e^{-\frac{1}{2} \boldsymbol{\eta}^\top \left( \boldsymbol{\Sigma}_k^{-1} + \beta s \mathbb{I}_d \right) \boldsymbol{\eta} - \beta s \boldsymbol{\mu}_k^\top \boldsymbol{\eta}}}_{P_{\text{eff.}}(\boldsymbol{\eta})}$$

$$\times e^{-\frac{\beta}{2} \sum_{a=1}^s \left[ \text{Tr} \left[ \frac{\boldsymbol{w}_a \boldsymbol{w}_a^\top}{d} \sigma \left( \frac{\boldsymbol{w}_a (\boldsymbol{\mu}_k + \boldsymbol{\eta})}{\sqrt{d}} \right)^{\otimes 2} \right] - 2\sigma \left( \frac{\boldsymbol{w}_a (\boldsymbol{\mu}_k + \boldsymbol{\eta})}{\sqrt{d}} \right)^\top \frac{\boldsymbol{w}_a (\boldsymbol{\mu}_k + \boldsymbol{\eta})}{\sqrt{d}} \right]}. \tag{110}$$

The effective prior over the $\boldsymbol{\eta}$ is therefore Gaussian with dressed mean and covariance

$$P_{\text{eff.}}(\boldsymbol{\eta}) = \mathcal{N}(\boldsymbol{\eta}; \boldsymbol{\mu}, \boldsymbol{C}), \tag{111}$$

where

$$\boldsymbol{C}^{-1} = \boldsymbol{\Sigma}_k^{-1} + \beta s \mathbb{I}_d, \qquad\qquad \boldsymbol{\mu} = -\beta s \boldsymbol{C} \boldsymbol{\mu}_k. \tag{112}$$

Observe that this mean is $\mathcal{O}(s)$ and can be safely neglected. Then

$$\mathbb{E}_{\boldsymbol{\eta}}(\star) = \sum_{k=1}^K \rho_k \frac{e^{-\frac{\beta s}{2} \|\boldsymbol{\mu}_k\|^2 + \frac{1}{2} \boldsymbol{\mu}^\top \boldsymbol{C}^{-1} \boldsymbol{\mu}}}{\sqrt{\det \boldsymbol{\Sigma}_k} \det(\boldsymbol{C})^{-\frac{1}{2}}}$$

$$\underbrace{\left\langle e^{-\frac{\beta}{2} \sum_{a=1}^s \left[ \text{Tr} \left[ \frac{\boldsymbol{w}_a \boldsymbol{w}_a^\top}{d} \sigma \left( \frac{\boldsymbol{w}_a (\boldsymbol{\mu}_k + \boldsymbol{\eta})}{\sqrt{d}} \right)^{\otimes 2} \right] - 2\sigma \left( \frac{\boldsymbol{w}_a (\boldsymbol{\mu}_k + \boldsymbol{\eta})}{\sqrt{d}} \right)^\top \frac{\boldsymbol{w}_a (\boldsymbol{\mu}_k + \boldsymbol{\eta})}{\sqrt{d}} \right]} \right\rangle_{P_{\text{eff.}}(\boldsymbol{\eta})}}_{(a)} \tag{113}$$

Again, like Appendix A, we introduce the local fields:

$$\lambda_a^\eta = \frac{\boldsymbol{w}_a (\boldsymbol{\eta} - \boldsymbol{\mu})}{\sqrt{d}}, \qquad\qquad h_a^k = \frac{\boldsymbol{w}_a \boldsymbol{\mu}_k}{\sqrt{d}}, \tag{114}$$

which have correlation

$$\langle \lambda_a^\eta \lambda_b^\eta \rangle \approx \frac{\boldsymbol{w}_a \boldsymbol{\Sigma}_k \boldsymbol{w}_b^\top}{d}. \tag{115}$$

We used the leading order of the covariance $\boldsymbol{C}$. One therefore has to introduce the order parameters:

$$Q_{ab} = \frac{\boldsymbol{w}_a \boldsymbol{w}_b^\top}{d} \in \mathbb{R}^{p \times p}, \qquad S_{ab}^k = \frac{\boldsymbol{w}_a \boldsymbol{\Sigma}_k \boldsymbol{w}_b^\top}{d} \in \mathbb{R}^{p \times p}, \qquad m_a^k = \frac{\boldsymbol{w}_a \boldsymbol{\mu}_k}{\sqrt{d}} \in \mathbb{R}^p. \tag{116}$$

The distribution of the local fields $\lambda_a^\eta$ is then simply:

$$(\lambda_a^\eta)_{a=1}^s \sim \mathcal{N}\left(0, S^k\right). \tag{117}$$

Going back to the computation, $(a)$ can be rewritten as

$$\left\langle e^{-\frac{\beta}{2} \sum\limits_{a=1}^s \mathrm{Tr}\left[Q_{aa}\sigma\left(m_a^k + \lambda_a^\eta\right)^{\otimes 2}\right] - \frac{\beta}{2} \sum\limits_{a=1}^s \left[-2\sigma\left(m_a^k + \lambda_a^\eta\right)^\top \left(m_a^k \lambda_a^\eta\right)\right]} \right\rangle_{\{\lambda_a^\eta\}_{a=1}^s}. \tag{118}$$

Introducing Dirac functions enforcing the definitions of $Q_{ab}, m_a$ brings the replicated function in the following form:

$$\mathbb{E}Z^s = \int \prod_{a=1}^s db_a \prod_{a,b} dQ_{ab} d\hat{Q}_{ab} \prod_{k=1}^K \prod_{a=1}^s dm_a d\hat{m}_a \prod_{k=1}^K \prod_{a,b} dS_{ab}^k d\hat{S}^k abe^s$$

$$\underbrace{e^{-d \sum\limits_{a \leq b} Q_{ab}\hat{Q}_{ab} - d \sum\limits_{k=1}^K \sum\limits_{a \leq b} S_{ab}^k \hat{S}_{ab}^k - \sum\limits_{k=1}^K \sum\limits_a dm_a^k \hat{m}_a^k}}_{e^{\beta sd\Psi_t}}$$

$$\underbrace{\int \prod_{a=1}^s d\boldsymbol{w}_a e^{-\beta \sum\limits_a g(\boldsymbol{w}^a) + \sum\limits_{a \leq b} \hat{Q}_{ab}\boldsymbol{w}_a \boldsymbol{w}_b^\top + \sum\limits_{k=1}^K \sum\limits_{a \leq b} \hat{S}_{ab}^k \boldsymbol{w}_a \boldsymbol{\Sigma}_k \boldsymbol{w}_b^\top + \sum\limits_{k=1}^K \sum\limits_a \hat{m}_a^k \sqrt{d} \boldsymbol{w}_a \boldsymbol{\mu}_k}}_{e^{\beta sd\Psi_w}}$$

$$e^{-\frac{\beta s}{2}\|\boldsymbol{\mu}\|^2} \underbrace{\left[\sum_{k=1}^K \rho_k \frac{1}{\sqrt{\det \boldsymbol{\Sigma}_k} \det\left(\boldsymbol{C}^{-1}\right)^{\frac{1}{2}}}(a)\right]^{\alpha d}}_{e^{\alpha sd^2 \Psi_{\text{quad.}} + \beta sd\Psi_y}}. \tag{119}$$

As in Appendix A, we have introduced the trace, entropic and energetic potentials $\Psi_t, \Psi_w, \Psi_y$. Since all the integrands scale exponentially (or faster) with $d$, this integral can be computed using a saddle-point method. To proceed further, note that the energetic term encompasses two types of terms, scaling like $sd^2$ and $sd$. More precisely,

$$\left[\sum_{k=1}^K \rho_k \frac{1}{\sqrt{\det \boldsymbol{\Sigma}_k} \det\left(\boldsymbol{C}^{-1}\right)^{\frac{1}{2}}}(a)\right]^{\alpha d} = \left[1 - sd \sum_{k=1}^K \rho_k \frac{1}{sd} \ln\left(\sqrt{\det \boldsymbol{\Sigma}_k} \det\left(\boldsymbol{C}^{-1}\right)^{\frac{1}{2}}\right) + s \sum_{k=1}^K \rho_k \frac{1}{s} \ln(a)\right]^{\alpha d}$$

$$= e^{-s\alpha d^2 \sum\limits_{k=1}^K \rho_k \frac{1}{sd} \ln\left(\sqrt{\det \boldsymbol{\Sigma}_k} \det\left(\boldsymbol{C}^{-1}\right)^{\frac{1}{2}}\right) + s\alpha d \sum\limits_{k=1}^K \rho_k \frac{1}{s} \ln(a)} \tag{120}$$

Remark that in contrast to Appendix A, the quadratic term is *constant* with respect to the network parameters, and therefore one no longer needs to discuss its extremization. Following Appendix A, we assume the RS ansatz

$$\forall a, \qquad m_a^k = m_k, \qquad \hat{m}_a^k = \hat{m}_k$$

$$\forall a, b, \qquad S_{ab}^k = (r_k - q_k)\delta_{ab} + q_k, \qquad \hat{S}_{ab}^k = -\left(\frac{\hat{r}_k}{2} + \hat{q}_k\right)\delta_{ab} + \hat{q}_k,$$

$$\forall a, b, \qquad Q_{ab} = (r - q)\delta_{ab} + q, \qquad \hat{Q}_{ab} = -\left(\frac{\hat{r}}{2} + \hat{q}\right)\delta_{ab} + \hat{q}. \tag{121}$$

In the following, we sequentially simplify the potentials $\Psi_w, \Psi_y, \Psi_t$ under this ansatz.

**Entropic potential** It is convenient to introduce before proceeding the variance order parameters

$$\hat{V} \equiv \hat{r} + \hat{q}, \qquad\qquad\qquad \hat{V}_k \equiv \hat{r}_k + \hat{q}_k. \qquad (122)$$

The entropic potential can then be expressed as

$$
e^{\beta s d \Psi_w}
$$

$$
= \int \prod_{a=1}^{s} d\boldsymbol{w}_a e^{-\beta \sum_a g(\boldsymbol{w}^a) - \frac{1}{2} \sum_{a=1}^{s} \mathrm{Tr}[\hat{V}\boldsymbol{w}_a\boldsymbol{w}_a^\top] + \frac{1}{2} \sum_{a,b} \mathrm{Tr}[\hat{q}\boldsymbol{w}_a\boldsymbol{w}_b^\top] + \hat{m} \sum_{k=1}^{K} \sum_{a=1}^{s} \sqrt{d}\hat{m}_k^\top \boldsymbol{w}_a^\top \boldsymbol{\mu}_k}
$$

$$
\times e^{-\frac{1}{2} \sum_{k=1}^{K} \sum_{a=1}^{s} \mathrm{Tr}[\hat{V}_k \boldsymbol{w}_a \boldsymbol{\Sigma}_k \boldsymbol{w}_a^\top] + \frac{1}{2} \sum_{k=1}^{K} \sum_{a,b} \mathrm{Tr}[\hat{q}_k \boldsymbol{w}_a \boldsymbol{\Sigma}_k \boldsymbol{w}_b^\top]}
$$

$$
= \int D\Xi_0 \prod_{k=1}^{K} D\Xi_k
$$

$$
\left[ \int d\boldsymbol{w} e^{-\beta g(\boldsymbol{w}) - \frac{1}{2} \mathrm{Tr}\left[\hat{V}\boldsymbol{w}\boldsymbol{w}^\top + \sum_{k=1}^{K} \hat{V}_k \boldsymbol{w}\boldsymbol{\Sigma}_k\boldsymbol{w}^\top\right] + \left( \sum_{k=1}^{K} \sqrt{d}\hat{m}_k\boldsymbol{\mu}^\top + \sum_{k=1}^{K} \Xi_k \odot (\hat{q}_k\otimes\boldsymbol{\Sigma}_k)^{\frac{1}{2}} + \Xi_0\odot(\hat{q}\otimes\mathbb{I}_d)^{\frac{1}{2}} \right)\odot\boldsymbol{w}} \right]^s
$$

$$
= \int \underbrace{D\Xi_0 \prod_{k=1}^{K} D\Xi_k}_{\equiv D\Xi}
$$

$$
\left[ \int d\boldsymbol{w} e^{-\beta g(\boldsymbol{w}) - \frac{1}{2}\boldsymbol{w}\odot\left[\hat{V}\otimes\mathbb{I}_d + \sum_{k=1}^{K} \hat{V}_k\otimes\boldsymbol{\Sigma}_k\right]\odot\boldsymbol{w} + \left( \sum_{k=1}^{K} \sqrt{d}\hat{m}_k\boldsymbol{\mu}^\top + \sum_{k=1}^{K} \Xi_k\odot(\hat{q}_k\otimes\boldsymbol{\Sigma}_k)^{\frac{1}{2}} + \Xi_0\odot(\hat{q}\otimes\mathbb{I}_d)^{\frac{1}{2}} \right)\odot\boldsymbol{w}} \right]^s. \qquad (123)
$$

Therefore

$$
\beta\Psi_w
$$

$$
\frac{1}{d} \int D\Xi \ln \left[ \int d\boldsymbol{w} e^{-\beta g(\boldsymbol{w}) - \frac{1}{2}\boldsymbol{w}\odot\left[\hat{V}\otimes\mathbb{I}_d + \sum_{k=1}^{K} \hat{V}_k\otimes\boldsymbol{\Sigma}_k\right]\odot\boldsymbol{w} + \left( \sum_{k=1}^{K} \sqrt{d}\hat{m}_k\boldsymbol{\mu}^\top + \sum_{k=1}^{K} \Xi_k\odot(\hat{q}_k\otimes\boldsymbol{\Sigma}_k)^{\frac{1}{2}} + \Xi_0\odot(\hat{q}\otimes\mathbb{I}_d)^{\frac{1}{2}} \right)\odot\boldsymbol{w}} \right].
$$

$$(124)$$

**Energetic potential** The inverse of $S_k$ can be computed like in Appendix A, leading to the following expression for $(a)$:

$$
(a) = \int_{\mathbb{R}^p} \frac{\prod_{a=1}^{s} d\lambda_a^\eta}{(2\pi)^{ms/2}\sqrt{\det S_k}} e^{-\frac{1}{2} \sum_{a=1}^{s} (\lambda_a^\eta)^\top (\tilde{r}_k - \tilde{q}_k)\lambda_a^\eta - \frac{1}{2} \sum_{a,b} (\lambda_a^\eta)^\top \tilde{q}_k \lambda_b - \frac{\beta}{2} \sum_{a=1}^{s} (*)}
$$

$$
= \int_{\mathbb{R}^p} \frac{D\eta}{(2\pi)^{ms/2}\sqrt{\det S_k}} \left[ \int_{\mathbb{R}^p} d\lambda_\eta e^{-\frac{1}{2}\lambda_\eta^\top V_k^{-1}\lambda_\eta + \lambda_\eta^\top V_k^{-1} q_k^{\frac{1}{2}}\eta - \frac{\beta}{2}(*)} \right]^s
$$

$$
= \int_{\mathbb{R}^p} D\eta \left[ \int \mathcal{N}\left(\lambda_\eta; q_k^{\frac{1}{2}}\eta, V\right) e^{-\frac{\beta}{2}(*)} \right]^s
$$

$$
= 1 + s \int_{\mathbb{R}^p} D\eta \ln \left[ \int \mathcal{N}\left(\lambda_\eta; q_k^{\frac{1}{2}}\eta, V\right) e^{-\frac{\beta}{2}(*)} \right], \qquad (125)
$$

where we noted with capital $D$ an integral over $\mathcal{N}(0, \mathbb{I}_p)$. Therefore

$$
\beta\Psi_y = \sum_{k=1}^{K} \rho_k \int_{\mathbb{R}^p} D\eta \ln \left[ \int \mathcal{N}\left(\lambda_\eta; q_k^{\frac{1}{2}}\eta, V\right) e^{-\frac{\beta}{2}(*)} \right]. \qquad (126)
$$

**Zero-temperature limit** In this subsection, we take the zero temperature $\beta \to \infty$ limit. Rescaling

$$\frac{1}{\beta}V_k \leftarrow V_k, \qquad \beta\hat{V}_k \leftarrow \hat{V}_k, \qquad \beta^2\hat{q}_k \leftarrow \hat{q}_k, \qquad \beta\hat{m}_k \leftarrow \hat{m}_k \tag{127}$$

one has that

$$\Psi_w = \frac{1}{2d}\operatorname{Tr}\left[\left(\hat{V} \otimes \mathbb{I}_d + \sum_{k=1}^K \hat{V}_k \otimes \boldsymbol{\Sigma}_k\right)^{-1} \odot \left(\hat{q} \otimes \mathbb{I}_d + \sum_{k=1}^K \hat{q}_k \otimes \boldsymbol{\Sigma}_k + d\left(\sum_{k=1}^K \hat{m}_k\boldsymbol{\mu}_k^\top\right)^{\otimes 2}\right)\right]$$
$$- \frac{1}{d}\mathbb{E}_{\boldsymbol{\Xi}}\mathcal{M}_r(\boldsymbol{\Xi}), \tag{128}$$

where we introduced the Moreau envelope

$$\mathcal{M}_r(\boldsymbol{\Xi})$$
$$= \inf_{\boldsymbol{w}}\left\{\frac{1}{2}\left\|\left(\hat{V}\otimes\mathbb{I}_d + \sum_{k=1}^K \hat{V}_k\otimes\boldsymbol{\Sigma}_k\right)^{\frac{1}{2}}\odot\boldsymbol{w} - \left(\hat{V}\otimes\mathbb{I}_d + \sum_{k=1}^K \hat{V}_k\otimes\boldsymbol{\Sigma}_k\right)^{-\frac{1}{2}}\odot\left((\hat{q}\otimes\mathbb{I}_d)^{\frac{1}{2}}\odot\boldsymbol{\Xi}_0 + \sum_{k=1}^K (\hat{q}_k\otimes\boldsymbol{\Sigma}_k)^{\frac{1}{2}}\odot\boldsymbol{\Xi}_k + \sqrt{d}\sum_{k=1}^K \hat{m}_k\boldsymbol{\mu}_k^\top\right)\right\|^2 + g(\boldsymbol{w})\right\}. \tag{129}$$

For the case of $\ell_2$ regularization, the Moreau envelope presents a simple expression, and the entropy potential assumes the simple form

$$\Psi_w = +\frac{1}{2d}\operatorname{Tr}\left[\left(\lambda\mathbb{I}_m \odot \mathbb{I}_d + \hat{V} \otimes \mathbb{I}_d + \sum_{k=1}^K \hat{V}_k \otimes \boldsymbol{\Sigma}_k\right)^{-1} \odot \left(\hat{q} \otimes \mathbb{I}_d + \sum_{k=1}^K \hat{q}_k \otimes \boldsymbol{\Sigma}_k + d\left(\sum_{k=1}^K \hat{m}_k\boldsymbol{\mu}_k^\top\right)^{\otimes 2}\right)\right]. \tag{130}$$

The energetic potential also simplifies in this limit to

$$\Psi_y = -\sum_{k=1}^K \rho_k\mathbb{E}_\eta\mathcal{M}_k(\eta) \tag{131}$$

where

$$\mathcal{M}_k(\eta) = \frac{1}{2}\inf_{x,y}\left\{\operatorname{Tr}\left[V_k^{-1}\left(y - q_k^{\frac{1}{2}}\eta - m_k\right)^{\otimes 2}\right] + \operatorname{Tr}\left[q\sigma(y)^{\otimes 2}\right] - 2\sigma(y)^\top y\right\}. \tag{132}$$

**Trace potential** It is immediate to see that the trace potential can be expressed as

$$\Psi_t = \frac{\hat{V}q - \hat{q}V}{2} + \frac{1}{2}\sum_{k=1}^K (\operatorname{Tr}\left[\hat{V}_k q_k\right] - \operatorname{Tr}[\hat{q}_k V_k]) - \sum_{k=1}^K m_k\hat{m}_k. \tag{133}$$

Then the total free entropy for RAEs (107) reads

$$\Phi = \operatorname*{extr}_{q,m,V,\hat{q},\hat{m},\hat{V},\{q_k,m_k,V_k,\hat{q}_k,\hat{m}_k,\hat{V}_k\}_{k=1}^K} \frac{\hat{V}q - \hat{q}V}{2} + \frac{1}{2}\sum_{k=1}^K (\operatorname{Tr}\left[\hat{V}_k q_k\right] - \operatorname{Tr}[\hat{q}_k V_k]) - \sum_{k=1}^K m_k\hat{m}_k$$
$$- \alpha\sum_{k=1}^K \rho_k\mathbb{E}_\eta\mathcal{M}_k(\eta) + \frac{1}{2}\int d\nu(\gamma,\tau)\operatorname{Tr}\left[\left(\lambda + \gamma_k\hat{V}_k\right)^{-1}\left(\sum_{k=1}^K \gamma_k\hat{q}_k + \sum_{1\le k,j\le K} \tau_j\tau_k\hat{m}_j\hat{m}_k^\top\right)\right]. \tag{134}$$

Comparing to the free energy derived in Appendix A, it is immediate to see that the Moreau envelope $\mathcal{M}_k$ for the RAE just corresponds to removing the second term from the DAE $\mathcal{M}_k$, and setting $x = 0$. The disappearance of the second term in the Moreau envelope has the effect to kill the $V$-dependence, resulting in $\hat{q} = 0$ when extremizing with respect to $V$. These observations, in addition to the already taken limits $\Delta, b = 0$, finish the derivation of Corollary 3.5. $\square$

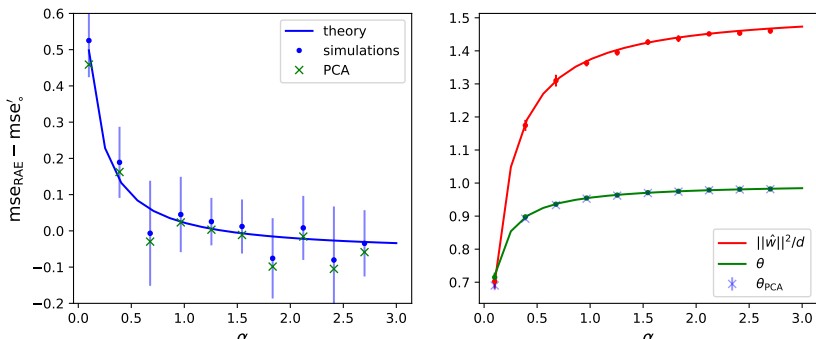

Figure 14: $\alpha = 1, K = 2, \rho_{1,2} = 1/2, \Sigma_{1,2} = 0.3 \times \mathbb{I}_d, p = 1, \sigma = \tanh$; the cluster mean $\boldsymbol{\mu}_1 = -\boldsymbol{\mu}_2$ was taken as a random Gaussian vector of norm 1. (left) In blue, difference in test reconstruction MSE between the RAE (Corollary 3.5) and the leading order term $\mathrm{mse}'_\circ$ (135). Solid lines correspond to the sharp asymptotic characterization of Corollary 3.5. Dots represent numerical simulations for $d = 700$, training the RAE using the Pytorch implementation of full-batch Adam, with learning rate $\eta = 0.05$ over 2000 epochs, weight decay $\lambda = 0.1$, averaged over $N = 10$ instances. Error bars represent one standard deviation. (right) Cosine similarity $\theta$ (6) (green), squared weight norm $\|\hat{\boldsymbol{w}}\|^2/d$ (red). Solid lines correspond to the theoretical characterization of Corollary 3.5; dots are numerical simulations. The cosine similarity with the cluster means of the principal component of the training samples is represented in blue crosses, and superimposes perfectly with the analogous curve learnt by the RAE.

## F.2 Reconstruction MSE

Having obtained a sharp asymptotic characterization of the summary statistics for the RAE, we now turn to the learning metrics. It is easy to see that the formula for the cosine similarity $\theta$ (6) carries over unchanged from the DAE case, see Appendix A. We therefore focus on the test MSE:

$$
\begin{aligned}
\mathrm{mse}_{\mathrm{RAE}} &= \mathbb{E}_{k,\boldsymbol{\eta},\xi} \left\| \boldsymbol{\mu}_k + \boldsymbol{\eta} - \left( \frac{\hat{\boldsymbol{w}}^\top}{\sqrt{d}} \sigma \left( \frac{\hat{\boldsymbol{w}}(\boldsymbol{\mu}_k + \boldsymbol{\eta})}{\sqrt{d}} \right) \right) \right\|^2 \\
&= \underbrace{\sum_{k=1}^K \rho_k \left( \|\boldsymbol{\mu}_k\|^2 + \mathrm{Tr}\,\boldsymbol{\Sigma}_k \right) + \sum_{k=1}^K \rho_k \mathbb{E}_z^{\mathcal{N}(0,1)} \left[ \mathrm{Tr}\left[ q\sigma \left( m_k + \sqrt{q_k}z \right)^{\otimes 2} \right] \right]}_{\equiv \mathrm{mse}'_\circ} \\
&\qquad - 2 \sum_{k=1}^K \rho_k \mathbb{E}_{u,v}^{\mathcal{N}(0,1)} \left[ \sigma \left( m_k + \sqrt{q_k}u \right) \right]^\top \left( m_k + \sqrt{q_k}u \right)
\end{aligned}
\tag{135}
$$

## F.3 RAEs are limited by the PCA MSE

We close this Appendix by showing how the RAE Corollary 3.5 recovers the well-known Eckart-Young theorem [10]. The difference $\mathrm{mse} - \mathrm{mse}'_\circ$ for an RAE learning to reconstruct a binary, isotropic homoscedastic mixture (see Fig. 1) is shown in Fig. 14, from a training set with sample complexity $\alpha = 1$. Because $\mathrm{mse} - \mathrm{mse}'_\circ$ is essentially a correction to the leading term $\mathrm{mse}'_\circ$, estimating this difference is slightly more challenging numerically, whence rather large error bars. Nevertheless, it can be observed that the agreement between the theory (solid blue line) and the simulations (dots) gathered from training an $p = 1$ RAE using the Pytorch implementation of Adam is still very good. In terms of cosine similarity and learnt weight norm, the agreement is perfect, see Fig. 3.5 (right). For comparison, the performance of PCA reconstruction (crosses) has been overlayed on the RAE curves. Importantly, observe that in terms of MSE, PCA reconstruction always leads to smaller MSE, in agreement with the well-known Eckart-Young theorem [10]. From the cosine similarity, it can be seen that the RAE essentially learns the principal component of the train set, see the discussions in e.g. [12, 11, 8, 9].

We finally stress that, in contrast to all previous figures, the $x$ axis in Fig. 3.5 is the *sample complexity* $\alpha$, rather than the noise level $\Delta$. As a matter of fact, while our result recover the well-known fact that the MSE of RAEs is limited (lower-bounded) by the PCA test error, Corollary 3.5 allows us to investigate how the RAE approaches the PCA

performance as a function of the number of training samples, a characterization which has importantly been missing from previous studies for non-linear RAEs [8, 9].

