# OpenReview forum: "High-dimensional Asymptotics of Denoising Autoencoders"
_NeurIPS.cc/2023/Conference — NeurIPS 2023 spotlight_

### Official Review · Reviewer_pba7 · 2023-07-04

**Soundness:** 3 good
**Presentation:** 3 good
**Contribution:** 3 good
**Rating:** 7
**Confidence:** 3

**Summary:**

This paper studies the performance of denoising autoencoders (DAEs) in high-dimensional settings. The DAEs are trained on data sampled from a Gaussian mixture with $K$ components perturbed by isotropic Gaussian noise. The DAEs are one-layer networks with arbitrary activation functions, tied weights, and a skip connection. They are trained with an $L_2$-regularized loss function, and the high-dimensional limit is considered, where the ratio of the number of training points to the input dimension converges to a constant α.

The main results of the paper are formulas for the denoising test mean squared error (MSE) for the full DAE, as well as two simpler architectures: a "bottleneck network" in which the skip connection is removed, and a "scalar linear network" that simply rescales the input by a scalar. The final formulas involve complicated integrals and optimization problems, but they only depend on low-dimensional quantities, such as the number of neurons in the hidden layer and the number of components in the Gaussian mixture.

The authors use the derived asymptotic test error to analyze the role and importance of each component in the performance of DAEs. They find that the skip connection is essential for good performance and that DAEs can outperform other denoising methods, such as principal component analysis (PCA).

Taking the noiseless limit, where the variance of the noise converges to zero, they can also derive the test error of reconstruction autoencoders, which are trained with noiseless inputs.

**Strengths:**

The authors managed to derive fairly complicated asymptotic formulas via the replica method from statistical physics. The empirical experiments seem to confirm the accuracy of their predictions. This provides yet another example of the successful application of the replica method and its surprisingly powerful nature.
This paper fills a gap in the literature and complements previous results that mainly focused on RAEs and the infinite data regime (optimization of the population loss).

The exact formulas derived in this paper are utilized to derive interesting non-trivial results, which can serve as stimuli for future research. Specifically:

a) The MSE obtained by training a DAE with gradient descent matches the asymptotic predictions, suggesting that while non-convex, the optimization problem has a favorable landscape.

b) The exact formulas obtained for the Gaussian mixture predictions align with those obtained for a DAE trained on MNIST datasets, indicating the presence of universality phenomena.

c) As α increases, the denoising MSE of the DAE approaches the asymptotic performance of the oracle denoiser (derived in the appendix).

**Weaknesses:**

The paper exhibits weaknesses similar to other papers in this field. The exact analytical formulas derived are somewhat obscure, and it seems that several of the observations made in the paper can be deduced directly from the results of the empirical experiments, without relying on the analytical formulas (e.g. the improved performance of the network with skip connections compared to the one without can be easily inferred). The authors should emphasize the results that their analytical formulas allow to obtain.


**Questions:**

- It appears that all the experiments were conducted with a number of neurons $p = 1$. Could you please clarify if the empirical results align with the replica predictions when $p > 1$? Additionally, does the "universality phenomenon" manifest itself also in this case?

- Based on Figure 3, it appears that the performance of the DAE approaches that of the oracle denoiser. Can this convergence be rigorously derived from equation (13)?

- What happens when $K = 1$? Do the DAE, PCA, and bottleneck network exhibit similar performance in this scenario?

**Limitations:**

This theoretical paper does not possess any immediate potential negative societal impacts.

---

> ### Author Rebuttal · Authors · 2023-08-06
>
> We thank the reviewer for their insightful comments. We answer their questions below:
>
> >The authors should emphasize the results that their analytical formulas allow to obtain.
>
> We believe a strength of our analysis lies in the fact that it allows to characterize learning metrics at the global optima of the empirical risk, whereas there is a priori no guarantee that a purely experimental study can consistently reach them. Furthermore, our analytical formulae allow to probe the asymptotic behaviour of the learning problem, while a purely empirical approach would require to disentangle the finite size effects from the true asymptotic behaviour. Finally, the expressions further make it possible to establish the rates of convergence with the sample complexity of the learning metrics to their infinite data (population) $\alpha\to \infty$ limit. We shall include this analysis in the final version of the manuscript. For enhanced clarity, we will also add this discussion on the insights afforded by our analytical formulae.
>
> >It appears that all the experiments were conducted with a number of neurons $p=1$
> . Could you please clarify if the empirical results align with the replica predictions when
> $p>1$. Additionally, does the "universality phenomenon" manifest itself also in this case?
>
> While we state the asymptotic characterization in full generality, we indeed restrict the discussion of results to $p=1$ hidden units. This already presents a number of interesting and novel properties that we describe. Solving the equations for $p\ge 2$ requires more analytical work, along the lines of e.g in [29]. We further anticipate that a richer phenomenology, such as a specialization transition like [29], will arise for $p\ge2$, and a thorough exploration thereof warrants a full separate line of work that we are undertaking currently, and falls out of the scope of this first work where we introduce the model, framework and present the interesting observations for $p=1$.
>
> >Based on Figure 3, it appears that the performance of the DAE approaches that of the oracle denoiser. Can this convergence be rigorously derived from equation (13)?
>
> Thank you for this question. While the oracle denoiser (B4) admits the functional form of a DAE, the corresponding encoder and decoder weights are proportional, but not strictly equal, and therefore cannot be realized exactly by the weight-tied DAE -- although the difference of performance is quantitatively small, see Fig. 3. In the final version of the manuscript, we shall provide a characterization of the $\alpha\to\infty$ limit of the DAE performance, and characterize precisely the difference with the oracle performance.
>
> >What happens when $K=1$? Do the DAE, PCA, and bottleneck network exhibit similar performance in this scenario?
>
> The $K=1$ corresponds to setting $||\mu|| =0$ in our analytical characterization (Result 3.3). We shall include a discussion of this special case in the supplementary material of the final manuscript. In short, the oracle denoiser (B4) reduces for $K=1$ to a simple rescaling, so only the rescaling component of the DAE is actually needed. Indeed, as discussed in section 4, l.278-290, the role of the bottleneck component is to learn the data structure (as given by $\mu$), which it leverages to improve the denoising performance. In the unstructured $K=1$ ($\mu=0$) case, this component is not needed, and its presence actually causes the DAE to overfit the data, leading it to perform worse than the rescaling. Similarly, PCA denoising performs worse on unstructured data and leads to a mse worse by $\Theta(d)$, like in the $K>1$ case. In Fig. 2 of the attached pdf, we reproduce Fig.1 (left) and Fig. 3 (b) for $K=1$, comparing the oracle, rescaling, DAE, bottleneck and PCA denoisers. This figure will be included in the supplementary material of the final manuscript.

---

> > ### Comment · Reviewer_pba7 · 2023-08-13
> >
> > I would like to thank the authors for their thorough answers to my questions and for running additional experiments. These have clarified my doubts and I have accordingly increased my score. I look forward to reading the final version of the paper!

---

### Official Review · Reviewer_ZGsS · 2023-07-04

**Soundness:** 3 good
**Presentation:** 3 good
**Contribution:** 4 excellent
**Rating:** 7
**Confidence:** 4

**Summary:**

The authors consider a two layer weight-tied denoising autoencoder with a skip connection in the regime of vanishing rate and samples proportional to the dimension. They heuristically derive an exact characterization of the optimal network parameters and the corresponding network performance in the the high-dimensional limit using the replica method. Their experiments show that their results match on synthetic data and are very close to the performance on more practical data.

**Strengths:**

- Exact characterization of all quantities of interest in the high-dimensional limit.
- Strong experimental validation of theoretical claims

**Weaknesses:**

- From a theoretical perspective there is a lack of rigor to this method


**Questions:**

- Why do the limits in Line 152 exist?
- In Figure 3 for high noise and low sample complexity the performance of the full DAE is worse than simply rescaling. Since the full network can also act as a rescaling network by setting $w=0$, this implies that during training the global optimum is not found. Can you elaborate on this?

**Limitations:**

- It should be stated more clearly that the way the replica method is carried out only serves as a strong heuristic and not a rigorous proof.

---

> ### Author Rebuttal · Authors · 2023-08-06
>
> We thank the reviewer for their constructive comments. We answer below their questions:
>
> >It should be stated more clearly that the way the replica method is carried out only serves as a strong heuristic and not a rigorous proof.
>
> The reviewer is entirely correct about the heuristic nature of the result, and we will further emphasize this fact after the statement of result 3.1 in the final version of the manuscript.
>
> >Why do the limits in Line 152 exist?
>
> This is used as an assumption in the replica computation, which has been verified in related settings in previous works, see e.g. [47]. We will state the main assumptions of the method, i.e. concentration of overlaps and the uniqueness of the replica limit, explicitly.
>
> >In Figure 3 for high noise and low sample complexity the performance of the full DAE is worse than simply rescaling. Since the full network can also act as a rescaling network by setting $w=0$
> , this implies that during training the global optimum is not found. Can you elaborate on this?
>
>  Thank you for the question. While the full DAE can indeed act as a simple rescaling, it has access to a limited amount $n$ of training samples, and at the global optimum of the *empirical* loss, $w\ne 0$. For large noise levels $\Delta$, the DAE overfits the training data, leading to a performance worse than simple rescaling as the reviewer correctly observes in Fig. 1 and 3. Note that as the sample complexity $\alpha$ increases, this overfitting disappears, as the empirical loss becomes closer to the population loss (see Fig. 3 (a)). The fact that our study captures this overfitting phenomenon is a strength of our analysis, which allows to cover the effect of a finite amount of training data. Finally, note that the overfitting can be mitigated by adjusting the strength $\lambda$ of the regularization. In Fig. 1 of the attached pdf, we reproduce the same curve as Fig.1, for various $\lambda$. Observe that the overfitting disappears for larger regularization --e.g.$\lambda=0.8$ (instead of $\lambda=0.1$ in Fig.1)--, at the expense of worsened performance for small noise levels $\Delta$. We shall include this figure and additional discussion in the revised version of the manuscript.

---

> > ### Comment · Reviewer_ZGsS · 2023-08-17
> >
> > I would like to thank the authors for their clear answers and the added plot which has cleared up my doubts. In accordance I slightly increased the evaluation.

---

### Official Review · Reviewer_DrHV · 2023-07-05

**Soundness:** 3 good
**Presentation:** 3 good
**Contribution:** 3 good
**Rating:** 7
**Confidence:** 4

**Summary:**

This paper presents theoretical results of the test error of 2-layer denoising auto-encoder. In a high-dimensional limit regime where data distribution follows from Mixture of Gaussian, closed-form expressions of the error are obtained. The results are further analyzed and supported by numerical experiments on real data sets.

**Strengths:**

- The main result (result 3.3) is highly non-trivial and provides a tight formula for the test error of the 2-layer denoising auto-encoder (DAE).
- It shows that DAE can perform much better than PCA in terms of MSE error.
- The role and importance of the skip connection is also highlighted in both theory and in practice, as well as the non-linearity in DAE.

**Weaknesses:**

- The optimal MSE error mse_o grows with the data dimension d (eq. 9), however the gap between mse_f and mse_o (eq 8) remains a constant, this suggests that the difference between DAE and PCA is not so significant in terms of the relative error of MSE, i.e. | mse_f - mse_o | / mse_o vs. | mse_PCA - mse_o | / mse_o. Thus it is still not very clear whether DAE brings something essentially different to PCA or not (I agree that at least they are different).
- In the main result 3.3, it is not clear how the regularization term g is used, i.e. under Assumption 3.2, the parameter lambda does not appear in eq. (8), this is a bit strange to me .

**Questions:**

- (line 33) Usually we say the sample complexity is related to n, it is here a bit strange to say it is alpha
- (line 89) What does this mean sigma(.) stays order 1 ?
- (eq 13) is there a unique solution in this system of 10 equations of 10 variables ?
- (line 219) regarding the Gaussian universality, do you need to perform any normalization on the data for this to hold ? The dimension of MNIST seems quite small, what is it in your mind for the universality to hold ?
- (line 255) to clarify a previous point regarding PCA, does mse_f grows linearly with d ?  Do you have an idea of what is the relative error of MSE of DAE and PCA ?

**Limitations:**

- The main result is very specific, but this is quite normal in the literature .

---

> ### Author Rebuttal · Authors · 2023-08-06
>
> We thank the reviewer for their insightful questions, which we answer below:
>
> >The optimal MSE error $mse_o$ grows with the data dimension d (eq. 9), however the gap between $mse_f$ and $mse_o$ (eq 8) remains a constant, this suggests that the difference between DAE and PCA is not so significant in terms of the relative error of MSE, i.e. $| mse_f - mse_o | / mse_o$ vs. $| mse_PCA - mse_o | / mse_o$. Thus it is still not very clear whether DAE brings something essentially different to PCA or not (I agree that at least they are different).[...]to clarify a previous point regarding PCA, does $mse_f$ grows linearly with d ? Do you have an idea of what is the relative error of MSE of DAE and PCA ?
>
> The difference between the mse achieved by the DAE $mse_{f}$ and the mse achieved by PCA is in fact significant and of order $\Theta(d)$. As we explain in l.255, $| mse_{PCA} - mse_\circ|=\Theta(d)$ and  $| mse_{PCA} - mse_f|=\Theta(d)$ , so both  relative errors $| mse_{PCA} - mse_\circ|/mse_\circ$.  and  $| mse_{PCA} - mse_f|/mse_f$ are $\Theta(1)$. Therefore, both the DAE $\hat{f}$ and the rescaling $\hat{r}$ are considerably better than PCA. On the other hand, $|mse_{f} - mse_\circ|=\Theta(1)$, which hence implies $| mse_{f} - mse_\circ|/ mse_\circ=\Theta(1/d)$. Note that this also means the improvement of the full DAE $\hat{f}$ upon the rescaling MSE $mse_\circ$ is subleading. It is in a sense a limitation of our theoretical model that the interesting phenomenology appears in the subleading order. At the same time, our real-data experiments show that the effects we predict from the theory are visible in real data, and correspond to visually significant changes (see images in Fig. 2 (left) and Fig. 4).
>
> >In the main result 3.3, it is not clear how the regularization term g is used, i.e. under Assumption 3.2, the parameter lambda does not appear in eq. (8), this is a bit strange to me .
>
> Equation (8) involves the summary statistics $m,q,V, m_k,q_k, V_k$, which in turn depend on $\lambda$ through (13). We agree that a comment on this implicit dependence will improve the readability of the manuscript, and shall include one in the final version.
>
> >(line 33) Usually we say the sample complexity is related to n, it is here a bit strange to say it is alpha
>
> The denomination of sample complexity for $\alpha$ (which is equal to the number of samples $n$ normalized by the input dimension $d$) has been consistently used in the exact high-dimensional asymptotics literature. We will comment on it to avoid possible confusion for readers used to a different terminology. Since the two parameters are straightforwardly related by a factor $\frac{1}{d}$, they essentially describe the same quantity, with $\alpha$ presenting the advantage of being $ \Theta(1)$ in the asymptotic limit considered. For the sake of clarity, we choose to keep this denomination, but will include an additional comment in l.130 when $\alpha$ is introduced.
>
> >(line 89) What does this mean sigma(.) stays order 1 ?
>
> What we mean is that the argument of the function $\sigma$ in (2), namely $\frac{w\tilde{x}}{\sqrt{d}}$, is of order $1$ as $d\to\infty$. We will clarify the phrasing in the revised manuscript.
>
> >(eq 13) is there a unique solution in this system of 10 equations of 10 variables ?
>
> While at this point we do not have a proof of (13) having a single solution, in all the studied cases, only a single solution was found, up to symmetry. (Note that indeed if $\sigma$ is odd, the function $f$ is invariant under $w\to -w$. Therefore, for each fixed point of (13), the solution obtained by flipping the sign of $m$ is also a solution, but leads to the same learning metrics $\theta$ and mse.) Further, this solution agrees with numerical experiments.
>
> >(line 219) regarding the Gaussian universality, do you need to perform any normalization on the data for this to hold ? The dimension of MNIST seems quite small, what is it in your mind for the universality to hold ?
>
> In Fig.2, the datasets are indeed flattened, centered, as described in Appendix D. Since the components of the resulting vectors are comprised between $0$ and $255$ (as they correspond to color levels), we further divide them by $400$ so as to have components of order $1$. The precise value of this normalization was not found to impact the agreement between the theory and simulations. Note that the quantitative characterization of the generic conditions under which Gaussian universality holds for real datasets is still very much ongoing work in machine learning theory, even for supervised learning settings. Informally, in the present work, the intuition lies in the fact that the width of the hidden layer is finite, and thus much smaller than the input dimension, leading to some form of central limit theorem to hold. Finally, note that Gaussian universality has also been observed in dimension $784$ in supervised regression settings, see e.g. [47].

---

> > ### Comment · Reviewer_DrHV · 2023-08-21
> > **accept**
> >
> > Dear authors,
> > Thanks for your detailed answer. I now understand better your contributions and have raised my score.

---

### Official Review · Reviewer_zMEV · 2023-07-05

**Soundness:** 4 excellent
**Presentation:** 3 good
**Contribution:** 4 excellent
**Rating:** 7
**Confidence:** 3

**Summary:**

The authors set out to characterize the non-linear behavior of denoising auto-encoders (DAEs), for Gaussian mixtures, in the high dimensional limit with the number of hidden units being fixed. The authors particularly tease out the role of the skip-connection, compared to the reconstruction auto-encoder (RAE) which is known to essentially perform principal component analysis (PCA).

Using the replica method, the authors obtain closed-form expressions for the mean-squared error (MSE), as well as the cosine-similarity (w.r.t cluster means). The obtained formulae are supported by experiments on both synthetic and real data sets, clearly highlighting the role of the sample complexity and the noise level.

**Strengths:**

- Presents an effective characterization of (non-linear) DAE behavior on Gaussian mixtures, clearly explaining the role of the reconstruction and scaling components supported by compelling empirical evidence (also on real data).
- Draws a number of important conclusions, pointing to fruitful directions of future work, e.g., L200, L218, L278.

**Weaknesses:**

One would wish the long sequence of mathematical expressions could be made less opaque.
  - This is remedied by a seemingly complete and well-composed supplementary.

**Questions:**

**Technical comments:**
- L178: could you briefly explain the asymptotic dependence on $\alpha$?
- L212: could you elaborate on the choice of closely related classes?
- I wonder how the method fairs with a mixture of more than 2 classes.
  - Is it true that $K=2$ also for the MNIST experiments?

**Presentation comments:**
- L158: perhaps it helps to discuss this rationale before stating the assumptions.
- Understandably, it is an essential part of the contribution to provide those closed-form expressions, but I wonder which subset of the readers would benefit from having Eqs.13 and 14 in the main paper.

**Limitations:**

Dedicating more space to help make the theoretical concepts and techniques employed more accessible, i.e., to a broader set of readers, would have been greatly appreciated.

It is a limitation of this reviewer that I'm unable to assess the full scale of the theoretical derivations.

---

> ### Author Rebuttal · Authors · 2023-08-06
>
> We thank the reviewer for their appreciation of our work. We address below their questions:
>
> >One would wish the long sequence of mathematical expressions could be made less opaque.
>
> We will add further discussion beneath (13) and (14) in the revised manuscript, and provide qualitative insights into their meaning and implications.
>
> >L178: could you briefly explain the asymptotic dependence on $\alpha$
>
> The characterization of Corollary 3.5 involves the sample complexity $\alpha=n/d$, whereas previous studies typically assumed an infinite amount of available training data ($\alpha\to \infty$). Our work therefore allows to characterize the learning metrics when learning from finite data sets. It can further be shown from Corollary $3.5$ that the mse of RAEs asymptotically converges to its infinite data limit $\alpha \to\infty$ as $ O(\frac{1}{\alpha})$. We will add this computation and the relevant discussion in the supplementary material of the revised manuscript.
>
> >L212: could you elaborate on the choice of closely related classes?
>
> As discussed in section (l.278-290), in order to have a good denoising performance, the DAE has to learn the data structure, given by the cluster mean $\mu$. A priori, one expects the learning problem to be more challenging -and therefore more interesting- when the clusters are close and less distinguishable, i.e. when $\mu$ is small. Our choice of closely related classes for the real data experiments stems from that qualitative intuition. We will include further discussion in the revised version of the manuscript.
>
> >I wonder how the method fairs with a mixture of more than 2 classes. Is it true that also for the MNIST experiments?
>
> We indeed took for simplicity $K=2$ also in MNIST experiments, by retaining $2$ out of the $10$ classes, assuming that each class can be modelled by a Gaussian cluster.
>
> While we state the asymptotic formulae for any $K$ of order $1$, as the phenomenology of the results we obtain is already rich and novel for $K=2$, we left the exploration of the more general case for future work.
>
> >L158: perhaps it helps to discuss this rationale before stating the assumptions.
>
> We agree that mentionning that assumptions 3.1 and 3.2 can be relaxed before stating them would improve the readability. We will take this into account in the revised manuscript.
>
> >Understandably, it is an essential part of the contribution to provide those closed-form expressions, but I wonder which subset of the readers would benefit from having Eqs.13 and 14 in the main paper.
>
> (13-14) have been included in the main paper so that the result statement is self-contained, thereby avoiding the need of referencing equations placed elsewhere in the manuscript. We believe that the inclusion of further discussion thereof, as suggested in the reviewer's previous remark, will provide further insight into these equations and justify their position in the main text.
>
> >Dedicating more space to help make the theoretical concepts and techniques employed more accessible, i.e., to a broader set of readers, would have been greatly appreciated.
>
> We will devote further discussion to an explanation of the replica method, which we employ to derive the main result, so as to improve the readability for a broader audience.

---

> > ### Comment · Reviewer_zMEV · 2023-08-11
> > **Acknowledgement**
> >
> > Thank you for addressing my comments.
> >
> > I'm adjusting my score in light of the discussion with reviewer ZGsS.

---

### Author Rebuttal · Authors · 2023-08-06

We attach here a .pdf, containing additional figures which we refer to in the separate rebuttals.

---

### Comment · Area_Chair_Ff8y · 2023-08-13

Dear reviewers and authors,

Thank you very much for your work on this submission and its evaluation. Now that the authors have responded to the reviews, I strongly encourage the reviewers to acknowledge the review, to look at other reviews and rebuttals for this submission, and to adjust their scores if needed. Thanks to those that have already done so.

Authors have the possibility to reply if further questions are needed, until the 16th.

Thank you very much to all,
Area Chair

---

### Decision · Program_Chairs · 2023-09-21

**Decision:**

Accept (spotlight)

**Comment:**

Auto-encoders are often used in applications. Thus it is important to understand their theoretical properties. The authors managed to derive an asymptotic formula for the test error of 2-layer denoising auto-encoder with skip connections. The results are supported by numerical experiments.

In particular, they demonstrated that DAE can perform much better than PCA in terms of MSE. The MSE of DAE coincides with the asymptotic predictions, thus the optimisation problem demonstrates some good behaviour. The authors also identified situations when MSE for the denoising AE is close to the asymptotic performance of the oracle denoiser.

Overall, the results are interesting and describe some observed empirical behaviours of the DAE approach.